# EduVerse: A User-Defined Multi-Agent Simulation Space for Education Scenario

## Abstract

Reproducing cognitive development, group interaction, and long-term evolution in virtual classrooms remains a core challenge for educational AI, as real classrooms integrate open-ended cognition, dynamic social interaction, affective factors, and multi-session development rarely captured together. Existing approaches mostly focus on short-term or single-agent settings, limiting systematic study of classroom complexity and cross-task reuse. We present **EduVerse**, one of the first *user-defined* multi-agent classroom simulator supporting customizable environment, customizable agents, and multi-session evolution. A distinctive human-in-the-loop interface further allows real users to join the space. Built on a layered **CIE** (**C**ognition–**I**nteraction–**E**volution) architecture, EduVerse ensures individual consistency, authentic interaction, and longitudinal adaptation in cognition, emotion, and behavior—reproducing realistic classroom dynamics with seamless human–agent integration. We validate EduVerse in middle-school Chinese classes across three text genres, environments, and multiple sessions. Results show: **(i) Instructional alignment**: simulated Initiate-Response-Feedback (IRF) rates (0.34–0.55) closely match real classrooms (0.37–0.49), indicating pedagogical realism; **(ii) Group interaction and role differentiation**: network density (0.27–0.40) with about one-third of peer links realized, while human–agent tasks indicate a balance between individual variability and instructional stability; **(iii) Cross-session evolution**: the positive transition rate $R^+$ increase by 11.7% on average, capturing longitudinal shifts in behavior, emotion, and cognition and revealing structured learning trajectories; **(iv) Cross-disciplinary generalization**: without any additional tuning, IRF rates and peer-interaction topologies naturally adapt to the discourse characteristics of history instruction while preserving the core instructional structure, demonstrating robust cross-disciplinary transfer. Overall, EduVerse balances realism, reproducibility, and interpretability, providing a scalable platform for educational AI. The system will be open-sourced to foster cross-disciplinary research.

## 1 Introduction

A central challenge in human-centered AI is to simultaneously reproduce cognitive development, group interaction, and long-term evolution within virtual environments (Wang et al., 2024c; Parisi et al., 2019; Zheng et al., 2024; Chen & Liu, 2018; Zheng et al., 2025). While large language models (LLMs) excel at language understanding and immediate task completion, most research remains confined to static tasks or short-term interactions, falling short of capturing evolving cognition, stable behavioral styles, and socially dynamic processes (Maharana et al., 2024; Wang et al., 2025a; Tan et al., 2025; Li et al., 2025a). Similarly, multi-agent systems have primarily targeted structured games or fixed collaboration, lacking frameworks that support developmental agents whose cognition, personality, and social relations evolve naturally over time (Wang et al., 2024b; Ashery et al., 2024).

Educational settings, particularly classrooms, offer a natural testbed for modeling cognition, social interaction, and instructional feedback (Hattie & Timperley, 2007a; Poropat, 2009; Johnson et al., 1998). For example, Chinese language classes feature open-ended tasks, emotional nuance, and rich role-based interactions, ideal for studying development and group dynamics. Yet most intelligent tutoring systems and dialogue agents treat students as static performers (Anderson et al., 1995;

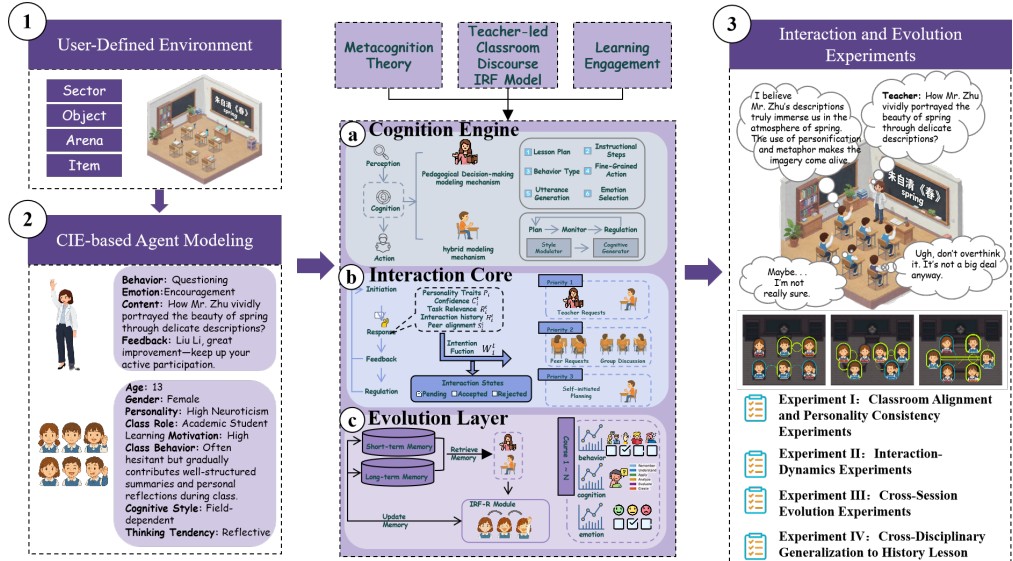

Figure 1: **Overview of EduVerse.** EduVerse comprises: (i) user-defined environment configuration; (ii) CIE-based agent modeling for teacher and student agents; (iii) interaction and evolution experiments spanning instructional alignment, group interaction, cross-session development, and cross-disciplinary generalization. Together, these components form a scalable, interpretable, and transferable multi-agent simulation platform for educational AI.

Nye et al., 2014; Lin et al., 2023), lacking persistent learner modeling, role-sensitive interaction, or longitudinal adaptation.

To overcome these limitations, we introduce **EduVerse**, one of the first *user-defined* multi-agent simulation space that supports environment customization through flexible physical layouts and seating arrangements, agent customization via a human-in-the-loop interface integrated with a layered **CIE** (**C**ognition–**I**nteraction–**E**volution) architecture, and session customization for modeling multi-lesson trajectories. In CIE, the cognition layer ensures individual consistency and instructional alignment, the interaction layer models priority-based authentic exchanges, and the evolution layer captures longitudinal changes in cognition, emotion and behavior. Together, these capabilities enable EduVerse to reproduce realistic classroom dynamics while supporting seamless human–agent interaction.

We instantiate EduVerse in middle school Chinese language classes—rich in open-ended discourse and emotional variation—and validate it through three CIE-aligned experiments: (i) Instructional alignment: simulated IRF rates (0.34–0.55) closely match real classrooms (0.37–0.49), indicating pedagogical realism. (ii) Group interaction: agent interaction networks reach a density of 0.27–0.40, approximating one-third of peer links and balancing individual variability with instructional stability. (iii) Cross-session evolution: positive transition rate $R^+$ improves by $11.7\%$, reflecting structured shifts in behavioral and cognitive engagement. (iv) Cross-disciplinary generalization: EduVerse adapts to history instruction while preserving IRF structures and peer dynamics, demonstrating robust transferability. Unlike conventional user-facing systems, the goal of EduVerse is to develop a foundational infrastructure for systematic educational simulations, shifting the evaluation focus from end-user experience to agent-level transparency and longitudinal interpretability. In sum, EduVerse captures individual- and group-level dynamics and reveals multi-dimensional learning trajectories across subjects and timescales.

**Contributions.** (1) We propose EduVerse, one of the first user-defined multi-agent classroom simulator, enabling reusable and customizable experimentation across tasks and disciplines. (2) We design the CIE architecture, which systematically models the cognitive, interactive, and evolutionary dynamics of developmental agents. (3) Through instantiated experiments, we demonstrate EduVerse's effectiveness in authentic educational contexts. As an extensible open framework, EduVerse redefines virtual classroom modeling and establishes a systematic, cross-disciplinary pathway for educational AI; it will be open-sourced to encourage transparency and collaboration.

## 2 RELATED WORK

**EduVerse** provides a user-defined multi-agent simulation space that supports cognitive development, group interaction, and long-term evolution in virtual classrooms. Although educational agents, multi-agent simulations, and LLM-based generation have advanced, a unified platform integrating these dimensions is still missing. We therefore review three research threads aligned with EduVerse's core dimensions (see App. B).

**Educational agents and virtual classrooms.** Early systems such as *Cognitive Tutor* (Anderson et al., 1995) and *SimStudent* (Matsuda et al., 2013) focused on skill acquisition and personalization, typically via rule- or model-based mechanisms (Christensen et al., 2011; Foley & McAllister, 2005; Carrington et al., 2011; Dotger et al., 2010). Teacher-training simulations used scripted virtual students as scaffolds but lacked adaptivity to feedback, peer influence, or classroom context (Kervin et al., 2006; Dieker et al., 2015; Delamarre et al., 2021; Shernoff et al., 2018; Özge Kelleci & Aksoy, 2021). Recent generative extensions enable task-level learning (Zhang et al., 2024a; Lee et al., 2023; Yue et al., 2025; Mollick et al., 2024; Markel et al., 2023b; Wang et al., 2025b; Fahid et al., 2024), but often omit emotional modeling, stylistic progression, and multi-agent coupling, limiting their suitability for open, dynamic classrooms.

**Multi-agent social simulations.** Works such as *Generative Agents* show that LLMs enhanced with memory, planning, and reflection can generate human-like social behaviors in sandbox settings (Xu et al., 2025; Arana et al., 2025; Park et al., 2023; Li et al., 2023; Chen et al., 2023a; Jinxin et al., 2023b). However, these focus on adult roles and informal contexts, overlooking classroom-specific structures such as IRF discourse, teacher–student roles, and goal alignment. They also lack mechanisms for knowledge progression tracking and temporal adaptivity.

**Personalized modeling and long-term coherence.** Persona conditioning and style control are widely used to maintain role consistency (Shao et al., 2023; Jiang et al., 2024; Wang et al., 2024d), with design patterns surveyed by Tseng et al. (2024). Yet long-term interactions often suffer from persona drift, leading to memory-based prompting (Zhong et al., 2023), style constraints (Roy et al., 2023), and metacognitive or reflective mechanisms (Madaan et al., 2023; Li et al., 2025b; Didolkar et al., 2024). Research on continual and lifelong learning also contributes to longitudinal coherence (Wang et al., 2024c; Parisi et al., 2019; Zheng et al., 2024; Chen & Liu, 2018; Zheng et al., 2025; Maharana et al., 2024; Wang et al., 2025a; Tan et al., 2025; Li et al., 2025a). However, these methods are mostly evaluated in single-agent or non-classroom contexts, rarely integrating group-level structures or pedagogically grounded evolution. *EduAgent* (Xu et al., 2024) models individual cognitive and metacognitive processes but lacks multi-agent coordination and group dynamics.

Prior work provides important components—feedback, generative behaviors, and role consistency—but remains fragmented across time scales, modeling levels, and educational contexts. **EduVerse** unifies these dimensions by integrating user-defined environments, agents, and multi-session settings within the **CIE** architecture, yielding a scalable and interpretable platform that coherently models individual behavior, group dynamics, and cross-session evolution.

## 3 EDUVERSE FRAMEWORK

**EduVerse** is a user-defined multi-agent simulation framework for educational settings, built to model the long-term cognitive, behavioral, and social dynamics of developing learners. It comprises three components: (i) a **user-defined environment** that configures layouts, seating, and interaction networks to support diverse instructional scenarios within a unified physical–social space (Sec. 3.1); (ii) **CIE-based agent modeling**, where student agents adopt a unified perception–cognition–action architecture with personalized embeddings and style modulation for cognitive coherence and expressive diversity, complemented by a human-in-the-loop interface for user customization (Sec. 3.2); and (iii) **interaction and evolution experiments** that integrate teacher-

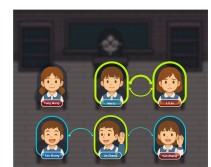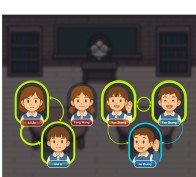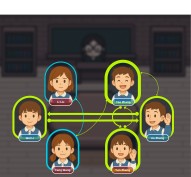

Figure 2: **Visualization of student group interactions across classroom environments.** The three panels correspond to: Lecture (left, traditional teacher-centered layout), Collab_Two_Tables (middle, two-group collaborative setting), and Round_Table (right, open discussion layout).

led guidance with student-initiated behavior to assess instructional alignment, group interaction, and cross-session evolution (Sec. 3.3). Together (Fig. 1), these components offer a scalable, interpretable, and transferable foundation for systematic educational analysis.

## 3.1 User-defined Environment

The environment module constructs an interaction space that integrates *physical constraints* with *social semantics*. We adopt a hierarchical spatial structure $\mathcal{Z} = \{\mathcal{Z}_S, \mathcal{Z}_A, \mathcal{Z}_O, \mathcal{Z}_I\}$: $\mathcal{Z}_S$ denotes functional *sectors* (e.g., teacher, student, activity zones), $\mathcal{Z}_A$ denotes localized *arenas* (e.g., a discussion circle or a podium), $\mathcal{Z}_O$ denotes interactive *objects* (e.g., blackboards, podiums, desks), and $\mathcal{Z}_I$ represents fine-grained *items* (e.g., textbooks, pens, chalk). This layered organization provides a unified mapping between physical distribution and pedagogical semantics, consistent with App. C.1.

Peer interaction is captured by a seat-adjacency graph $A^{seat} \in \{0, 1\}^{N \times N}$ defined as

$$A_{ij}^{seat} = \begin{cases} 1, & \text{if students } i \text{ and } j \text{ satisfy the adjacency rules,} \\ 0, & \text{otherwise,} \end{cases} \tag{1}$$

where the rules may combine distance $d(i, j)$, group membership $g(i)$, and layout-specific constraints. Researchers can instantiate different interaction topologies by editing configuration files, avoiding hard-coded seat links.

As shown in Fig. 2, under this unified definition, EduVerse implements three canonical classroom layouts: **Lecture** constrains peer links by distance and group, reflecting teacher-centered, largely unidirectional communication; **Round_Table** augments distance-based adjacency with face-to-face *opposite-seat* edges $j = \text{opp}(i)$ to encourage open peer dialogue; **Collab_Two_Tables** forms fully connected within-group subgraphs with no cross-group links, emphasizing intra-group collaboration and bounded social structure.

These layouts serve as illustrative cases rather than limitations. Users can freely customize the hierarchy $\mathcal{Z}$ and the adjacency-generation rules via configuration files to simulate classrooms of varying scales, tasks, and pedagogical styles. Leveraging the `seat_graph` mechanism, EduVerse tightly couples physical space with social semantics, providing a *realistic, controllable, and extensible* environment foundation for subsequent agent decision-making and group-level experiments.

## 3.2 CIE-based Agent Modeling

Agent modeling in CIE maps directly onto three layers: the Cognition layer models individual differences through the Plan–Monitor–Regulate (PMR) loop; the Interaction layer captures role-differentiated social behavior via extended IRF mechanisms; and the Evolution layer models cross-lesson adaptation through memory and phase updates. Together, these layers integrate individual, social, and temporal dynamics into a unified framework (see App. C.2–C.5).

### 3.2.1 Cognition Engine: Cognition-Driven Agent Decision Mechanism

All agents $\mathcal{A}_i$ follow the PCA architecture, formalized as:

$$\mathcal{A}_i^t : (\mathcal{O}_i^t, \mathbf{e}_i) \to a_i^t, \tag{2}$$

where $\mathcal{O}_i^t$ is the local observation extracted from the global state $\mathcal{S}^t$ via a perception function $\mathcal{P}_i$, and $\mathbf{e}_i = [\mathbf{p}_i; \mathbf{c}_i; \mathbf{m}_i]$ encodes personality traits, cognitive style, and motivation. The action is generated by a language model $a_i^t = f_{\text{LLM}}(\mathcal{O}_i^t, \mathbf{e}_i)$.

We adopt a lightweight interaction gate to decide whether an agent takes a turn at time $t$:

$$\mathcal{G}_i^t = \begin{cases} 1, & \text{if the scheduler assigns a teacher- or peer-directed turn to agent } i \text{ at time } t, \\ 0, & \text{otherwise (self-initiated behaviors may be scheduled separately).} \end{cases} \tag{3}$$

The cognitive loop consists of *Plan–Monitor–Regulate (PMR)*; the gate $\mathcal{G}_i^t$ determines whether the loop is executed at time $t$.

**Content–style separation.** To balance long-term consistency with expressive diversity, we decouple semantic planning from stylistic expression. Inspired by style transfer (Gatys et al., 2016; Deng et al., 2022), we adopt a two-component design: (1) a *style modulator* fine-tuned on educational data with InternVL (Chen et al., 2024b) that produces style-aware prompts conditioned on traits and task phase (e.g., hesitancy, verbosity, affective tone), rather than direct responses (see details in App. C.2.3); and (2) a *cognitive generator* that integrates the style prompt with $\mathcal{O}_i^t$, $\mathbf{e}_i$, and dialogue history to form a composite prompt for GPT-4 (Achiam et al., 2023), which focuses on semantic planning and content generation. This design improves **cognitive coherence** and **expressive diversity**, while offering an interpretable path for personality-conditioned behaviors.

**Role-specific parameterization.** While student agents $\mathcal{A}_S^i$ and the teacher agent $\mathcal{A}_T$ share the PCA backbone, they differ in input channels, gating, and prompting. As shown in Tab. A4, students are modulated by personality and willingness embeddings, whereas the teacher relies on scripted lesson plans and classroom metrics. This unified yet differentiated design supports dynamics from teacher-led instruction to student-initiated contributions.

### 3.2.2 INTERACTION CORE: GROUP INTERACTION AND BEHAVIORAL COORDINATION

To better capture classroom discourse, CIE extends the classic IRF (Initiation–Response–Feedback) cycle to $I_T^t \to R_S^t \to F_T^t \to \text{Regulate}_S^t$. At time $t$, the teacher agent $\mathcal{A}_T$ initiates a task $T^t$; student agents $\mathcal{A}_S^i$ respond, receive feedback, and then regulate subsequent actions, forming a micro-cycle aligned with instructional goals. Action selection follows Eq. 2 and is modulated by the interaction gate $\mathcal{G}_i^t$ (see Eq. 3) to determine participation.

**Intention function.** We conceptualize willingness and responsiveness as $\omega_i^t = \alpha_1 P_i + \alpha_2 C_i^t + \alpha_3 R_i^t + \alpha_4 H_i^t + \alpha_5 S_{ij}^t$. where $P_i$ denotes personality, $C_i^t$ confidence at time $t$, $R_i^t$ task relevance, $H_i^t$ interaction history, and $S_{ij}^t$ alignment with peer $j$; $\alpha_k$ are weighting coefficients. In practice, this abstraction is embedded into prompt design and scheduling to guide gating and response generation. Beyond teacher-led turns, students may self-initiate behaviors (e.g., questioning, head-up), enabling multi-party interaction and group discussion. The teacher monitors these dynamics and provides targeted or global feedback, closing the loop between pedagogical intent, social context, and adaptive regulation for analyzing group coordination and evolution.

### 3.2.3 EVOLUTION LAYER: CROSS-LESSON ADAPTATION AND EVOLUTION

CIE supports long-term, cross-lesson simulation through four mechanisms: knowledge progression, behavioral style regulation, instructional pacing control, and memory interaction flow.

**Knowledge progression.** Each student agent $\mathcal{A}_S^i$ initializes a knowledge state $\mathbf{s}_i^0$ derived from $\mathbf{e}_i$, which evolves as $\mathbf{s}_i^{t+1} = \mathcal{R}_i(\mathbf{s}_i^t, a_i^t, F_i^t)$, $\{\mathbf{s}_i^t\}_{t=1}^T$. where $\mathcal{R}_i$ adjusts state components (e.g., confidence, engagement) using annotated behavioral signals (e.g., Bloom level, response type) and structured feedback (positive, neutral, negative).

**Behavioral style regulation.** At each step, actions, reflections, affect, and cognitive states are logged in a structured growth log (JSON), enabling the tracking of style stability, engagement shifts, and recovery. These logs also feed subsequent scheduling and adaptation.

**Instructional pacing.** The teacher agent organizes instruction into *phases*, each comprising multiple *steps*. After each phase, pacing is updated as $\text{Phase}_{k+1} \leftarrow \text{Transition}(\text{Phase}_k, \{\mathbf{s}_i^t\}, \text{completion rate})$. with policy $\pi_T$ dynamically adjusting step granularity, tone, and targeting. In practice, cycles are initialized with 30 steps and then adapt to interaction dynamics, forming a closed loop for adaptive teaching control.

**Memory interaction flow.** To sustain cross-session continuity, CIE coordinates short- and long-term memory: long-term summaries are loaded at lesson start, short-term states are updated in real time, and aggregated records are written back after each session. This flow enables feedback-driven self-regulation and coherent developmental trajectories across lessons.

Taken together, the PCA backbone of CIE, coupled with mechanisms for cognition-driven decision-making, group interaction, and temporal evolution, yields behaviors that are *stable yet diverse* within

a lesson and *coherent* across lessons, providing a robust foundation for subsequent experiments and pedagogical analyses.

## 3.3 INTERACTION AND EVOLUTION EXPERIMENTS

This subsection formalizes EduVerse's experimental paradigm, enabling researchers to configure environments, agents, and multi-session tasks to study cognitive alignment, group interaction, and longitudinal development across subjects. Users can specify physical–social environment, agent personas, instructional scripts and so on. On this basis, four experiments can be defined: (i) Cognition-driven instructional alignment experiments; (ii) Group interaction analysis and role differentiation experiments; (iii) Cross-session evolution and long-term development experiments; (iv) Cross-disciplinary generalization experiments. All experiments are logged and evaluated with unified metrics (e.g., IRF discourse structure, network density/centrality, positive transition rate $R^+$), ensuring interpretability and reproducibility. In addition, EduVerse provides a *human-in-the-loop* interface that admits real students or teachers alongside virtual agents (all agent names are randomly generated and contain no identifiable or referential information), enabling simulation, causal testing, and validation within a single, low-cost, controllable, and interpretable framework. In Sec. 4, we demonstrate these capabilities in a **junior secondary classroom**, highlighting EduVerse's applicability and research value.

## 4 EXPERIMENTAL DESIGN AND EVALUATION

To demonstrate EduVerse's customizability and cross-context transferability, we conduct four experiments across junior-secondary Chinese lessons and an additional Renaissance history lesson, covering cognitive alignment, group interaction, long-term development, and cross-disciplinary generalization. This setup enables evaluation under distinct discourse styles and instructional structures. Full implementation details are provided in App. D. The four experiments align with EduVerse's core capacities: (1) **Experiment I** (Sec. 4.1): evaluates classroom authenticity and personality-conditioned alignment under customized environment settings; (2) **Experiment II** (Sec. 4.2): investigates group interaction and individual influence, and validates human–agent interaction through the open interface; (3) **Experiment III** (Sec. 4.3): tracks students' behavioral, emotional, and cognitive trajectories across four consecutive lessons to illustrate cross-session evolution; (4) **Experiment IV** (Sec. 4.4): tests whether EduVerse generalizes to new cultural contexts, narrative instructional styles, and knowledge structures beyond Chinese language arts.

### 4.1 EXPERIMENT I: ENVIRONMENT CUSTOMIZATION FOR COGNITION-DRIVEN INSTRUCTIONAL ALIGNMENT

Experiment I tests whether customized environments can reproduce realistic classroom dynamics while retaining personality-driven behaviors described in Sec. 3.2. We instantiate three user-defined environments (Sec. 3.1) under a teacher-led mode and run ablations to assess key module contributions.

Table 1: Average IRF distributions in simulated environments (**Sim**) vs. real classrooms (**Real**) across three text genres.

| Genre | Setting | I | R | F | IRF$_{rate}$ |
|---|---|---|---|---|---|
| *Lyrical Prose* | Sim | 0.454 | 0.166 | 0.293 | 0.336 |
| | Real | 0.513 | – | 0.703 | 0.486 |
| *Argumentative Essay* | Sim | 0.482 | 0.207 | 0.335 | 0.554 |
| | Real | 0.417 | – | 0.583 | 0.417 |
| *Foreign Fiction* | Sim | 0.310 | 0.230 | 0.407 | 0.379 |
| | Real | 0.367 | – | 0.515 | 0.367 |

**IRF discourse patterns.** To assess whether EduVerse reproduces the structural logic of teacher–student dialogue rather than exact numerical matching, we compute the IRF rate as $\text{IRF}_{\text{rate}} = \frac{1}{T}\sum_{t=1}^{T}\mathbf{1}(I_T^t = 1 \wedge R_S^t = 1 \wedge F_T^t = 1)$, where $I_T^t$, $R_S^t$, and $F_T^t$ indicate the occurrence of teacher initiation, student response, and teacher feedback. Real classroom IRF data are sourced from the national Smart Education Platform and annotated through a standardized expert-reviewed protocol (App. D.3.2). As shown in Tab. 1, simulated IRF rates (0.336–0.554) lie within the structural range of real classrooms (0.367–0.486). Although numerical variation emerges across teachers, genres, and discourse styles, the consistent presence of complete IRF cycles indicates that EduVerse captures the underlying interaction structure shaping classroom discourse. A qualitative comparison (Tab. A8) further shows alignment in the I and R stages, with similar questioning and response patterns. Divergence mainly appears in the F stage: real teachers provide more open-ended or affective feedback, whereas simulated teachers respond

more concisely. These differences reflect natural variation in teaching styles and do not affect the overall IRF structural consistency (App. D.3).

**BEC (Behavior-Emotion-Cognition) across environments.** To quantify students' BEC tendencies, we compute the normalized frequency of each category (App. D.2.3) over the class: $\text{BEC}(c) = \frac{1}{T} \sum_{t=1}^{T} \mathbf{1}(x_t = c)$, where $x_t$ denotes the BEC label at timestep $t$ and $T$ is the total number of timesteps. This metric captures the *distribution of learner states*, rather than correctness, and is applied consistently across all comparisons in this subsection. Fig. 3 shows clear environment-dependent patterns: collaborative layouts yield the most positive emotion (0.547) and higher-order cognition (0.261); round-table layouts show more disengagement (0.254) and lower interaction (0.357); and lecture layouts reflect traditional classrooms with dominant lower-order cognition (0.819). EduVerse effectively reproduces environment-shaped classroom dynamics.

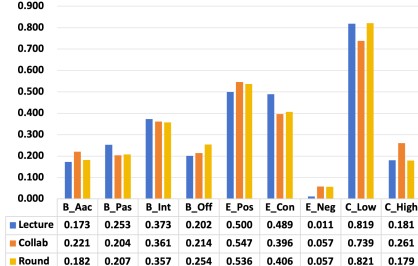

| | B_Aac | B_Pas | B_Int | B_Off | E_Pos | E_Con | E_Neg | C_Low | C_High |
|---|---|---|---|---|---|---|---|---|---|
| Lecture | 0.173 | 0.253 | 0.373 | 0.202 | 0.500 | 0.489 | 0.011 | 0.819 | 0.181 |
| Collab | 0.221 | 0.204 | 0.361 | 0.214 | 0.547 | 0.396 | 0.057 | 0.739 | 0.261 |
| Round | 0.182 | 0.207 | 0.357 | 0.254 | 0.536 | 0.406 | 0.057 | 0.821 | 0.179 |

Figure 3: **Distribution of students' BEC across environments.** Collaborative layouts promote positivity and higher-order cognition, round-table increases disengagement, and lecture maintains passive, lower-order patterns.

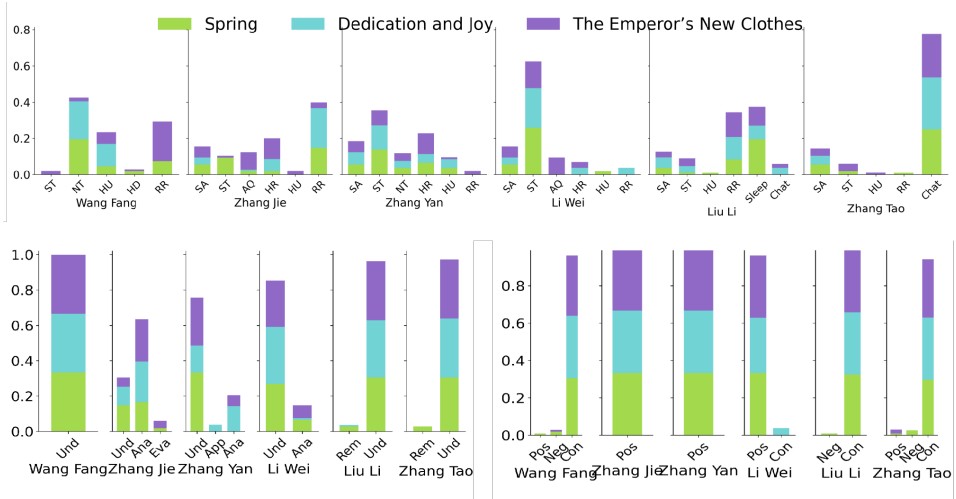

Figure 4: **Stable BEC patterns across genres within individual students.** Despite genre variation, individual BEC patterns remain stable and trait-consistent: highly extraverted students show active engagement and varied cognition, whereas low-openness or low-conscientiousness students tend toward disengagement, lower-order cognition, and confusion.

**Personality-driven stability.** As shown in Fig. 4, students maintain stable BEC distributions across genres, consistent with their traits: **Zhang Jie** (high extraversion) stays active and positive with varied cognition, whereas **Liu Li** (low openness) and **Zhang Tao** (low conscientiousness) show disengagement, lower-order cognition, and frequent confusion. These patterns demonstrate EduVerse's ability to preserve individual consistency and personality-conditioned behaviors.

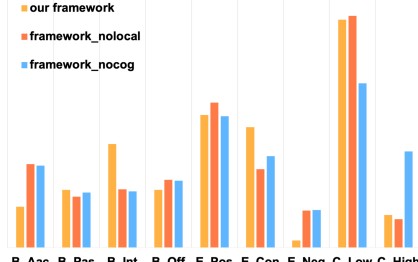

Figure 5: **Ablation study.** Removing the stylization module (framework_nolocal) increases negative emotions, while removing the PMR module (framework_nocog) inflates higher-order cognition and overly active behaviors.

**Ablation study.** Fig. 5 further validates the necessity of key modules. Removing the stylization module results (framework_nolocal) in evenly distributed emotions with amplified negativity, deviating from real classrooms where positive and confused states dominate. Removing the PMR module (framework_nocog) exaggerates higher-order cognition and active behaviors, resembling expert reasoning rather than gradual student development. Together, these results show that stylization ensures realistic emotional patterns, while the PMR module enforces educationally consistent cognitive

and behavioral decisions, and their integration is indispensable for reproducing authentic classroom dynamics.

**Summary.** Experiment I demonstrates that EduVerse reproduces authentic discourse, captures environment effects, preserves personality stability, and validates key modules, providing a solid basis for studying classroom design and instructional modes.

### 4.2 EXPERIMENT II: AGENT CUSTOMIZATION FOR GROUP INTERACTION ANALYSIS AND ROLE DIFFERENTIATION

This experiment evaluates EduVerse's ability to model interaction described in Sec. 3.2, moving from group-level networks to individual influence, and finally testing human–agent integration via the open interface.

**Group Interaction Analysis.** We model classroom interactions as undirected graphs, using density ($D = \frac{2E}{N(N-1)}$) and average degree ($k = \frac{2E}{N}$). As shown in Tab. 2, density (0.267–0.400) and degree (1.2–1.667) indicate that 27–40% of ties are realized, reflecting realistic yet localized participation in teacher-led classrooms. Genre–environment effects also appear: argumentative essays and prose show lower density in Lectures, while fiction remains around 0.3 across settings, underscoring the narrative appeal of storytelling. Overall, genre and layout jointly shape engagement.

Table 2: Group interaction analysis across lessons and environments. Values report nodes, edges, density, and average degree of the interaction graph. **Bold** marks the highest values per lesson.

| Lesson | Env. | Nodes | Edges | Density | Avg. Deg. |
|---|---|---|---|---|---|
| *Foreign Fiction* | *Lecture* | 6 | 5 | **0.333** | **1.667** |
| | *Collab* | 5 | 3 | 0.300 | 1.200 |
| | *Round* | 6 | 5 | **0.333** | **1.667** |
| *Argumentative Essay* | *Lecture* | 5 | 3 | 0.300 | 1.200 |
| | *Collab* | 5 | 4 | **0.400** | **1.600** |
| | *Round* | 6 | 5 | **0.333** | **1.667** |
| *Lyrical Prose* | *Lecture* | 6 | 4 | 0.267 | 1.333 |
| | *Collab* | 5 | 3 | 0.300 | 1.200 |
| | *Round* | 6 | 5 | **0.333** | **1.667** |

**Individual Influence Analysis.** We computed four directed metrics: *in-degree* (attention), *out-degree* (initiative), *degree centrality* (activity), and *betweenness* (bridging). As shown in Tab. 3, *Dedication and Joy* reveals mode-dependent roles. In the Lecture, Zhang Jie had a high in-degree but no initiative, while Zhang Tao and Zhang Yan showed the opposite. In Collab, reciprocity increased, with Zhang Tao emerging as a connector and Zhang Jie shifting to side-talk. In Round, roles were decentralized, yet Zhang Tao became most central (highest out-degree, degree, and betweenness), whereas Liu Li was marginalized. (see Fig. 2 for the visualization of student interactions). These results illustrate how classroom organization reshapes roles, moving individuals from peripheral to core positions.

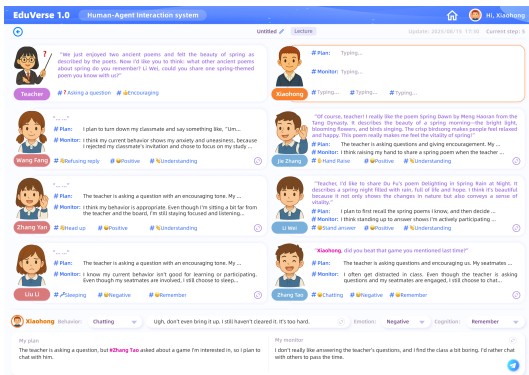

Figure 6: **Human–Agent interaction interface.**

Table 3: Distribution of students' network centrality indicators in *Dedication and Joy* across environments. Values are normalized to $[0, 1]$. **Deg.** = degree centrality; **Betw.** = betweenness centrality.

| Student | Lecture | | | | Collab | | | | Round | | | |
|---|---|---|---|---|---|---|---|---|---|---|---|---|
| | In | Out | Deg. | Betw. | In | Out | Deg. | Betw. | In | Out | Deg. | Betw. |
| Li Wei | 0.25 | 0.25 | 0.50 | 0 | 0.25 | 0.25 | 0.50 | 0 | 0.20 | 0.20 | 0.40 | 0 |
| Liu Li | 0.25 | 0.25 | 0.50 | 0 | 0.25 | 0.25 | 0.50 | 0 | 0.00 | 0.20 | 0.20 | 0 |
| Zhang Tao | 0.00 | 0.25 | 0.25 | 0 | 0.50 | 0.25 | 0.75 | 0.083 | 0.20 | 0.60 | 0.80 | 0.15 |
| Zhang Jie | 0.50 | 0.00 | 0.50 | 0 | 0.25 | 0.25 | 0.50 | 0 | 0.40 | 0.20 | 0.60 | 0.10 |
| Zhang Yan | 0.00 | 0.25 | 0.25 | 0 | 0.25 | 0.50 | 0.75 | 0.083 | 0.20 | 0.20 | 0.40 | 0 |
| Wang Fang | – | – | – | – | – | – | – | – | 0.40 | 0.00 | 0.40 | 0 |

**Human–Agent Interaction.** To evaluate human–agent integration, we evaluate human–agent integration through the EduVerse visual interface (Fig. 6), which enables ChatGPT-style interaction

with student agents. Four interaction types, including peer chat, peer academic response, teacher Q&A, and teacher intervention, were tested within this interface (Fig. A3,App. D.4). Results aligned with personality traits: **Zhang Tao** (talkative) responded most (0.53–0.73) and initiated conversations in Collab/Round, while **Zhang Jie** (high-extraversion) responded less (0.27–0.40). Teacher agents succeeded in all Q&A and interventions, ensuring robust instructional control. These findings confirm EduVerse's capacity for seamless human integration with role-driven realism.

**Summary.** EduVerse captures group patterns, individual traits, and human–agent integration, validating its ability to model authentic and adaptive classroom interactions.

### 4.3 EXPERIMENT III: SESSION CUSTOMIZATION FOR CROSS-SESSION EVOLUTION AND LONG-TERM DEVELOPMENT

Experiment III tests whether virtual students show progressive development described in Sec. 3.2 by mapping BEC states to ordered levels and treating upward transitions as positive shifts. Across four sessions (*Spring* I–II, *Dedication and Joy* I–II), we assess long-term engagement and learning trajectories.

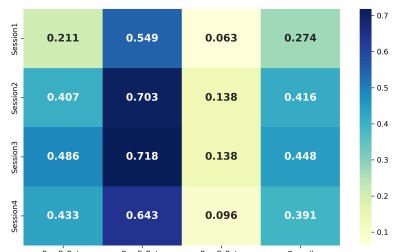

**Session-level Evolution.** We quantify progression using the positive–transition rate. For a state sequence $\{s_t\}$ with priority $P(s_t)$, a transition is positive when $P(s_{t+1}) > P(s_t)$. Let $T^+ = \sum_t \mathbf{1}[P(s_{t+1}) > P(s_t)]$, $R^+ = T^+/T$. As shown in Fig. 7, behavior improves rapidly across sessions, emotion rises early and stabilizes, and cognition grows more slowly, reflecting its need for sustained accumulation. These patterns indicate that EduVerse captures realistic dynamics where behavior and affect shift quickly, while cognitive development progresses gradually.

Figure 7: **Positive transition trends in cross-session evolution.** Rates of positive shifts in BCE increase over time, with behavior improving most rapidly, emotion rising steadily, and cognition progressing gradually.

**Individual-level Evolution.** As shown in Fig. 8 and Tab. A13, students exhibit clear divergence across four sessions. **Wang Fang** improves steadily (0.057→0.205); **Zhang Jie** remains consistently strong (0.483–0.603) with ceiling-level emotion (E_Pos = 1.000); **Zhang Yan** accelerates late, reaching 0.923 in behavior; **Li Wei** maintains high emotion ($E\_Pos \geq 0.966$) but shows limited cognitive gains; **Liu Li** progresses gradually overall (0.011→0.179); and **Zhang Tao** fluctuates markedly (0.080–0.256). These trends show EduVerse's ability to capture differentiated and realistic learner trajectories.

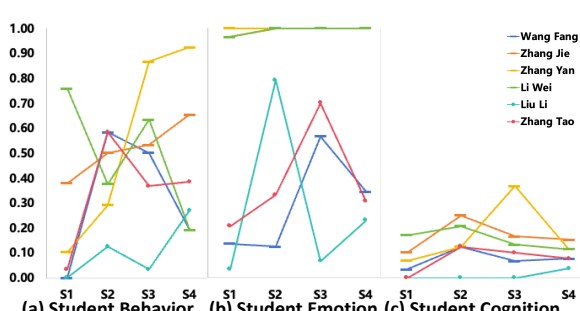

Figure 8: **Individual trajectories of positive shifts across sessions.** Students display differentiated developmental paths in behavior, emotion, and cognition, closely aligned with personality traits, validating EduVerse's capacity to model personalized learning evolution.

**Ablation on the Regulate Module.** To evaluate the role of BEC regulation, we remove the Regulate stage and compare cross-session positive transitions. As shown in Tab. 4, overall gains drop from 0.117 to 0.098, driven by declines in behavior (+7.0%) and emotion (+5.1%). The larger cognitive gain without regulation reflects unstable high-level jumps, mirroring the PMR ablation in Experiment I, rather than genuine progress. With regulation enabled, transitions become more stable and educationally plausible, confirming its importance for realistic long-horizon learning.

Table 4: Ablation on the Regulate.

| Variant | $\Delta B_{\text{Pos}}$ | $\Delta E_{\text{Pos}}$ | $\Delta C_{\text{Pos}}$ | $\Delta$All |
|---|---|---|---|---|
| w/o R | 0.152 | 0.043 | 0.098 | 0.098 |
| Full | 0.222 | 0.094 | 0.033 | 0.117 |

**Summary.** EduVerse captures long-term learning evolution: behavior and emotion improve quickly, while cognition develops more slowly, reflecting realistic developmental variation.

## 4.4 EXPERIMENT IV: CROSS-DISCIPLINARY GENERALIZATION TO HISTORY LESSON

To assess cross-disciplinary adaptability, we add a Renaissance history lesson, distinct from Chinese in cultural background and teaching style, while keeping all agent settings fixed, testing whether EduVerse adapts without additional tuning.

**IRF discourse structure**. Real classrooms show clear subject differences (Tab. 5): Chinese exhibits denser IRF patterns (0.423), whereas history is more explanatory and therefore sparser (0.333). EduVerse reproduces this divergence, yielding 0.423 for Chinese and 0.226 for history. This alignment with real trends shows that EduVerse captures discipline-specific interaction intensity while preserving the core IRF structure, demonstrating robust cross-disciplinary transfer.

Table 5: Comparison of simulated and real-classroom IRF patterns across subjects. "H" = History; "C" = Chinese; "Sim" = simulated classrooms; "Real" = real classroom data.

| Setting | Subjects | I | R | F | $IRF_{rate}$ |
|---|---|---|---|---|---|
| **Sim** | H_lecture | 0.205 | 0.269 | 0.385 | 0.282 |
| | H_collab | 0.256 | 0.226 | 0.415 | 0.282 |
| | H_round | 0.143 | 0.071 | 0.329 | 0.114 |
| | H_avg | 0.201 | 0.189 | 0.376 | 0.226 |
| | C_avg | 0.416 | 0.201 | 0.345 | 0.423 |
| **Real** | H | 0.359 | – | 0.513 | 0.333 |
| | C | 0.432 | – | 0.600 | 0.423 |

**Group interaction Analysis**. As shown in Tab. 6, student networks maintain substantial connectivity across layouts, with densities ranging from 0.333 in round and collaborative settings to 0.5 in the lecture layout—closely matching interaction levels observed in Chinese lessons. Beyond the numeric patterns, the visualized networks in Fig. 9 reveal clear topological distinctions: history lessons produce shorter, localized chains, whereas literature lessons form broader, discussion-oriented clusters.

**Summary**. These findings indicate that while disciplinary context shapes the magnitude of IRF patterns, EduVerse consistently generates coherent and adaptive interaction dynamics and network topology across domains.

Table 6: Group interaction analysis for the simulated history lesson. "Nodes" denote student agents; "Edges" represent realized peer interactions. Density and average degree reflect interaction intensity under Lecture, Collaborative, and Round layouts.

| Env. | Nodes | Edges | Density | Avg. Deg. |
|---|---|---|---|---|
| Lecture | 4 | 3 | 0.500 | 1.500 |
| Collab | 6 | 5 | 0.333 | 1.667 |
| Round | 6 | 5 | 0.333 | 1.667 |

## 5 CONCLUSION

We introduced EduVerse, a user-defined multi-agent simulation framework that unifies cognition, interaction, and evolution to model developmental student agents. Trait-conditioned learners generated by EduVerse behave consistently yet adaptively across instructional contexts. Across four experiments, EduVerse reproduces key classroom phenomena: realistic IRF structures and personality-driven behaviors (Exp. I), coherent peer-interaction patterns and smooth human–agent integration (Exp. II), and meaningful long-term development with an 11.7% in-

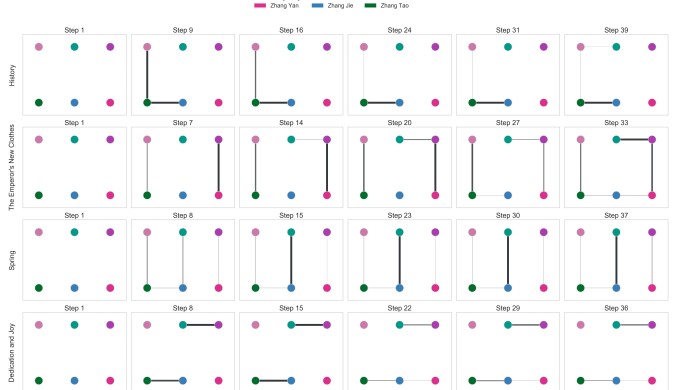

Figure 9: **Interaction network under the lecture layout for four subjects.** Nodes represent student agents and edges denote peer interactions. History lessons yield more localized exchanges, while literature lessons form broader discussion chains, reflecting disciplinary variation with stable core interaction patterns.

crease in positive transitions (Exp. III). A Renaissance history lesson further demonstrates cross-disciplinary generalization, with IRF density and interaction topology reorganizing in discipline-consistent ways while core pedagogical structures remain stable (Exp. IV). Together, these results show EduVerse as a scalable and transferable platform for modeling learning processes and supporting research in adaptive learning, human-in-the-loop teaching, and cross-disciplinary educational AI.

ETHICS STATEMENT

This study adheres to established ethical standards for research involving educational data and human–AI interaction. All real classroom materials used in this work, including video content and instructional transcripts, were obtained exclusively from nationally released open educational platforms. These resources are officially published by educational authorities, publicly accessible, and contain no private or restricted information. The instructional transcripts used for analysis include only teacher–student dialogue from publicly available lessons and do not contain identifiable student images or personal data.

No new data involving minors were collected for this research. The study did not conduct experiments, interviews, or interactions with students, nor did it gather additional information from schools, teachers, or learners. Aside from publicly released classroom transcripts, all classroom dialogues, behavioral annotations, and emotion labels were generated by large language models and do not contain any real personal information.

Human–agent interaction experiments in this work were limited to internal system functionality and robustness testing. All interactions were performed solely by members of the research team, with no recording, storage, or analysis of personal information. The purpose of these interactions was to validate the technical performance of the system rather than to study human subjects. As such, these activities do not constitute human-subject research.

All data used in this study have been anonymized and are employed strictly for academic research. Raw transcripts from public educational platforms are not released or reproduced within the paper. No data are used for commercial purposes. The research design follows principles of transparency, privacy protection, and minimal risk.

REPRODUCIBILITY STATEMENT

To support reproducibility, we will release the EduVerse codebase, environment settings, preprocessing scripts, and trained models. Processed resources and evaluation protocols will also be provided. All resources will be made available upon acceptance.

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

# APPENDIX

# APPENDIX CONTENTS

## A    THE USE OF LARGE LANGUAGE MODELS (LLMS)

During manuscript preparation, Large Language Models (LLMs) were employed solely for language refinement and stylistic polishing.

## B    COMPREHENSIVE RELATED WORKS

### B.1    EDUCATIONAL AGENTS AND VIRTUAL STUDENT SIMULATIONS

Modeling virtual student agents has long been a central thread in educational AI research (Dai & Ke, 2022; Chou et al., 2003; Chheang et al., 2024). As early as the 1990s, researchers introduced "teachable agents" to support teacher training and foster learning by teaching. Representative examples include VanLehn et al.'s work in physics tutoring (VanLehn et al., 1994) and the Betty's Brain system (Biswas et al., 2005), which formalized the learning-by-teaching paradigm. However, these systems relied heavily on scripted behavioral policies and static knowledge-update mechanisms, limiting their ability to capture temporal dynamics in student behavior and affect. Similarly, traditional intelligent tutoring systems emphasized knowledge mastery (e.g., skill tracing) but paid little attention to developmental trajectories.

The advent of large language models (LLMs) has enabled a new generation of educational agents, characterized by open-ended interaction, behavioral diversity, and learner-like imperfections (Russin et al., 2024; Chu et al., 2025; Moore et al., 2023). Recent work has leveraged prompt-conditioned LLMs to simulate students with varying ability levels, personalities, and misconceptions (Wang et al., 2024a; Glaese et al., 2022). For example, Lu and Wang (Lu & Wang, 2024) proposed the Generative Students framework to assess item difficulty through simulated learners; Markel et al. (Markel et al., 2023a) introduced GPTeach to enhance teacher training with diverse LLM-generated student responses; and Jin et al. (Jin et al., 2025a) developed TeachTune, which evaluates instructional agents against a spectrum of student personas, highlighting how pedagogical strategies adapt to personality-driven differences.

Beyond individual interactions, researchers have explored classroom-level simulations with multi-agent LLM frameworks. AgentVerse (Chen et al., 2023b) and CGMI (Jinxin et al., 2023a) enable heterogeneous agents to engage in collaborative and instructional roles. Building on this, Sim-Class (Zhang et al., 2024b) incorporates a class manager to coordinate Initiation–Response–Feedback (IRF) dialogues between teacher and student agents, achieving high realism: agents respond to teacher prompts, initiate follow-ups, maintain turn-taking, and even exhibit emergent behaviors such as spontaneous group discussions and peer-led task completion. These advances mark an important step from scripted interactions toward socially grounded, self-organizing classroom learning.

Nevertheless, existing systems remain limited in modeling long-term behavioral evolution, affective regulation, and stylistic coherence, with most constrained to single-session interactions. In contrast, **EduVerse** provides a unified, multi-level simulation platform that supports user-defined modeling of physical environments, agent configurations, and cross-session evolution. Its open interfaces allow researchers to flexibly configure classroom layouts, seating arrangements, and interaction networks, and systematically examine learners' cognitive development and group dynamics. Through this design, EduVerse bridges the gap between scripted tutor–learner simulations and scalable, development-oriented classroom modeling.

### B.2    LLM-DRIVEN MULTI-AGENT SOCIAL SIMULATIONS

The rise of large language models has fueled a new wave of multi-agent simulations, where agents act as generative entities capable of open-ended dialogue and socially coherent behavior (Guo et al., 2024; Gao et al., 2024; Hua et al., 2023). A landmark example is *Generative Agents* by Park et al. (Park et al., 2023), which deployed 25 GPT-powered agents in a virtual town. Each agent followed a daily routine, formed human-like relationships, and even organized a collective event without external control, illustrating how memory and reflection mechanisms support emergent behaviors. This work sparked broad interest in agent-based modeling grounded in persistent memory and self-regulation.

Subsequent studies extended this paradigm to collaborative and adversarial settings. ChatDev (Qian et al., 2023) and MetaGPT (Hong et al., 2023) simulated professional teams by embedding role-

specific prompting and structured communication flows, thereby improving task-level coherence. Wang et al. (Wang et al., 2023) showed that even single-agent simulations can emulate multi-agent reasoning by activating multiple internal personas through self-dialogue, enhancing creativity and logical reasoning.

Competitive scenarios provide further evidence of emergent strategy. Xu et al. (Xu et al., 2023) employed GPT agents in the game Werewolf to model deception, persuasion, and coalition dynamics. In economics, Horton (Horton, 2023) and Aher et al. (Aher et al., 2023) embedded LLM agents in negotiation and public goods games, demonstrating rational decision-making alongside human-like biases. Together, these studies underscore the feasibility of LLM agents as proxies for large-scale human social behavior.

Nonetheless, these systems largely focus on adult interactions and informal contexts, lacking the structured pedagogical mechanisms required for educational environments. Most do not support teacher–student dialogue based on IRF cycles, nor do they capture learner development over time or adapt to knowledge progression and group dynamics. In contrast, **EduVerse** introduces LLM-driven multi-agent simulation into education, integrating structured classroom discourse (e.g., IRF cycles), metacognitive regulation, and cross-session evolution. This provides a scalable and interpretable platform for modeling instructional interactions and systematically examining the developmental trajectories of learners.

### B.3 PERSONALIZED MODELING AND BEHAVIORAL CONSISTENCY MECHANISMS

Maintaining long-term consistency of personality, language style, and behavioral patterns remains a central challenge in LLM-driven multi-agent environments (Guo et al., 2024; Li et al., 2024b; Tran et al., 2025). Despite their generative fluency, LLMs often suffer from "persona drift" in extended interactions, where role-specific traits weaken or deviate over time (Xu et al., 2023). For example, a student agent initially designed to be introverted may gradually adopt assertive conversational patterns, undermining the credibility of long-term simulations. Such issues arise from limited identity retention and emotional coherence, motivating research on role stabilization mechanisms.

Early dialogue systems attempted to preserve character identity through structured profiles and task-specific memory modules. PersonaChat (Zhang et al., 2018), for instance, embedded fixed persona facts, while later work introduced memory modules to retain role-consistent traits across turns (Ouyang et al., 2022). These approaches proved effective in short sessions but struggled with complexity and duration. In LLM-based agents, persona-conditioned prompting became a common strategy, though it often failed under topic shifts or multi-phase tasks (Xu et al., 2023).

Recent advances emphasize fine-tuning and reward-based alignment to improve role fidelity (Ji et al., 2025b; Chen et al., 2025). CharacterGLM (Zhou et al., 2023) and Ditto (Lu et al., 2024) enhanced intra-role consistency by training on persona-labeled dialogues or generating synthetic role-specific corpora. Reinforcement Learning from Human Feedback (RLHF) further penalized off-character outputs, reinforcing behavioral alignment. However, these methods face scalability challenges, including data costs and generalization trade-offs.

To mitigate reliance on supervised correction, self-monitoring strategies have emerged (Li et al., 2025d; Behore et al., 2024). Ji et al. (Ji et al., 2025a) introduced a role-aware reflection loop that allows models to detect and revise misaligned outputs. Coupled with contrastive training, this approach significantly improved long-term role consistency. Memory-augmented designs provide additional scaffolding: systems log key interactions and behavioral states, with summarization or vector retrieval supporting continuity in future outputs (Park et al., 2023; Ouyang et al., 2022).

These developments are particularly salient in education. To simulate realistic learners, researchers have designed student agents with distinct cognitive and non-cognitive traits. TeachTune (Jin et al., 2025a) showed that teacher agents adjust feedback strategies depending on student profiles such as confidence or anxiety, while Li et al. (Li et al., 2025c) modeled "imperfect learners" by injecting errors into outputs, prompting teacher agents to practice remediation. Such findings highlight the pedagogical utility of stable yet differentiated learner personas.

**EduVerse** advances this line of work by integrating profile-driven planning, memory-aligned regulation, and behavior-consistent generation into its Cognition–Interaction–Evolution framework. Compared to single-agent approaches, EduVerse achieves both intra-agent coherence and inter-agent

variation at the classroom scale, providing a scalable and interpretable solution for educational simulation.

Table A1: Comparison of multi-agent educational simulation frameworks.

| Framework | Environment Customization | | | Agent Customization | | | | Interaction and Evolution Experiments | | | |
| --- | --- | --- | --- | --- | --- | --- | --- | --- | --- | --- | --- |
| | Environment | Physical Layouts | Seating | Theory-based Cognition Modeling | Teacher-Student | Agent Interaction (Group) | Human-Agent | Long-time Evolution | Real-vs-Simulated Comparison | Group-Level Analysis | Individual-Level Analysis |
| AgentSchool Jin et al. (2025b) | ✓ | × | × | ✓ | ✓ | ✓ | × | ✓ | ✓ | ✓ | ✓ |
| PEERS Arana et al. (2025) | ✓ | × | × | ✓ | ✓ | ✓ | × | × | × | × | ✓ |
| Contextual Agents Xu et al. (2025) | × | × | × | ✓ | × | × | × | ✓ | ✓ | ✓ | ✓ |
| Simulating Classroom Zhang et al. (2024a) | × | × | × | ✓ | ✓ | ✓ | ✓ | ✓ | × | ✓ | ✓ |
| MathVC Yue et al. (2025) | × | × | × | ✓ | × | ✓ | ✓ | × | × | ✓ | ✓ |
| AgentVerse Chen et al. (2023b) | ✓ | × | × | × | ✓ | ✓ | ✓ | ✓ | × | — | |
| GPTeach Markel et al. (2023b) | × | × | × | ✓ | × | × | ✓ | × | × | × | ✓ |
| Generative Agent Lee et al. (2023) | ✓ | × | × | ✓ | × | × | ✓ | × | × | × | ✓ |
| EduVerse | ✓ | ✓ | ✓ | ✓ | ✓ | ✓ | ✓ | ✓ | ✓ | ✓ | ✓ |

## C  DETAILED INFORMATION FOR EDUVERSE FRAMEWORK

### C.1  DETAILED DESCRIPTION OF ENVIRONMENT MODULE

#### C.1.1  HIERARCHICAL SPATIAL ORGANIZATION

EduVerse models classroom environments using a four-tiered hierarchical spatial structure $\mathcal{Z} = \{\mathcal{Z}_S, \mathcal{Z}_A, \mathcal{Z}_O, \mathcal{Z}_I\}$, corresponding to **Sector**, **Arena**, **Object**, and **Item**. This design supports pedagogically meaningful interactions and dynamic behavior generation, and provides semantic alignment between agent perception, spatial context, and task execution.

At the top level, a **Sector** partitions the virtual classroom into functional instructional zones, e.g., *Teacher Sector*, *Student Sector*, and *Activity Sector*. Within each sector, an **Arena** specifies localized interactive regions—such as a discussion circle or a teaching podium—that constrain mobility and determine perceptual access to nearby objects. **Objects** embedded within Arenas (e.g., desks, blackboards, presentation screens) serve as anchors for attention and instructional actions. **Items** (e.g., chalk, textbooks, notebooks) are the most granular perceivable units and constitute the basic elements of fine-grained interaction.

For example, in the *Teacher Podium Arena*, a blackboard and lectern support behaviors such as lecturing and board writing. In contrast, a *Student Group Arena* contains desks and personal learning materials, enabling small-group collaboration or individualized learning. A complete layout instance typically includes multiple sectors, each containing several arenas populated with pedagogically structured objects and items (e.g., chalk, erasers, water cups), collectively forming a coherent four-level spatial graph (see Tab. A2). This hierarchical structure supports classroom configurations ranging from lecture-centric layouts to collaborative or round-table setups, and allows real-time adaptation of spatial roles, interaction boundaries, and perception zones as classroom dynamics evolve.

Table A2: **Hierarchical Classroom Environment Structure**

| Sector (ID) | Arena | Object | Items |
|---|---|---|---|
| Teacher Zone (10001) | Resource Display Area | Blackboard | Chalk, Blackboard Eraser |
| Teacher Zone (10001) | Podium Area | Podium Desk | Cup, Textbook, Mobile Phone, Chalk |
| Teacher Zone (10001) | Instruction Area | Electronic Whiteboard | — |
| Student Zone (20001) | Group A Area | Student $A_{1...N}$ | Textbook, Cup, Pencil Case, Pen, Backpack |
| Student Zone (20001) | Group B Area | Student $B_{1...N}$ | — |
| Activity Zone (30001) | Storage Area | Cleaning Supplies | Broom, Mop |
| Activity Zone (30001) | Storage Area | Daily Utilities | Clock, Cabinet, Water Dispenser |

Note: This table instantiates the 4-level spatial structure in **EduVerse**—Sector, Arena, Object, and Item. Each arena defines a functional subspace containing interactive objects for perception and behavior planning.

#### C.1.2  INTERACTION AFFORDANCES AND CONSTRAINTS

The spatial hierarchy also functions as an interaction scaffold that governs how agents perceive, move, and act. Layouts map to pedagogical strategies: lecture-centric configurations emphasize teacher-led, unidirectional communication, while round-table or collab-based layouts promote peer collaboration. The "Sector–Arena–Object–Item" hierarchy accommodates these variations and enables dynamic spatial adaptation during simulation.

In the current implementation, spatial location primarily conditions behavioral availability. For example, only students within a *Presentation Arena* can access teacher content in real time; in group tasks, shared Arena–Object associations trigger peer-based interactions. EduVerse thus enforces mode-specific constraints (lecture, presentation, group discussion), with corresponding action permissions.

Beyond this, the framework is designed to be extensible. For instance, each student can be assigned a personalized `perception_config` (e.g., perceptual radius and maximum trackable items), and collision constraints or spawn configurations can be integrated to enhance physical realism. While not activated in the current experiments, these extensions illustrate the scalability of EduVerse towards more fine-grained behavioral planning and environment fidelity.

### C.1.3 SEAT_GRAPH FORMALIZATION AND LAYOUT TEMPLATES

We define the seat graph as an unweighted adjacency matrix $A^{\text{seat}} \in \{0,1\}^{N \times N}$, where

$$A_{ij}^{\text{seat}} = \begin{cases} 1, & \text{if } i \text{ and } j \text{ satisfy the layout-specific adjacency rule,} \\ 0, & \text{otherwise.} \end{cases}$$

Formally, EduVerse provides three canonical layouts:

- Lecture Adjacency is primarily defined within rows, restricted to students sitting close to each other:
$$A_{ij}^{\text{seat}} = \mathbb{I}\big[d(i,j) \leq \tau_d \ \wedge \ \text{row}(i) = \text{row}(j)\big].$$

- Round Table Students are seated around circular tables; adjacency includes both immediate neighbors and face-to-face counterparts:
$$A_{ij}^{\text{seat}} = \mathbb{I}\big[j = \text{neighbor}(i) \ \vee \ j = \text{opp}(i)\big].$$

- Collab_Two Tables Within-group students are fully connected, while across-group edges are suppressed:
$$A_{ij}^{\text{seat}} = \mathbb{I}\big[g(i) = g(j)\big].$$

Thus, the seat graph provides a baseline, layout-dependent topological structure:

$$A^{\text{seat}} = \begin{cases} A_{\text{Lecture}}^{\text{seat}}, & \text{if layout = Lecture,} \\ A_{\text{Round}}^{\text{seat}}, & \text{if layout = Round Table,} \\ A_{\text{Collab}}^{\text{seat}}, & \text{if layout = Collab\_Two\_Tables.} \end{cases}$$

This structure is a binary, unweighted topology that captures only peer adjacency. It does not yet encode higher-order social or instructional factors. Instead, it serves as a foundation upon which *cognitive planning* can later be modeled by incorporating teacher and board visibility.

### C.1.4 COGNITIVE PLANNING AND DISTANCE EFFECTS

The physical distances encoded in the seat graph directly affect students' cognitive planning. While the seat graph itself captures only peer proximity relations, the cognitive planning process requires a richer consideration of spatial constraints, including student–student, student–teacher, and student–board distances. Importantly, these spatial factors constitute only *one component* of cognitive planning; other psychological, pedagogical, and contextual variables also play a crucial role, but here we explicitly highlight the influence of the physical environment.

**Distance-based cognitive factors**

Let $p_i \in \mathbb{Z}^2$ denote the spatial position of student $i$. We define three forms of distance measures relevant to cognitive planning:

- **Peer distance:**
$$d(i,j) = \|p_i - p_j\|_2, \quad i \neq j,$$
capturing the Euclidean distance between students $i$ and $j$.

- **Teacher distance:**
$$d_T(i) = \|p_i - p_T\|_2,$$
where $p_T$ denotes the teacher's position. This distance reflects the cognitive accessibility of the teacher, which is essential for attention allocation and interaction.

Table A3: Environment configuration parameters in EduVerse. Parameters specify spatial layout, interaction rules, and perception constraints.

| Parameter | Description |
|---|---|
| Grid size | $30 \times 20$ discrete lattice |
| Layout type | Lecture / Round Table / Collaborative (two tables) |
| Table radius | Default $= 3$ grid units |
| Table gap | Default $= 10$ grid units (between two tables) |
| Objects | Students, teacher, podium, board, tables |
| Peer adjacency | Edge $(i, j)$ if $d(i, j) \leq \tau_d = 4.5$ |
| Group adjustment | Threshold $+1.0$ if $g(i) = g(j)$ |
| Round Table rule | Add cross-table "opposite" edges |
| Collaborative rule | Within-table complete graph; no inter-table edges |
| Perception radius | $r = 8$ grid units |
| Perception capacity | $\kappa = 5$ objects per student |
| Collision/occupancy | One object per grid cell; invalid moves rejected |

- **Board distance:**

$$d_B(i) = \|p_i - p_B\|_2,$$

where $p_B$ denotes the board's position. This measure captures the visibility and salience of instructional materials.

**Integration into cognitive planning**

In cognitive planning, a student's effective engagement is influenced by a combination of these distances. We define a cognitive planning function

$$C(i) = f\big(\{d(i,j)\}_{j \neq i},\ d_T(i),\ d_B(i),\ \Omega(i)\big),$$

where $f(\cdot)$ aggregates peer proximity, teacher distance, and board distance together with additional factors $\Omega(i)$ (e.g., individual motivation, prior knowledge, or task demands).

Thus, physical proximity among peers (as encoded in the seat graph) provides the structural baseline, while teacher and board distances add instructional and attentional dimensions. These spatial factors, combined with non-spatial determinants $\Omega(i)$, jointly shape the student's cognitive planning process.

### C.1.5 CONFIGURATION SCHEMA AND PHYSICAL REALISM

Our environment is instantiated on a two-dimensional grid of size $30 \times 20$. Each classroom layout is parameterized by a configuration schema, including layout type (*Lecture*, *Round Table*, or *Collaborative Two Tables*), table radius, table spacing, and object coordinates (students, teacher, podium, board, and tables).

Peer adjacency is determined by a Euclidean distance threshold $\tau_d = 4.5$, with an additional tolerance of $+1.0$ if two students belong to the same group. In the *Round Table* layout, opposite students are explicitly connected, while in the *Collaborative* layout, within-table students form a complete subgraph and inter-table connections are suppressed.

Perception is bounded by a radius ($r = 8$) and a capacity constraint ($\kappa = 5$), ensuring that each student can only attend to a limited number of peers or objects. Collision masks and occupancy constraints are implicitly enforced by the grid representation, while actions violating spatial feasibility (e.g., moving into occupied cells or colliding with fixed objects) are rejected. This ensures reproducibility and interpretability of spatially grounded interactions.

### C.2 DETAILED DESCRIPTION OF COGNITION ENGINE

### C.2.1 THEORETICAL FOUNDATIONS OF PCA ARCHITECTURE

CIE adopts a three-stage **Perception–Cognition–Action (PCA)** architecture to simulate agent decision-making (Chen et al., 2024a; Davis & Gao, 2003; Hancock et al., 1996; Yao, 2020). This

Table A4: Functional differentiation between student and teacher agents in **CIE**.

| Component | Student Agent $\mathcal{A}_S^i$ | Teacher Agent $\mathcal{A}_T$ |
|---|---|---|
| Perception Input | Teacher actions, peer behavior, social requests, seat graph, interaction graph | Class-wide metrics, participation logs |
| Cognitive Objective | Interpret prompts, decide on participation | Track engagement, guide task flow |
| Action Output | Interactive action, Active action, Passive action, Off-task action | Task instruction, Interactive teaching acts |
| Triggering Condition | Willingness + gating check | Scripted plans + real-time updates |
| Prompt Conditioning | *Personalized traits:* personality, style, motivation | *Fixed templates:* instructional role |

design is grounded in both classical AI agent models and foundational theories in educational psychology, capturing the trajectory from environmental sensing to goal-directed behavior.

From a psychological perspective, PCA aligns with the Atkinson–Shiffrin model of information processing (Atkinson & Shiffrin, 1968; Izawa, 1999; Cheng & Schwing, 2022) and resonates with constructivist and metacognitive theories of learning (Gunstone, 1992; Bonanno, 2004). It integrates affect, cognition, and behavior, emphasizing how learners perceive, interpret, and regulate actions in evolving classroom contexts.

Within this structure, the **Perception** stage gathers contextual cues such as teacher prompts, peer behaviors, and environmental signals. The **Cognition** stage transforms these inputs into internal reasoning and action intent, and the **Action** stage executes verbal, physical, or social behaviors accordingly.

A central innovation in CIE is the integration of a metacognitive loop—**Plan, Monitor, Regulate**—within the cognition stage. Student agents are not merely reactive: they proactively plan whether and how to act, monitor the clarity and outcomes of their behaviors, and regulate subsequent strategies based on feedback. For example, an agent may decide whether to speak (**Plan**), evaluate the appropriateness of its utterance (**Monitor**), and adjust behavior after receiving teacher feedback (**Regulate**). This bidirectional cycle links teacher interventions with student adaptations, producing temporally grounded learning trajectories that approximate authentic development.

The same PCA structure extends to teacher agents, where cognitive stages underpin pedagogical decisions such as goal-setting, feedback selection, interpretation of class-level signals, and adjustment of emotional tone. This unified yet role-differentiated framework enables interpretable and traceable behaviors across heterogeneous agents.

In summary, the PCA architecture provides a theoretically grounded, modular, and extensible control mechanism, serving as the cognitive backbone of the CIE framework.

### C.2.2 PERSONALITY-CONDITIONED COGNITION

To capture individualized learning dynamics, **CIE** embeds a triadic psychological model—**personality traits, cognitive style, and learning motivation**—into each student agent's cognition engine. This design ensures that agents are not only responsive to instructional context but also conditioned by stable psychological dispositions, thereby enabling heterogeneous patterns of planning, action execution, and self-regulation.

This modeling approach is anchored in established educational psychology theories. (1) The *Big Five Personality Traits* inform dispositional tendencies such as agreeableness or conscientiousness (Gerber et al., 2011; Komarraju et al., 2011). (2) Dimensions like *Field Dependence–Independence* and *Impulsivity–Reflection* characterize cognitive style, shaping how learners process information and regulate decisions (Swinnen et al., 1986; Jamieson, 1992; Yang & Chen, 2023). (3) Motivational dimensions are guided by *Self-Determination Theory* and *Expectancy-Value Theory*, which capture intrinsic versus extrinsic drivers of engagement (Yue & Lu, 2022; Gladstone et al., 2022; Loh, 2019). Together, these constructs define how agents interpret classroom signals, prioritize goals, and sustain effort over time, even under identical task conditions.

Each student agent is initialized with a structured personality profile encoded in JSON format, including its trait vector, cognitive preferences, and motivational disposition. During execution, these parameters are accessed primarily in the *Plan* stage of the PCA loop, guiding intention formulation and prompt conditioning. For instance, a highly agreeable agent tends to produce affirming or cooperative responses during group discussion, whereas a low-conscientiousness agent is more likely to show task-avoidant tendencies. Reflective learners adopt deliberate monitoring strategies and gradual adjustment, while impulsive learners often exhibit frequent shifts and inconsistent participation.

By embedding these individualized profiles into the cognitive control cycle, CIE generates agents that combine **behavioral realism** with **systematic variability**. The heterogeneity among agents promotes the simulation of authentic classroom dynamics, such as uneven participation or role differentiation. Moreover, these embedded traits provide interpretable signals for teacher agents, enabling differentiated instruction and context-sensitive feedback strategies. In this way, the personality-conditioned cognition module supports both fine-grained behavioral modeling and pedagogical analysis, bridging psychological theory with computational simulation.

---

**Virtual Student Profile:** Wang Fang

**Basic Information:**

- Age: 13    Gender: Female
- Spawn Position: (5, 1)
- Perception Config: Range = 3, Max Tracked Items = 5

**Learning Engagement:**

- Behavior = Head-down reading
- Emotion = Positive
- Cognition = Understanding

**Personality Traits (Big Five):**

- Personality Type: High Neuroticism
- Neuroticism: 0.9, Extraversion: 0.5, Agreeableness: 0.5, Openness: 0.5, Conscientiousness: 0.5

**Cognitive Style:**

- Field Independent
- Reflective

**Learning Motivation:** High

**Class Role:** Academic-Oriented Student

**Behavior Profile:**

- **Language Style:** Hesitant, emotionally influenced; uses fillers like "um", backtracks responses, shows fragmented expression.
- **Class Behavior:** Anxious under pressure; avoids raising hand but listens attentively in groups; completes work slowly but with logical clarity and strong motivation.
- **Learning Preference:** Prefers independent tasks, reflective practice; likes using notes or diagrams for organizing knowledge; enjoys self-paced deep exploration.
- **Teacher Guidance:** Reduce performance anxiety via timely feedback; provide structured task breakdowns and staged learning goals; encourage written or non-verbal responses.
- **Role Description:** Acts as knowledge summarizer in groups; organizes ideas clearly; prefers private channels for expressing ideas to build confidence.

### C.2.3 PERSONALITY-DRIVEN STYLE MODULATION

To ensure that agents maintain trait-aligned consistency in language and behavior generation across multi-turn interactions, **CIE** introduces a dual-module mechanism: the **Style Modulator** and the **Cognitive Generator**. This design balances the generalization strength of large language models (LLMs) with the need for personality-conditioned expression, thereby ensuring that each virtual student consistently exhibits distinctive communicative styles and cognitive tendencies during simulation.

**Style Modulator.** We adopt multimodal large models (InternVL (Chen et al., 2024b), Qwen (Bai et al., 2023), MiniCPM (Hu et al., 2024), and LLaVa (Li et al., 2024a)) as the linguistic foundation for virtual students. Although these models support visual inputs, in this work we use only their language-generation modules, as the core task is to produce persona-consistent classroom utterances within EduVerse. We choose VLMs rather than pure LLMs to ensure future extensibility, since EduVerse will later incorporate multimodal instructional materials and nonverbal classroom cues. To select the primary generator, we first evaluated all candidates on Chinese language–instruction tasks, assessing text understanding and text reconstruction accuracy on middle-school materials. InternVL achieved the strongest baseline performance. We then conducted persona fine-tuning (Hu et al., 2021) using more than 6,000 real classroom utterances covering multiple instructional phases and questioning modes. Post–fine-tuning evaluation, combining human judgments and large-scale GPT-4 scoring, showed that InternVL produced the most stable and distinctive learner styles. We therefore adopt InternVL as the main student model and fine-tune it using LoRA.

**Cognitive Generator.** At each stage of the metacognitive loop—*Plan, Monitor, Regulate*—the system issues role-specific prompts that drive context-aware decision-making. These prompts integrate the agent's personality profile, current instructional signals, and historical behavioral traces, thereby enabling dynamic adjustment of cognitive strategies. Prompt templates are separately designed for student and teacher agents, supporting the complete perception–cognition–action loop described in earlier sections.

Overall, this dual-module design not only enhances the realism and stability of personality-conditioned expression but also provides an interpretable framework for modeling diverse educational behaviors in simulated classrooms.

---

**Plan Prompt for Virtual Student Agent**

You are a student named *{self.name}*, and your character profile is (*{self.profile}*). Based on your persona and the following contextual information, reflect on your learning goal and behavioral plan in the current class session:

- Current physical distance: *{self.profile['dist_teacher']}, {self.profile['dist_blackboard']}*
- Teacher's current behavior: *{teacher_behavior}*
- Teacher's emotional tone: *{teacher_emotion}*
- Teacher's instructional content: *{teacher_content}*
- Other students' previous standing responses: *{stu_response}*
- Previous student questions to the teacher: *{stu_request}*
- Learning status of neighboring classmates: *{neighbors}*
- Relevant instructional objects: *{objects}*
- Your recent learning experiences and regulation suggestions: *{memory[:3]}*
- Your personality type: *{self.personality_type}*
- Your learning behavior style: *{self.class_behavior}*
- Your classroom role: *{self.class_role}*
- Current lesson content: *{self.shared_state['lesson_content']}*

Based on the above information and your personality traits, describe your behavioral plan using a first-person reasoning chain. The behavior must fall into one of the following four categories:

---

- **Active behaviors:** taking notes, raising hand
- **Passive behaviors:** listening attentively, reading silently, reading aloud
- **Interactive behaviors:** side talk with peers, asking the teacher questions
- **Disengaged behaviors:** sleeping on desk, chatting with others

**Instructions:**

1. Carefully consider the teacher's state, peer learning status, your persona, and recent self-regulation experience.

2. Your behavior should align with your habitual classroom pattern—e.g., if you are disengaged by nature, plan accordingly, unless your self-regulation history indicates change.

3. Output must follow a first-person reasoning chain—concise and limited to a single sentence.

4. Select **only one final behavior** from the list (e.g., taking notes / raising hand / listening attentively / reading silently / reading aloud / side talk / asking questions / sleeping / chatting).

5. Strictly follow the output format below:
   The teacher is asking about spring-related poetry, with a calm tone. My neighbors are actively raising their hands XXXX.
   Final behavior: side talk with peers

---

**Monitor Prompt for Virtual Student Agent**

You are a student named *{self.name}*, and your character profile is *({self.profile})*. Based on your persona and the following contextual information, reflect on the current instructional situation and evaluate whether your understanding and behavior are appropriate:

- Teacher's behavior: *{teacher_behavior}*
- Neighboring student states: *{neighbors}*
- Your previous plan: *{plan_output}*
- Recent memory fragments: *{memory}*
- Your current behavior: *{action_result.get('behavior', 'unknown')}*
- Feedback received: *{action_result.get('response', 'none')}*
- Your personality type: *{self.personality_type}*
- Your classroom behavior habit: *{self.class_behavior}*

Based on the information above, assess your current emotional and cognitive state under the behavior you just performed. Emotions should be classified into three categories: **positive**, **negative**, or **confused**. Cognitive states should follow Bloom's taxonomy and be selected from: **Remembering**, **Understanding**, **Applying**, **Analyzing**, **Evaluating**, **Creating**.
**Instructions:**

1. Briefly monitor your behavioral process using a first-person perspective.

2. Output your emotional and cognitive category results strictly in the following format:
   I think XXX
   Emotion: XXX
   Cognition: XXX

---

**Regulate Prompt for Virtual Student Agent**

You are a student named *{self.name}*, and your character profile is *{self.profile}*. Based on your persona, your recent monitoring reflection *({monitor_output})*, memory state *({memory})*,

and the teacher's feedback (*{teacher_feedback}*), reflect on what adjustment strategy you should adopt next (e.g., asking a question, taking notes, communicating with peers, etc.).

**Additional contextual information:**

- Teacher's current behavior: *{teacher_behavior}*
- Your personality type: *{self.personality_type}*
- Your classroom learning habits: *{self.class_behavior}*
- Current social interaction partners: *{self.shared_state["social_interaction"]}*
- Peers who rejected your conversation attempts: *{social_request_reject}*

**Instructions:**

1. Briefly analyze how you would like to adjust your learning strategy in the next step.
2. The output should be concise and stated in the first person, using only one sentence.

**Example:** "I got distracted just now, and the teacher called on me; I hope to refocus and pay better attention."

---

**Prompt for Virtual Teacher: Reconstructing Lesson Plan into Instructional Phases**

You are an experienced and professional middle school Chinese language teacher. Below is the instructional plan for the lesson titled *{self.lesson_id}*. Please read the lesson plan carefully and reconstruct it according to the required structure.

**Task:** Carefully review the content of the *{self.lesson_id}* lesson plan. Reorganize the instructional content into five standard teaching phases: **Lesson Introduction, New Content Instruction, Knowledge Consolidation, In-Class Practice, and Lesson Summary**.

The lesson consists of a total of 30 time steps. For each phase:

- Provide a concise summary of the instructional content to be covered, written in paragraph form (no bullet points), within 50 characters (or equivalent).
- Allocate a specific number of time steps to each phase.

**Lesson Plan Content:**

```
--- Lesson Plan Start ---
{self.lesson_plan_text}
--- Lesson Plan End ---
```

**Please strictly follow the output format below:**

```
Lesson Introduction: XXXX, Steps: XXX
New Content Instruction: XXXX, Steps: XXX
Knowledge Consolidation: XXXX, Steps: XXX
In-Class Practice: XXXX, Steps: XXX
Lesson Summary: XXXX, Steps: XXX
```

---

**Prompt for Virtual Teacher: Instructional Step Planning**

You are an experienced and professional middle school Chinese language teacher. You are currently teaching the course *{self.lesson_id}* and are now in the instructional phase titled *"{teaching_phase}"*. Please plan the instructional content for this specific phase.

This phase is expected to span *{total_steps}* instructional steps. Below is a brief summary of the content you are expected to teach in this phase: *{teaching_phase}*

**Instructions:**

1. Break down the above instructional content into **{total_steps}** individual teaching steps.
2. Each step should consist of one concise sentence (no more than 20 Chinese characters or equivalent length in English).
3. Ensure that each step aligns clearly with the goal of the current instructional phase.

4. Avoid repetition, vague statements, or logical leaps between steps.

5. Use the following format for output:

```
1. xxx
2. xxx
3. xxx
...
```

---

**Prompt for Virtual Teacher: Selecting Instructional Behavior Type**

You are an experienced and professional middle school Chinese language teacher. You are currently teaching the lesson *{self.lesson_id}*, and are now in the instructional phase *"{phase}"*. The content planned for the current time step is: *{teaching_step_content}*.

Please determine the most appropriate instructional behavior type based on the following contextual information:

- Student participation, emotional, and cognitive states: *{perception_result['teaching']['statistics']}*
- Your most recent teaching feedback: *{teacher_feedback}*
- Student self-regulation output: *{student_regulation}*
- Your last behavior category: *{category}*

**Instructions:**

1. Consider whether to maintain or shift from your previous behavior category (*{category}*).

2. It is generally preferred to vary your instructional strategy across consecutive time steps for richer pedagogical dynamics.

3. Select one behavior category from the following list:
   - Classroom Instruction (e.g., lecturing, giving directions)
   - Classroom Interaction (e.g., expressing emotion, praise, incorporating student input, asking questions, giving criticism, organizing discussion)
   - Classroom Behavior Management (e.g., addressing students sleeping or chatting)

4. **Important:** Output only the final selected category name without numbers. *For example:* `Classroom Interaction`

---

**Prompt for Virtual Teacher: Selecting Fine-Grained Teaching Action**

You are an experienced and professional middle school Chinese language teacher. You are currently teaching the lesson *{self.lesson_id}* and are in the instructional phase *"{phase}"*. The planned instructional content for the current time step is: *{teaching_step_content}*.

You have decided to perform a *{category}* type of teaching behavior. Based on the following contextual information, please select one fine-grained instructional action that aligns with your selected behavior type.

- Current phase: *{phase}*
- Behavior type selected: *{category}*
- Student participation and emotional state: *{perception_result['teaching']['statistics']}*

**Available fine-grained behaviors by category:**

- **Classroom Instruction:** lecturing, giving directions

- **Classroom Interaction:** expressing emotion, praising, adopting student input, asking questions, giving criticism, organizing group discussion
- **Classroom Behavior Management:** addressing students sleeping, addressing students chatting

**Instructions:**

- Select only one fine-grained behavior that best suits the context.
- Do not include any numbering or extra explanation.
- *Example:* lecturing

---

**Prompt for Virtual Teacher: Generating Instructional Utterance**

You are an experienced and professional middle school Chinese language teacher. You are currently teaching the lesson *{self.lesson_id}*, and are in the instructional phase *"{phase}"*. The instructional content planned for this time step is approximately: *{teaching_step_content}*. Your selected teaching behavior is: *{behavior}*. Please generate the instructional utterance you will deliver to students based on the following context:

- Current instructional phase: *{phase}*
- Teaching behavior: *{behavior}*
- Planned teaching content for this time step: *{teaching_step_content}*
- Current student states: *{perception_result['teaching']['statistics']}*
- Recent content already covered: *{[h['content'] for h in history[-3:]]}*
- Text material being taught: *{self.shared_state['lesson_content']}*

**Instructions:**

1. Your utterance must logically follow previously delivered content and align with the current teaching goal.
2. **Avoid repeating prior statements.**
3. If your behavior is **lecturing**, you may deliver up to 5 informative sentences focused on knowledge delivery.
4. For all other behavior types, limit the output to **2–3 sentences**.
5. You may refer to specific students based on what you know about them (e.g., call them by name), except when the behavior is **organizing classroom discussion**. In that case, pose an open-ended prompt to all students, optionally setting up a collaborative or competitive task.

**Please output only the generated utterance (no metadata).**

---

**Prompt for Virtual Teacher: Selecting Instructional Emotion**

You are an experienced and professional middle school Chinese language teacher. You are currently teaching the lesson *{self.lesson_id}* and are in the instructional phase *"{phase}"*. The teaching content for the current time step is: *{content}*, and your current instructional behavior is: *{behavior}*.
Please determine the most appropriate emotional tone for this moment based on the following student emotional state:

- Student emotional distribution: *{emo}*

**Instructions:**

- Choose one emotional tone from the following three options:
  - **Encouraging**
  - **Critical**

> – **Neutral**
> - Please output **only one** of the three tones, with no additional explanation.
> - *Example output: Encouraging*

To ensure stylistic and semantic consistency across multi-turn dialogues, **CIE** introduces a mechanism that aligns dialogue history with continuity of style. Specifically, the output generated at each *Regulate* stage is propagated forward into the subsequent *Plan* stage, thereby preserving coherence in both personality expression and cognitive trajectory. Response generation follows a hybrid pipeline: a LoRA-tuned LLM first produces a personality-aligned draft, which is then refined by GPT-4 to guarantee pedagogical validity and cognitive plausibility. This dual-stage strategy integrates personality intent, cognitive structure, and contextual awareness into a unified and adaptive response process.

---

**Prompt for Personality-Driven Style Modulation**

You are an assistant that can revise student responses based on their personality characteristics while preserving their individual speaking style.
The student's personality type is: *{self.personality_type}*. Below is a one-sentence sample response in their characteristic style. Your task is to refine it based on the student's persona without altering the style.

**Student Information:**
- Name: *{self.name}*
- Personality Type: *{self.personality_type}*
- Language Style: *{self.language_style}*
- Classroom Behavior: *{self.class_behavior}*

**Context:**
- Teacher's question: *{query}*
- The student plans to raise their hand to respond.
- Their drafted response: *{draft}*
- Learning plan for this time step: *{plan}*

**Instructions:**
1. Evaluate whether the drafted response is reasonable. If it is not, you may disregard it and instead generate a new answer based on the student's personality traits.
2. Middle school students typically speak concisely—your revised response should follow similar length and tone as the sample.
3. Your final output should:
   - Maintain the student's original speaking style;
   - Reflect their personality and classroom behavior;
   - Align with the current instructional context.
4. **Output only the revised response.**

---

## C.3 DETAILED DESCRIPTION OF SOCIAL SITUATEDNESS

### C.3.1 THEORETICAL EXPANSION OF IRF PARADIGM

The Initiation–Response–Feedback (IRF) model, first proposed by Sinclair and Coulthard, remains a cornerstone in classroom discourse analysis (Waring, 2009; Rustandi, 2017). It organizes interaction into three stages: the teacher *initiates* with prompts or questions (I), the student *responds* verbally or behaviorally (R), and the teacher provides *feedback* (F) in the form of evaluation or elaboration. While concise and widely applicable, this formulation is limited in capturing the cognitive and social dynamics of modern classrooms.

Under constructivist and inquiry-oriented pedagogies, students act as reflective learners rather than passive recipients (Walker & Shore, 2015; Renninger, 2024). They monitor, regulate, and socially negotiate their learning in response to both internal states and external cues—capacities insufficiently represented in the original IRF framework.

To address this gap, **CIE** extends IRF into a four-phase structure, termed IRF-R, by introducing a **Regulation** stage. After receiving feedback, students engage in metacognitive processing: reassessing their performance, adjusting goals, and modifying strategies in light of emotional state, peer interaction, and task relevance. In this expanded cycle, *Initiation* stimulates attention and motivation, *Response* generates verbal or behavioral engagement, *Feedback* reinforces or redirects cognition, and *Regulation* transforms feedback into adaptive behavior.

The IRF-R paradigm thus supports multi-turn interaction loops that conceptualize learning as a continuous cycle of stimulation, expression, feedback, and self-adjustment. It enables teacher agents to track not only immediate responses but also downstream learning adjustments, thereby improving the interpretability of student behavior and supporting deeper trajectories of engagement.

### C.3.2 IMPLEMENTATION OF SOCIAL PRIORITIZATION

In the **CIE** multi-agent system, student agents must simultaneously handle instructional prompts from teacher agents and spontaneous peer-initiated interactions during lessons. To resolve conflicts among these competing inputs, we introduce a **social prioritization mechanism** inspired by gated decision control. This ensures that at each time step, every agent responds to the interaction with the highest *pedagogical relevance* (see Alg. A.1).

---

**Algorithm A.1: Social Priority Gating Mechanism**

**Input:** Local perception $s_{i_p}^t$ of agent $i$, shared state $S$, social threshold $\theta$

**Output:** Behavioral decision $a_i^t$

1   **if** *TeacherRequestExists*$(i, s_{i_p}^t)$ **then**
2     $a_i^t \leftarrow$ respond to teacher (e.g., "stand and answer");
3     **RejectSocialRequest**$(i, \text{'Teacher Priority'})$;
4   **else if** *SocialRequestExists*$(i, S)$ **then**
5     $r_i^t \leftarrow$ **GetSocialRequest**$(i, S)$;
6     $H_{i,j}, S_{i,j} \leftarrow$ **AnalyzeChatHistory**$(i, r_i^t.\text{from\_id})$;
7     $W_i^t \leftarrow$ **ComputeIntention**$(P_i, C_i, R_i, H_{i,j}, S_{i,j})$;
8     **if** $W_i^t \geq \theta$ **then**
9       $a_i^t \leftarrow r_i^t.\text{type}$;
10      $r_i^t.\text{status} \leftarrow$ accepted;
11     **else**
12      $a_i^t \leftarrow$ Self-Initiated Learning;
13      **RejectSocialRequest**$(i, \text{'Low Intention'})$;
14   **else**
15     $a_i^t \leftarrow$ Self-Initiated Learning;
16   **return** $a_i^t$;

---

Interaction priorities are structured into three tiers. First, **teacher requests** (e.g., direct questioning or task assignments) override all other interactions, placing the student agent in an uninterruptible execution state. Second, if no teacher request is present, the agent evaluates **peer requests** and **group discussions** (e.g., side chats, peer questions). Here, an LLM-based reasoning process integrates personality traits, task context, and interaction history to compute a *social willingness score* $W_i^t \in [0, 1]$. If $W_i^t \geq \theta$ (e.g., $\theta = 0.6$), the request is accepted; otherwise, it is rejected and the agent returns to self-regulated learning. Third, in the absence of external input, the agent continues self-initiated planning and behavior execution.

To ensure transparent tracking, all requests are logged in the shared interaction state pool with one of three tags: **Pending** (awaiting response), **Accepted** (engagement initiated; dialogue content logged for future reasoning), or **Rejected** (declined without side effects; agent resumes autonomous learning).

This tagging protocol enables fine-grained regulation of peer dialogue without disrupting the core instructional flow.

Overall, the gating-based prioritization mechanism preserves instructional coherence while still allowing socially grounded behaviors to emerge adaptively and contextually in classroom simulations.

### C.3.3   SOCIAL INTENTION FUNCTION

In multi-agent classroom environments, modeling whether a student agent is willing to accept peer-initiated interactions is crucial for simulating realistic social behavior. To this end, **CIE** introduces a language model–driven **Social Intention Function**, which dynamically determines willingness based on the current instructional context. This function integrates five factors—personality traits, learning confidence, task relevance, historical interaction frequency, and social closeness—into a context-aware decision process.

To ensure educational interpretability, each factor is anchored in established psychological theories:

- **Personality Match** ($P$): Following the *Big Five Personality Theory*, students high in extraversion tend to be socially responsive, while those high in neuroticism are more likely to avoid interaction (Zhao & Seibert, 2006; John et al., 1999).

- **Current Learning Confidence** ($C$): Based on *Bandura's Self-Efficacy Theory*, confidence in task performance directly shapes one's propensity for social engagement (Schunk & DiBenedetto, 2016; Bandura & Adams, 1977).

- **Task Relevance** ($R$): Grounded in *Situated Cognition* and *Constructivist Learning Theory*, this factor assesses whether a social request aligns with the ongoing instructional objective (Wilson & Myers, 2000; Hedegaard, 1998).

- **Historical Interaction Frequency** ($H$) and **Social Closeness** ($S$): Informed by *Social Identity Theory*, these capture group belonging and accumulated positive peer interactions (Hogg, 2016; Ellemers & Haslam, 2012).

Each component is scored within the range $[0, 100]$ using GPT-4 via fine-tuned prompts, represented as $P, C, R, H, S$. Simultaneously, the system generates a context-dependent weight vector:

$$\boldsymbol{\alpha} = [\alpha_1, \alpha_2, \alpha_3, \alpha_4, \alpha_5], \quad \text{subject to} \quad \sum_{i=1}^{5} \alpha_i = 1. \tag{4}$$

The model also outputs a short justification for each weight to enhance interpretability. The final **Social Intention Score** is computed as:

$$W = \alpha_1 P + \alpha_2 C + \alpha_3 R + \alpha_4 H + \alpha_5 S. \tag{5}$$

If $W \geq 0.6$ (default threshold), the student accepts the request and enters a dialogue; otherwise, the request is rejected and the agent resumes self-directed learning.

This mechanism functions not merely as a scoring model, but as a cognitively and socially grounded reflection of student behavior. It captures a learner's "social rationality" and "regulatory capacity" across tasks and roles, enhancing both the behavioral realism and interpretability of CIE's agent-based interaction model.

### C.4   DETAILED DESCRIPTION OF TEMPORAL DYNAMICS

### C.4.1   THEORETICAL FOUNDATIONS OF AGENT STATE PROGRESSION AND REGULATION

In **CIE**, the learning process of virtual students is conceptualized not as isolated one-step reactions but as a temporally extended trajectory unfolding across multiple time steps. This trajectory reflects progressive **cognitive development**, **emotional regulation**, and **behavioral adaptation**. To capture these dynamics, we introduce a dual mechanism of **State Progression** and **State Regulation**, grounded in established theories of educational psychology, thereby modeling agent-level development across lessons and instructional phases.

**Cognitive development** is anchored in Bloom's Taxonomy (Krathwohl, 2002; Forehand, 2010) and Bruner's Spiral Curriculum (Clark, 2010; Joseph, 2021). Bloom's hierarchy delineates a progression from lower- to higher-order cognition:

Remembering $\rightarrow$ Understanding $\rightarrow$ Applying $\rightarrow$ Analyzing $\rightarrow$ Evaluating $\rightarrow$ Creating.

In CIE, each student agent's cognitive output is annotated accordingly at each step, enabling temporal trend analysis. Complementarily, Bruner's spiral principle emphasizes cyclical revisiting of concepts with increasing complexity, allowing longitudinal tracking of knowledge deepening, reinforcement, and occasional regression.

**Emotional and behavioral adaptation** is informed by Emotion Regulation Theory (Gross, 2008; 1999), Self-Determination Theory (SDT) (Deci et al., 2017; Deci & Ryan, 2012), and Expectancy–Value Theory (Wigfield, 1994; Wigfield & Eccles, 2000). Gross highlights that learners regulate emotions through strategies such as support-seeking, withdrawal, or task switching. In CIE, these processes shape the `Regulate` module, where agents evaluate affective states to guide strategic adjustment. SDT introduces autonomy, competence, and relatedness as motivational parameters; these are embedded into CIE's motivation and social intention functions, influencing persistence and behavioral shifts across time. Expectancy–Value perspectives further explain how perceived value and anticipated success jointly determine sustained engagement.

**Teacher–student feedback dynamics** build upon Hattie and Timperley's Feedback Model (Hattie & Timperley, 2007b) and Vygotsky's Sociocultural Theory (Lantolf, 2000; Scott & Palincsar, 2013). The former stresses that effective feedback triggers metacognitive reassessment and regulation beyond error correction; in CIE, such feedback modifies both goal-setting and emotional states in subsequent `Plan` phases. Vygotsky's theory adds a social dimension: knowledge construction is mediated by interaction, and teacher or peer feedback indirectly shapes confidence, regulation, and discourse strategies.

In summary, the integration of state progression and regulation mechanisms allows CIE to simulate learning as a temporally grounded, theory-consistent developmental process. Virtual students are thus modeled not as reactive output devices but as evolving educational agents whose longitudinal behavioral trajectories provide interpretable evidence of learning dynamics and cognitive pathway development.

### C.4.2 TEACHER-CONTROLLED INSTRUCTIONAL PACING

---

**Algorithm A.2:** Teacher-controlled Instructional Phase Pacing

**Input:** Current instructional phase $\text{Phase}_k$, student state set $\{s_i^t\}_{i=1}^n$, step count $t$
**Output:** Next instructional phase $\text{Phase}_{k+1}$

1 Initialize total steps per phase: $T \leftarrow 30$ ;
2 Initialize current step index: $t \leftarrow 1$ ;
3 Initialize teacher policy: $\pi_T$ ;
4 **while** $t \leq T$ **do**
5      Observe group student state $S^t = \{s_i^t\}_{i=1}^n$ ;
6      Execute teacher action $a_T^t \sim \pi_T(S^t, \text{Phase}_k)$ ;
7      Broadcast action $a_T^t$ to all student agents ;
8      Collect responses and update $S^{t+1}$ ;
9      $t \leftarrow t+1$ ;
10 Evaluate completion rate: $r_k \leftarrow \text{Evaluate}(S^t)$ ;
11 Determine transition: $\text{Phase}_{k+1} \leftarrow \text{Transition}(\text{Phase}_k, S^t, r_k)$ ;
12 **return** $\text{Phase}_{k+1}$

---

In multi-step instructional simulations, effective pacing control is essential not only for synchronizing with students' learning rhythms but also for maintaining interactional coherence and managing cognitive load. To address this challenge, the **CIE** framework incorporates a **Teacher Agent Rhythm Control Module**, which dynamically governs the progression of instructional phases throughout a lesson.

Each lesson is preconfigured into five canonical instructional phases (e.g., introduction, explanation, consolidation), further decomposed into discrete teaching steps. During simulation, the teacher

agent adaptively decides whether to continue, delay, or advance phase transitions based on real-time classroom signals rather than following a fixed timeline.

This decision process operates through a **Perception–Cognition–Action (PCA)** loop. At each step $t$, the teacher agent first perceives aggregated student states $S^t = \{s_i^t\}_{i=1}^n$ (e.g., participation density, emotional distribution, cognitive indicators). In the cognition stage, it evaluates whether the current phase $\text{Phase}_k$ should be sustained or transitioned by integrating lesson plan constraints, system logs, and recent feedback. Finally, the action stage executes the pacing decision, broadcasting teacher actions $a_T^t$ and updating the global state.

The pacing controller thus implements a data-driven mechanism that ensures phase boundaries remain pedagogically aligned and interpretable across the session. Each time step is explicitly logged with its transition rationale, enabling post-hoc analysis and iterative refinement of instructional design. The full scheduling logic is summarized in Alg. A.2.

## C.5 Memory Mechanisms for Agent Cognition

### C.5.1 Theoretical Foundations of Memory Mechanisms

In cognitive psychology and the learning sciences, memory mechanisms are central to understanding how learners encode, retain, and retrieve knowledge for decision-making (Terry, 2017; Sprenger, 1999). To simulate this process in virtual agents, CIE implements a dual-layer memory architecture inspired by the classical *Working Memory–Long-Term Memory* model. This design ensures both real-time responsiveness and cross-session continuity in agent cognition and behavior.

**Working memory** provides short-term storage of salient instructional information during ongoing sessions—such as teacher actions, peer interactions, emotional states, and cognitive indicators. These records are stored in a global `shared_state` structure, which is updated at each time step $t$. This memory layer enables the real-time execution of the *Perception–Cognition–Action (PCA)* loop, featuring high temporal resolution and frequent access, thereby forming the basis for moment-to-moment decision-making.

**Long-term memory**, in parallel, functions as a persistent repository of knowledge accumulation and behavioral trajectories. Implemented as a structured database, it logs each student's historical records across lessons, including cognitive progression, emotional trends, and task engagement. At the beginning of each new session, the long-term memory is reloaded into the `shared_state`, enabling agents to adapt based on prior experiences. This mechanism supports *experience-informed learning* and retrospective reasoning across multiple episodes.

By integrating these two layers, CIE models both the ephemeral and cumulative aspects of learning. The separation between rapidly evolving working states and persistent knowledge encoding ensures that agents can respond fluidly to the immediate instructional context while also continuously adapting to their developmental history. The following sections provide a detailed description of the design and flow between these two memory layers.

### C.5.2 Implementation of Short-term Shared Memory

In CIE, the short-term memory mechanism—termed **short-term shared memory**—defines a unified interaction state pool, `shared_state`, which facilitates high-frequency, real-time information exchange among all agents, including both students and teachers. Drawing inspiration from the psychological construct of working memory, this module temporarily stores task-relevant perceptual information and cognitive-affective states within the current instructional phase, thereby enabling synchronized decision-making across agents.

The `shared_state` consists of several structured components, initialized at the start of each session:

- **Teacher state**: records the teacher's current behavior, utterances, and emotional tone.

- **Request pools (instructional & social)**: manage the lifecycle of agent-to-agent interaction requests, including initiation, acceptance, and rejection.

- **Student logs**: capture each agent's cognitive level, emotional state, and behavior trace at every time step $t$.

- **Spatial state**: represents agent-specific surroundings and perceptible objects, grounding interactions in physical context.

- **Task & group context**: contains the active lesson plan, group composition, and instructional content segments.

- **Interaction history**: maintains continuity of dialogue rounds and the evolution of collaborative strategies.

- **Long-term memory reference**: integrates episodic summaries from previous lessons to initialize experience-informed behaviors.

This state pool is updated dynamically at each instructional step. For example, when a student initiates a peer request, the entry is appended to the social pool; when the teacher provides feedback, the teacher state is refreshed; when students respond, their logs are updated with behavioral and cognitive annotations. This continuous update cycle ensures that all agents maintain a synchronized representation of the evolving classroom environment.

In summary, the short-term shared memory functions as the temporal backbone of coordinated multi-agent interaction in CIE. By supporting real-time perception and regulation within the Perception–Cognition–Action loop, it enables adaptive and coherent decision-making in high-frequency educational scenarios.

### C.5.3 IMPLEMENTATION OF LONG-TERM MEMORY

To sustain behavioral continuity and cumulative cognitive development across sessions, CIE incorporates a structured **long-term memory system** consisting of two SQLite databases: `student_memory.db` and `teacher_memory.db`. These databases respectively record student learning trajectories and teacher instructional behaviors over time.

**Student Memory**: The `student_memory.db` contains a `long_term_memory` table with the following schema:

- `student_id`: unique identifier for each student agent.

- `event_type`: type of record (e.g., *Cognitive Planning*, *Monitoring*, *Regulation*, *Behavioral Record*).

- `event_content`: natural language logs produced during metacognitive stages, including goal setting or strategic reflection.

- `timestamp`: temporal marker that enables reconstruction of student-specific learning sequences.

This structure ensures traceability of self-regulated learning activities, forming a temporally grounded chain along the *Perception–Cognition–Action* loop.

**Teacher Memory**: The `teacher_memory.db` mirrors this design, maintaining a `long_term_memory` table that logs instructional records at the phase level:

- `teacher_id`: unique identifier for the teacher agent.

- `lesson_id, phase`: identifiers of the lesson and instructional phase.

- `event_type`: type of teacher behavior (e.g., *Instructional Planning*, *Feedback*).

- `content`: natural language descriptions of teacher intentions, evaluations, and scaffolding strategies.

- `timestamp`: time of execution, supporting longitudinal modeling of instructional dynamics.

A particular focus is placed on feedback chains (e.g., student question $\rightarrow$ teacher evaluation $\rightarrow$ student regulation), which provide the basis for context-aware instructional planning in future sessions and adaptive modeling of scaffolding behaviors.

Together, these two databases enable bidirectional memory transfer: teacher agents analyze longitudinal patterns in student performance, while student agents draw upon accumulated knowledge, goals, and feedback to guide future behavior. The long-term memory system thus allows CIE to simulate instructional interaction as an evolving and authentic temporal process.

### C.5.4 MEMORY INTERACTION FLOW

To support cognitive development and behavioral adaptation across instructional sessions, CIE implements a structured **Memory Interaction Flow** that coordinates short-term and long-term memory. This mechanism enables student agents to accumulate, apply, and evolve learning experiences within and across sessions, thereby sustaining coherence and continuity in personalized learning trajectories.

At the start of each session, the system loads individual long-term memory summaries—such as prior performance, cognitive tendencies, and emotional traits—from the database into the `shared_state`. These values immediately inform in-session planning and response generation.

During instruction, student agents continuously update the short-term memory at each time step by recording their actions, peer interactions, and teacher feedback. These records ensure real-time context awareness and guide micro-level cognitive regulation.

At the end of a session, the system aggregates time-step records into structured learning summaries. These include updated cognitive markers, emotional trajectories, and selected behavior patterns, which are written back into the long-term memory for use in subsequent sessions.

This continuous memory flow enables student agents to engage in feedback-informed, data-driven self-regulation. Over time, they develop individualized learning patterns reflective of authentic developmental trajectories in classroom environments. The complete execution logic of this flow is outlined in Alg. A.3.

---

**Algorithm A.3: Memory Interaction Flow for Agent $i$**

**Input:** Agent ID $i$, shared state $S$, current step $t$, total steps $T$
**Output:** Updated shared state $S$

1 **if** $t = 1$ **then**
2    $L_i \leftarrow$ DATABASE.retrieve_long_term_summary($i$);
3    $S[\text{longterm\_summary}][i] \leftarrow L_i$;
4 $S[\text{interaction\_log}][i][t] \leftarrow \{$
5    teacher_interactions : $s_i^t$.teacher_content,
6    social_interactions : $s_i^t$.social_requests,
7    student_responses : $s_i^t$.stu_response,
8    student_requests : $s_i^t$.stu_request,
9    environment_context : $s_i^t$.visible_items,
10    emotional_states : $s_i^t$.teacher_emotion,
11    cognitive_states : MEMORY.retrieve_cog()
12 $\}$
13 **if** $t = T$ **then**
14    $M_i \leftarrow$ Summarize_Session($S[\text{interaction\_log}][i]$);
15    DATABASE.store_long_term_summary($i, M_i$);
16    $S[\text{longterm\_summary}][i]$.pre_lesson_summary $\leftarrow M_i$.summary;
17    $S[\text{longterm\_summary}][i]$.pre_lesson_portrait $\leftarrow M_i$.portrait;
18    $S[\text{longterm\_summary}][i]$.pre_lesson_regulation $\leftarrow M_i$.regulation;
19    $S[\text{longterm\_summary}][i]$.teacher_evaluation $\leftarrow M_i$.teacher_eval;
20 **return** $S$;

---

## D DETAILED INFORMATION FOR EXPERIMENTAL DESIGN AND EVALUATION

### D.1 EXPERIMENT SETUP AND CONFIGURATION

To investigate how virtual students respond cognitively, emotionally, and socially across diverse instructional contexts, we developed a Chinese language classroom simulation integrating genre diversity, phased pedagogy, and structured interaction protocols.

For materials, three representative texts from the junior secondary curriculum were selected: the lyrical prose *Spring* (Zhu Ziqing), the foreign fable *The Emperor's New Clothes*, and the argumentative essay *Dedication and Joy*. These texts differ in linguistic style, cognitive demand, and emotional resonance, supporting heterogeneous tasks such as expressive description, character analysis with moral reasoning, and logical argumentation for critical debate.

Interaction was structured through an extended IRF model. By adding a regulation phase, we formed an "I–R–F–Regulate" loop: the teacher initiates, students respond, the teacher provides feedback, and students regulate through reflection or social actions (e.g., questioning, discussion, or strategy adjustment). This design preserves traditional instructional dialogue while enhancing agents' behavioral and emotional expressiveness.

Each lesson comprised five pedagogical phases—introduction, instruction, consolidation, practice, and summarization—mapped to approximately 30 steps but dynamically adjusted by the teacher's policy to about 36 steps per session, depending on task completion and engagement signals.

The agent architecture followed a two-tier design: GPT-4 was responsible for natural language generation and reasoning, while a fine-tuned InternVL model modulated style. Each student was encoded as $[p_i; c_i; m_i]$—personality, cognitive style, and motivation—and combined with phase, memory, and context in prompt templates. Prompts were configured with temperature $= 0.5$, max tokens $= 512$, top-$p = 0.9$, and frequency penalty $= 0.2$.

Experiments were run on a server with H20-NVLink GPUs (96GB VRAM) and 200GB RAM. Each inference step averaged 25 seconds, and a full class of six students plus one teacher lasted 1–2 hours. All session data—including cognitive, emotional, and behavioral annotations—were stored in a MongoDB backend for longitudinal continuity analysis.

To simulate learner diversity, six virtual student archetypes were designed based on the Big Five personality model, motivational theory, and cognitive style literature. These archetypes reflect typical learner profiles in real classrooms and enable the evaluation of interactional variance and pedagogical robustness.

---

**Wang Fang**

**Age:** 13
**Gender:** Female
**Personality:** High Neuroticism
**Class Role:** Academic Student
**Learning Motivation:** High
**Class Behavior:** Often hesitant but gradually contributes well-structured summaries and personal reflections during class.
**Cognitive Style:** Field-dependent
**Thinking Tendency:** Reflective

---

**Zhang Jie**

**Age:** 14
**Gender:** Male
**Personality:** High Extraversion
**Class Role:** Academic Student
**Learning Motivation:** High
**Class Behavior:** Actively initiates discussions, shares opinions confidently, and frequently stands up to respond or ask questions.

---

**Cognitive Style:** Field-independent
**Thinking Tendency:** Reflective

### Zhang Yan

**Age:** 13
**Gender:** Female
**Personality:** High Agreeableness
**Class Role:** Academic Student
**Learning Motivation:** High
**Class Behavior:** Frequently engages in peer interaction, supports others' ideas, and shows strong cooperative communication.
**Cognitive Style:** Field-independent
**Thinking Tendency:** Reflective

### Li Wei

**Age:** 14
**Gender:** Male
**Personality:** Low Openness
**Class Role:** Discussion Student
**Learning Motivation:** High
**Class Behavior:** Leads group discussion with structured logic, seeks consensus, and promotes balanced participation.
**Cognitive Style:** Field-dependent
**Thinking Tendency:** impulsive

### Liu Li

**Age:** 13
**Gender:** Female
**Personality:** Low Openness
**Class Role:** Off-task Student
**Learning Motivation:** Low
**Class Behavior:** Easily distracted in class, often avoids eye contact, but occasionally responds with emotional expressions.
**Cognitive Style:** Field-dependent
**Thinking Tendency:** Impulsive

### Zhang Tao

**Age:** 14
**Gender:** Male
**Personality:** Low Conscientiousness
**Class Role:** Off-task Student
**Learning Motivation:** Low
**Class Behavior:** Tends to disengage from class tasks, shows low participation, and often chats about irrelevant topics.
**Cognitive Style:** Field-dependent
**Thinking Tendency:** Impulsive

## D.2 BEC GENERATED FRAMEWORK

### D.2.1 THEORETICAL FOUNDATIONS

To annotate learner states in a structured and interpretable manner, we adopt a three-dimensional **Behavior–Emotion–Cognition (BEC)** framework grounded in well-established educational theories.

- **Behavior** builds on and extends the ICAP theory of cognitive engagement, refining observable classroom actions into granular categories that differentiate active engagement, passive engagement, off-task behavior, and peer interaction.
- **Emotion** follows widely used classroom affect taxonomies in educational psychology, focusing on three core emotional states: positive, negative, and confused.
- **Cognition** adheres to Bloom's revised taxonomy, encompassing six levels of cognitive processing: remembering, understanding, applying, analyzing, evaluating, and creating.

This framework enables consistent annotation of multimodal learner behaviors and supports downstream quantitative and qualitative analysis of instructional interactions.

### D.2.2 BEC GENERATED PROCEDURE

In EduVerse, BEC labels are generated through structured prompts rather than used as predictive accuracy targets. Their purpose is to model each virtual student's *subjective self-perception* during the learning process, forming part of the agent's metacognitive cycle (Plan–Monitor–Regulate). In the Monitor stage, the model is prompted to externalize its current behavioral, emotional, and cognitive state; thus, BEC serves as a prompt-guided self-report mechanism rather than an external evaluation label. This design follows practices in psychology and agent-based modeling where self-reporting is used to express internal states, and is consistent with systems such as Generative Agents (Park et al. (2023)), which rely on agent-generated reflections to support memory writing and long-term development.

### D.2.3 CATEGORY DEFINITIONS AND MAPPING

We employ a three-dimensional BEC framework to annotate learner states at each timestep. All labels are fully in English to ensure cross-platform compatibility and avoid rendering issues.

Table A5: BEC annotation categories used in EduVerse.

| Dimension | Categories |
|---|---|
| Behavior | Note Taking (NT), Hand Raise (HR), Head Up (HU), Head Down (HD), Read Aloud (RA), Refuse Reply (RR), Stand Answer (SA), Side Talk (ST), Answer Questions (AQ), Sleep, Chat |
| Emotion | Positive, Negative, Confused |
| Cognition | Remember, Understand, Apply, Analyze, Evaluate, Create |

### D.2.4 BEC PRIORITY SCHEME

For downstream aggregation, we map fine-grained BEC labels into ordered priority levels so that higher values indicate more engaged or higher–order states.

**Behavior priority.** We define a 4–level ordinal variable $p^B \in \{0, 1, 2, 3\}$:

- 0 (Off–task): off–task behaviors such as *Sleep* or *Chat*.
- 1 (Passive): passive on–task behaviors such as *Head Down* listening.
- 2 (Active): individual active behaviors such as *Note Taking*, *Hand Raise*, *Head Up*, *Read Aloud*, or *Stand Answer*.
- 3 (Interactive): socially interactive behaviors such as *Side Talk*, *Refuse Reply*, *Answer Questions*, or *Group Discussion*.

**Emotion priority.** We define a 3–level variable $p^E \in \{0, 1, 2\}$:

- 0 (Negative): negative affect.
- 1 (Confused): confused or uncertain affect.
- 2 (Positive): positive or engaged affect.

**Cognition priority.** We define a 2–level variable $p^C \in \{0, 1\}$ following Bloom's taxonomy:

- 0 (Lower–order): *Remember* and *Understand*.
- 2 (Higher–order): *Apply ,Analyze ,Evaluate* and *Create*.

Table A6 summarizes the mapping used when converting prompt–generated BEC labels into ordinal scores.

Table A6: Priority levels for behavior, emotion, and cognition used in BEC aggregation.

| Dimension | Level | Description |
|---|---|---|
| Behavior | 3 | Interactive (peer/teacher interaction) |
|  | 2 | Active (individual active learning) |
|  | 1 | Passive (on–task but low engagement) |
|  | 0 | Off–task (disengaged behavior) |
| Emotion | 2 | Positive |
|  | 1 | Confused |
|  | 0 | Negative |
| Cognition | 1 | Higher–order (Apply, Analyze, Evaluate, Create) |
|  | 0 | Lower–order (Remember, Understand) |

The BEC framework provides a unified labeling standard across behavior, emotion, and cognition, enabling fine-grained analysis of learner trajectories and supporting the interpretability and reproducibility of EduVerse's simulation results.

## D.3 EXPERIMENT I

### D.3.1 DETAILED EXPERIMENT RESULTS

Table A7: **IRF distribution for three text genres across four environments.** Values are relative frequencies of Initiation (I), Response (R), and Feedback (F). **IRF$_{rate}$** denotes the overall completion ratio.

| Text Genre | Env. | Steps | I | R | F | IRF$_{rate}$ |
|---|---|---|---|---|---|---|
| *Lyrical Prose* | Lecture | 37 | 0.514 | 0.167 | 0.275 | 0.432 |
| | Collab | 41 | 0.439 | 0.179 | 0.321 | 0.293 |
| | Round | 39 | 0.410 | 0.154 | 0.282 | 0.282 |
| | Real Class | 37 | 0.513 | – | 0.703 | 0.486 |
| *Argumentative Essay* | Lecture | 36 | 0.556 | 0.194 | 0.282 | 0.639 |
| | Collab | 40 | 0.375 | 0.213 | 0.367 | 0.475 |
| | Round | 31 | 0.516 | 0.215 | 0.355 | 0.548 |
| | Real Class | 36 | 0.417 | – | 0.583 | 0.417 |
| *Foreign Fiction* | Lecture | 33 | 0.364 | 0.253 | 0.475 | 0.455 |
| | Collab | 33 | 0.242 | 0.247 | 0.394 | 0.303 |
| | Round | 37 | 0.324 | 0.189 | 0.351 | 0.378 |
| | Real Class | 33 | 0.367 | – | 0.515 | 0.367 |

IRF = Initiation, Response, and Feedback ratio in dialogue.
Real Class = Real classroom environment.

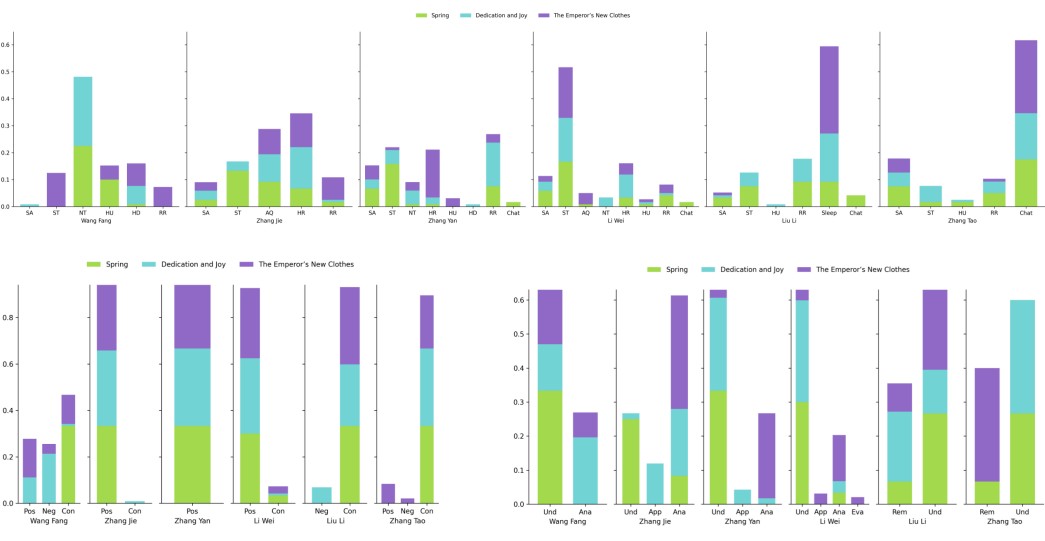

Figure A1: Student behavior, emotion and cognition frequencies across three lessons in collab environment.

To complement the main findings of Experiment I, this section provides additional analyses using IRF statistics, behavior–cognition–emotion (BCE) distributions, individual-level visualizations, and ablation studies.

First, the full **IRF statistics across genres and environments** (Tab. A7) show that simulated classrooms maintain comparable IRF completion rates with real classes. For instance, in lyrical prose, Lecture/Collab/Round yield IRF$_{rate}$s between 0.28 and 0.43, versus 0.486 in the real classroom. Argumentative essays and foreign fiction also exhibit distributions of similar magnitude. These findings substantiate the main-text claim that simulated discourse structures align closely with real classrooms, while minor deviations (e.g., higher rates in Lecture for argumentative essays) reflect

Table A8: Representative IRF dialogue excerpts across genres in simulated and real classroom.

*Note:* "Setting" refers to classroom type (**S**:Simulation or **RC**: Real Class) and Genres (**LP**: Lyrical Prose, **AE**: Argumentative Essay, **FF**: Foreign Fiction.)

| Setting | I (Initiation) | R (Response) | F (Feedback) |
|---|---|---|---|
| **S_LP** | Li Wei, could you share your understanding of this sentence? The rest of you may also think about which images or characters in the text left a strong impression on you. | I think Zhu Ziqing's description is very vivid. He humanized spring, making readers feel its warmth and vitality. The contrast between "little grass" and "new leaves" makes the scene full of freshness and life. | Excellent sharing and observation! You captured the emotional tone of the classroom very well. |
| **S_AE** | Zhang Yan, could you try to create a phrase using *Take out of context*? This might help us better understand how to apply this expression. | Okay, for example, we can't just take one sentence and explain it in isolation. We need to connect it to the context. | Great example and reasoning! Keep up your analytical depth. |
| **S_FF** | This story reveals that many people chose to follow authority rather than uphold truth and justice. Based on "The Emperor's New Clothes," why do you think some people stay silent before power? | I think... maybe because they are afraid of losing status or being excluded if they tell the truth. People often go along with others to protect themselves. | Excellent reflection and teamwork! Maintain this thoughtful analysis and courage to question. |
| **RC_LP** | There's a sentence here without additional modifiers. Tiantian, what do you think makes it effective? | Hmm, it's very concise. The author used pure description to highlight the beauty of the scene. | Good, that's correct. |
| **RC_AE** | The author discusses several issues. Which one do you find most convincing? Please share your opinion. You please. | First, he emphasizes the meaning of dedication. I think this value is most inspiring. | Good point. Who would like to add to that? |
| **RC_FF** | Who can tell me what the first function of clothing is in this story? What does it symbolize? | It shows one's social status — for instance, how the emperor's clothes represent vanity and hypocrisy. | Exactly, well answered. |

Table A9: **Distribution of students' behavior, cognition, and emotion across different text genres and environments.**

| Env. | Text | B_Aac | B_Pas | B_Int | B_Off | E_Pos | E_Con | E_Neg | C_Low | C_High |
|---|---|---|---|---|---|---|---|---|---|---|
| Lecture | *Lyrical Prose* | 0.157 | 0.222 | 0.398 | 0.222 | 0.509 | 0.463 | 0.028 | 0.875 | 0.125 |
| | *Argumentative Essay* | 0.200 | 0.276 | 0.324 | 0.200 | 0.481 | 0.519 | 0.000 | 0.790 | 0.210 |
| | *Foreign Fiction* | 0.161 | 0.260 | 0.396 | 0.182 | 0.510 | 0.484 | 0.005 | 0.792 | 0.208 |
| Collab | *Lyrical Prose* | 0.171 | 0.204 | 0.454 | 0.171 | 0.483 | 0.517 | 0.000 | 0.942 | 0.058 |
| | *Argumentative Essay* | 0.303 | 0.205 | 0.316 | 0.175 | 0.547 | 0.312 | 0.141 | 0.697 | 0.303 |
| | *Foreign Fiction* | 0.188 | 0.203 | 0.313 | 0.297 | 0.609 | 0.359 | 0.031 | 0.578 | 0.422 |
| Round | *Lyrical Prose* | 0.154 | 0.154 | 0.461 | 0.232 | 0.482 | 0.518 | 0.000 | 0.807 | 0.193 |
| | *Argumentative Essay* | 0.189 | 0.200 | 0.383 | 0.228 | 0.511 | 0.406 | 0.083 | 0.794 | 0.206 |
| | *Foreign Fiction* | 0.204 | 0.269 | 0.227 | 0.301 | 0.616 | 0.296 | 0.088 | 0.861 | 0.139 |

the annotation protocol. To complement the quantitative results, we further provide a qualitative analysis of IRF structures across genres in both real and simulated classrooms (Tab. A8). Following

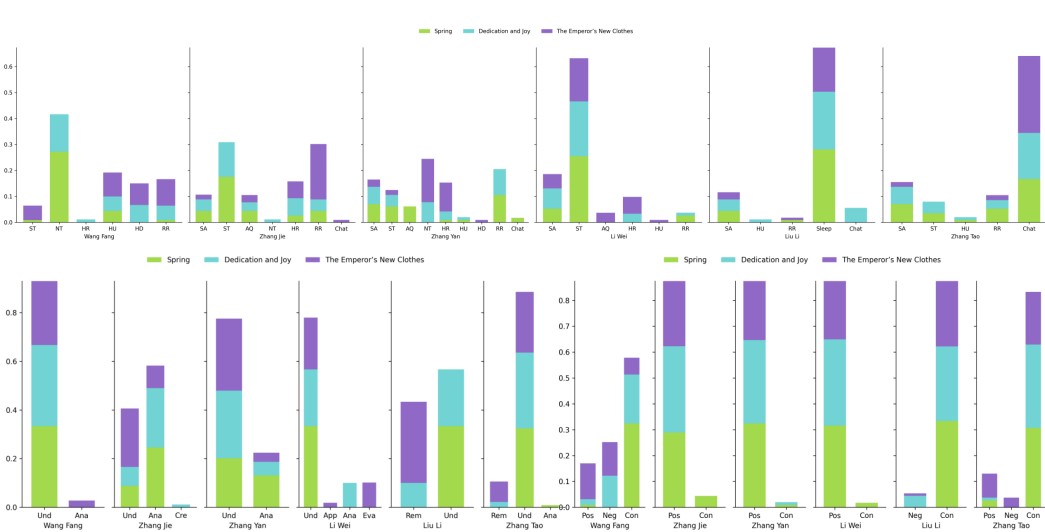

Figure A2: Student behavior, emotion and cognition frequencies across three lessons in round environment.

Table A10: **Ablation experiments across different frameworks.** Results are reported on three text genres. "nolocal" denotes removing local interaction rules, while "nocog" denotes removing cognitive mechanisms.

| Text Genre | Framework | B_Aac | B_Pas | B_Int | B_Off | E_Pos | E_Con | E_Neg | C_Low | C_High |
|---|---|---|---|---|---|---|---|---|---|---|
| *Lyrical Prose* | Ours | 0.157 | 0.222 | 0.398 | 0.222 | 0.509 | 0.463 | 0.028 | 0.875 | 0.125 |
| | nolocal | 0.321 | 0.195 | 0.224 | 0.260 | 0.557 | 0.301 | 0.142 | 0.890 | 0.110 |
| | nocog | 0.315 | 0.212 | 0.216 | 0.257 | 0.505 | 0.351 | 0.144 | 0.631 | 0.369 |
| *Argumentative Essay* | Ours | 0.200 | 0.276 | 0.324 | 0.200 | 0.481 | 0.519 | 0.000 | 0.790 | 0.210 |
| | nolocal | 0.221 | 0.225 | 0.329 | 0.225 | 0.523 | 0.302 | 0.176 | 0.883 | 0.117 |
| | nocog | 0.229 | 0.233 | 0.233 | 0.304 | 0.563 | 0.429 | 0.008 | 0.479 | 0.521 |
| *Foreign Fiction* | Ours | 0.161 | 0.260 | 0.396 | 0.182 | 0.510 | 0.484 | 0.005 | 0.792 | 0.208 |
| | nolocal | 0.210 | 0.271 | 0.295 | 0.224 | 0.524 | 0.390 | 0.086 | 0.748 | 0.252 |
| | nocog | 0.266 | 0.306 | 0.158 | 0.270 | 0.509 | 0.468 | 0.023 | 0.176 | 0.824 |

the classical definition of the IRF framework, we compare the linguistic patterns of Initiation (I), Response (R), and Feedback (F). At the I and R stages, simulated exchanges closely mirror real classroom discourse: question framing, elicitation styles, and student response types exhibit highly consistent language forms, indicating that EduVerse captures the core logic of teacher-led interaction. Differences appear primarily at the F stage. Real teachers often employ open-ended or affective feedback cues (e.g., "Who would like to add to that?"), while simulated teachers tend to adopt more concise, evaluative feedback (e.g., "Great example and reasoning!"). Such variation aligns with natural differences across teacher styles and instructional strategies rather than model deficiencies.

Second, the **BCE distribution** (Tab. A9) further illustrates classroom ecology. Overall, lower-order cognition dominates (e.g., Lecture–LP $C\_Low = 0.875$), positive and confused emotions prevail, and negative affect remains low. Environment effects vary by genre: in foreign fiction, Collab produces higher-order cognition ($C\_High = 0.422$), while Round leads to more off-task behavior ($B\_Off = 0.301$) and less interaction ($B\_Int = 0.227$). In contrast, Lecture settings in lyrical prose and argumentative essays show more passive participation and lower-order cognition. These results highlight a genre–environment interaction that systematically shapes classroom dynamics.

Third, the **individual-level visualizations** (Fig. A1 and Fig. A2) confirm personality-driven stability. Highly extraverted or conscientious students sustain active engagement and positive affect, whereas

low-openness or low-conscientiousness students display more off-task behavior, low-level cognition, and frequent confusion. This consistency across lessons aligns with the main-text analysis.

Finally, the **ablation experiments** (Tab. A10) disentangle the role of key modules. Removing localized interaction rules (`nolocal`) increases negative emotions and undermines affective realism. Removing cognitive mechanisms (`nocog`) exaggerates higher-order reasoning and overly active behaviors, deviating from gradual, student-like learning. Together, these results confirm that style modulation preserves emotional plausibility, while the cognitive layer ensures educational consistency.

In sum, this section corroborates three central claims of Experiment I: the **authenticity of discourse structures**, the **stability of personality-driven patterns**, and the **necessity of modular design**, while also emphasizing that genre–environment interactions play a critical role in shaping simulated classroom dynamics.

### D.3.2   REAL-CLASSROOM DATA SOURCE AND IRF ANNOTATION PROTOCOL

**Real-Classroom Source and Selection Criteria.** The real classroom data used in this study were obtained from the National Smart Education Platform (https://basic.smartedu.cn/ ), a national-level open platform administered by the Ministry of Education of China. The platform provides free access to a wide range of high-quality, government-reviewed instructional videos, ensuring strong representativeness and consistency. For this study, we selected three Chinese language lessons and one history lesson that strictly correspond to the three genres examined in our simulations: Lyrical Prose, Argumentative Essay, Foreign Fiction and world-history. All selected lessons were taught by experienced middle-school Chinese teachers and followed the national curriculum standards. These lessons exhibit stable instructional organization, clear audio–video quality, consistent discourse patterns, and well-structured IRF sequences. As such, they serve as reliable baseline classrooms for comparison against simulated lessons in EduVerse.

**IRF Annotation Mechanism and Quality Control.** To ensure annotation accuracy and reliability, we adopted a two-stage human-calibrated, AI-assisted annotation protocol. The process consists of the following steps:

**(1) Establishing the Annotation Benchmark.** A researcher with expertise in educational discourse analysis conducted multiple rounds of IRF labeling based on canonical literature examples until full intra-annotator consistency (100%) was achieved. A 15-minute segment from each real classroom was then manually transcribed and annotated sentence-by-sentence to create a gold-standard reference set. Time-aligned transcription was generated using automated tools to maintain consistent segmentation across annotation stages.

**(2) Calibrating the AI-Assisted Annotation Paradigm.** We employed KIMI, a large Chinese language model developed by Moonshot AI, to assist with IRF label generation. The model outputs were compared to the gold-standard annotations, and discrepancies were corrected through iterative human calibration. During this process, we established a unified decision protocol covering boundary criteria for Initiation (I) and Feedback (F); classification principles for probe-F and multi-turn follow-up questions; segmentation rules for long or multi-clause student responses; handling of group or choral responses; detection of implicit feedback embedded within teacher explanations. This protocol ensures that AI-assisted annotations are stable, rule-based, and reproducible, rather than dependent on model idiosyncrasies.

**(3) Batch Annotation and Human Verification.** With the annotation rules finalized, KIMI was used to generate IRF labels for the remaining $\bar{1}20$ minutes of classroom transcripts. All AI-generated labels were then manually reviewed and corrected by the researcher to ensure consistency across lessons and genres. This hybrid process balances annotation efficiency with high-quality control, yielding a reliable IRF dataset that supports subsequent quantitative and qualitative analyses.

## D.4 EXPERIMENT II

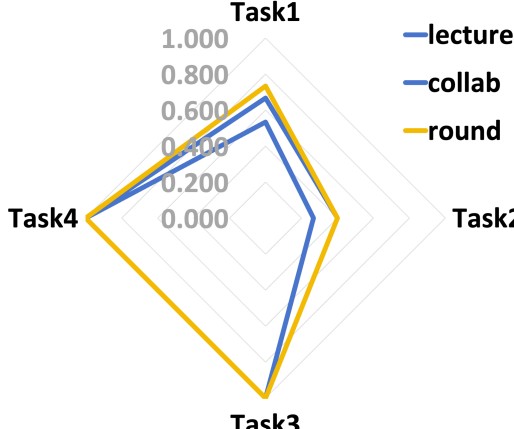

Figure A3: **Human–Agent Interaction Across Four Tasks.** Results show strong alignment with personality traits and robust instructional control, confirming that EduVerse enables seamless integration of human participants while preserving realistic classroom dynamics.

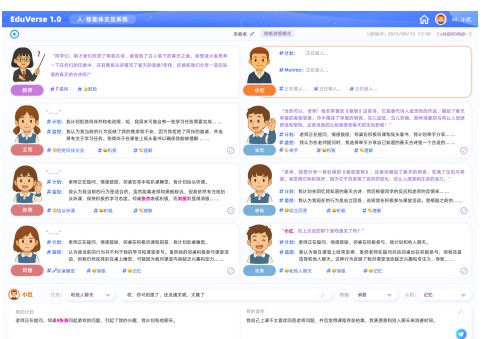

(a) Main Interface of EduVerse Human–Agent Interaction System)

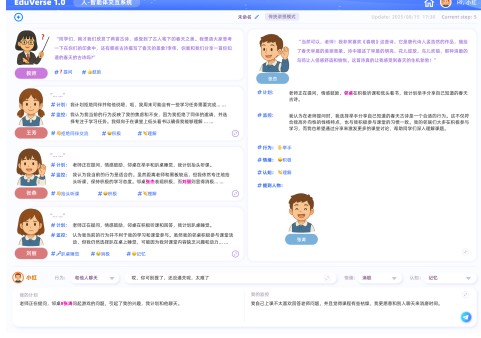

(b) Student Detail Interface of EduVerse Human–Agent Interaction System)

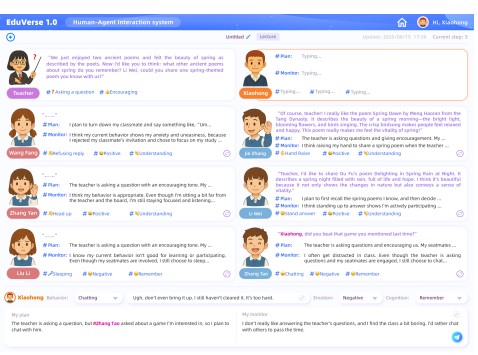

(c) Main Interface of EduVerse Human–Agent Interaction System (Translated Version).

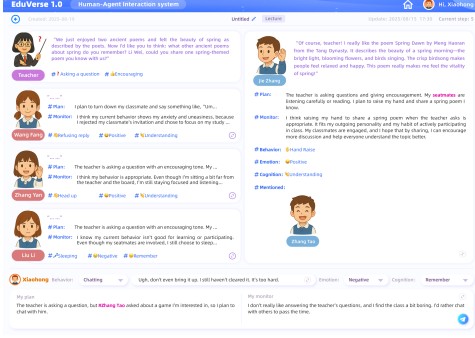

(d) Student Detail Interface of EduVerse Human–Agent Interaction System (Translated Version).

Figure A4: Interface of EduVerse human–agent interaction system.

In the supplementary analyses of Experiment II, we further validated the stability and realism of EduVerse through network centrality indicators and human–agent interaction tasks. As shown in Tab. A11, students displayed distinct role patterns across environments. For the *Foreign Fiction* les-

Table A11: **Distribution of students' network centrality indicators in classroom interaction contexts.** Values are normalized to $[0, 1]$; **Degree** = in+out centrality.

| Text | Env. | Student | In | Out | Degree | Betweenness |
|---|---|---|---|---|---|---|
| Foreign Fiction | Lecture | Wang Fang | 0.20 | 0.20 | 0.40 | 0 |
| | | Zhang Tao | 0.20 | 0.40 | 0.60 | 0.20 |
| | | Li Wei | 0.20 | 0.20 | 0.40 | 0 |
| | | Liu Li | 0.40 | 0.40 | 0.80 | 0.30 |
| | | Zhang Yan | 0.40 | 0.40 | 0.80 | 0.40 |
| | | Zhang Jie | 0.40 | 0.20 | 0.60 | 0.30 |
| | Collab | Wang Fang | 0.25 | 0.25 | 0.50 | 0 |
| | | Li Wei | 0.25 | 0.25 | 0.50 | 0 |
| | | Zhang Tao | 0.25 | 0.50 | 0.75 | 0.083 |
| | | Zhang Jie | 0.25 | 0.00 | 0.25 | 0 |
| | | Zhang Yan | 0.25 | 0.25 | 0.50 | 0 |
| | Round | Wang Fang | 0.40 | 0.20 | 0.60 | 0.15 |
| | | Li Wei | 0.20 | 0.20 | 0.40 | 0 |
| | | Zhang Tao | 0.40 | 0.40 | 0.80 | 0.25 |
| | | Zhang Jie | 0.20 | 0.20 | 0.40 | 0 |
| | | Zhang Yan | 0.00 | 0.40 | 0.40 | 0 |
| | | Liu Li | 0.20 | 0.00 | 0.20 | 0 |
| Lyrical Prose | Lecture | Li Wei | 0.20 | 0.20 | 0.40 | 0 |
| | | Zhang Jie | 0.40 | 0.40 | 0.80 | 0.15 |
| | | Liu Li | 0.20 | 0.20 | 0.40 | 0 |
| | | Zhang Yan | 0.20 | 0.20 | 0.40 | 0 |
| | | Zhang Tao | 0.20 | 0.40 | 0.60 | 0.10 |
| | | Wang Fang | 0.20 | 0.00 | 0.20 | 0 |
| | Collab | Li Wei | 0.25 | 0.25 | 0.50 | 0 |
| | | Liu Li | 0.25 | 0.25 | 0.50 | 0 |
| | | Zhang Tao | 0.25 | 0.25 | 0.50 | 0 |
| | | Zhang Yan | 0.50 | 0.50 | 1.00 | 0.167 |
| | | Zhang Jie | 0.25 | 0.25 | 0.50 | 0 |
| | Round | Wang Fang | 0.20 | 0.20 | 0.40 | 0.10 |
| | | Liu Li | 0.20 | 0.00 | 0.20 | 0 |
| | | Li Wei | 0.20 | 0.40 | 0.60 | 0.10 |
| | | Zhang Jie | 0.20 | 0.40 | 0.60 | 0.10 |
| | | Zhang Tao | 0.20 | 0.20 | 0.40 | 0 |
| | | Zhang Yan | 0.40 | 0.20 | 0.60 | 0.10 |

Table A12: **Human–agent interaction success rates across four tasks and three environments.** Values represent completion ratios (0–1).

| Task | Lecture | Collab | Round |
|---|---|---|---|
| Task 1: Peer chatting | 0.533 | 0.667 | 0.733 |
| Task 2: Peer academic response | 0.267 | 0.400 | 0.400 |
| Task 3: Teacher answering | 1.000 | 1.000 | 1.000 |
| Task 4: Teacher intervention | 1.000 | 1.000 | 1.000 |

son, in the Lecture setting Zhang Yan (Degree=0.80, Betweenness=0.40) and Liu Li (Degree=0.80, Betweenness=0.30) emerged as core participants, while Zhang Tao exhibited stronger initiative through higher outward connections (Out=0.40). In the Collab setting, interactions became more reciprocal, with most students maintaining a Degree of around 0.50, but Zhang Tao rose to 0.75 with a nonzero betweenness (0.083), serving as a connector. In the Round setting, the network is further decentralized, with bridging roles distributed: Zhang Tao (Degree=0.80, Betweenness=0.25) and Wang Fang (Betweenness=0.15) acted as key connectors, while Zhang Yan showed a distinctive one-directional output pattern (In=0.00, Out=0.40). For the *Lyrical Prose* lesson, Zhang Jie (Degree=0.80, Betweenness=0.15) was central in Lecture, Zhang Yan dominated in Collab (Degree=1.00, Betweenness=0.167), and Round produced a more balanced structure with several students sharing moderate centralities (Degree $\approx 0.60$, Betweenness $\approx 0.10$). These results indicate that while classroom environment reshapes group structures, personality-driven individual traits remain relatively stable and interpretable.

To support seamless human–agent interaction, we developed an initial version of the EduVerse visual interaction interface. The interface adopts a ChatGPT-style input panel through which users can communicate with virtual student agents in real time. At each timestep, EduVerse displays each agent's automatically generated Plan–Monitor–Regulate reasoning chain alongside its updated behavior–emotion–cognition states, providing an interpretable snapshot of classroom dynamics. Users may also click on any student avatar to open a detailed panel that reveals the agent's internal reasoning trace and state transitions (see Fig. A4). This interface design enhances transparency, facilitates interactive debugging, and enables human participants to flexibly inspect or influence agent behavior within multi-agent classroom simulations.

Human–agent interaction tests, reported in Tab. A12, further demonstrate EduVerse's adaptability. In Task 1 (peer chatting), the socially inclined Zhang Tao responded in most cases, with success rates of 0.533, 0.667, and 0.733 in Lecture, Collab, and Round respectively, and occasionally initiated chats himself in collaborative or roundtable settings. By contrast, in Task 2 (peer academic response), the conscientious Zhang Jie rarely engaged during class, yielding lower success rates (0.267–0.400). In Tasks 3 (teacher answering) and 4 (teacher intervention), success rates consistently reached 1.000 across all environments, confirming that the teacher agent reliably answered questions and actively intervened in off-task behaviors.

Taken together, these supplementary results confirm that EduVerse not only reproduces realistic group dynamics and individual differences but also sustains authentic role-driven behavior when human users are integrated. Socially oriented students show high willingness to engage, academically conscientious students remain task-focused, and teacher agents reliably maintain instructional order. This underscores the robustness and applicability of EduVerse in complex interaction and human-in-the-loop classroom scenarios.

## D.5 EXPERIMENT III

### D.5.1 DEFINITION AND THEORETICAL BASIS OF EFFECTIVE LEARNING TRAJECTORIES

Importantly, an effective trajectory does *not* imply monotonic improvement. Educational psychology and dynamic systems theories of cognitive development (Council et al. (2000); Thelen & Smith (1994); Rodrigues et al. (2023)) show that real learners typically exhibit **nonlinear, oscillatory, and spiral** patterns of growth. Short-term plateaus, fluctuations, or temporary regressions commonly occur when students encounter complex concepts or increased cognitive load, and these patterns possess meaningful pedagogical significance.

Within this theoretical framework, we define an **effective learning trajectory** as one that:

- exhibits an **overall upward developmental trend** across sessions;
- demonstrates **locally plausible fluctuations** reflecting cognitive load or task difficulty;
- shows evidence of **self-correction** driven by feedback and metacognitive regulation.

This definition is grounded in three considerations. First, real student behavior is inherently phase-dependent and unstable; fluctuations often signal knowledge consolidation or strategic adjustment. Second, smooth, strictly increasing curves typically represent *idealized expert models* rather than authentic student learning dynamics. Third, because EduVerse incorporates memory and metacognitive mechanisms through the Plan–Monitor–Regulate (PMR) cycle, it aims to generate trajectories that follow a realistic **"progress–adjust–advance"** pattern consistent with cognitive-development theory.

### D.5.2 DETAILED RESULTS

Table A13: **Longitudinal positive transition rates of student agent behavior, emotion, and cognition across four instructional sessions.** Values are normalized to $[0, 1]$; **Overall** indicates the average across three dimensions.

| Session | Student | B_Pos | E_Pos | C_Pos | Overall |
|---|---|---|---|---|---|
| *1* | Wang Fang | 0.000 | 0.138 | 0.034 | 0.057 |
| | Zhang Jie | 0.379 | 0.966 | 0.103 | 0.483 |
| | Zhang Yan | 0.103 | 1.000 | 0.069 | 0.391 |
| | Li Wei | 0.759 | 0.966 | 0.172 | 0.632 |
| | Liu Li | 0.000 | 0.034 | 0.000 | 0.011 |
| | Zhang Tao | 0.034 | 0.207 | 0.000 | 0.080 |
| *2* | Wang Fang | 0.583 | 0.125 | 0.125 | 0.278 |
| | Zhang Jie | 0.500 | 1.000 | 0.250 | 0.583 |
| | Zhang Yan | 0.292 | 1.000 | 0.125 | 0.472 |
| | Li Wei | 0.375 | 1.000 | 0.208 | 0.528 |
| | Liu Li | 0.125 | 0.792 | 0.000 | 0.306 |
| | Zhang Tao | 0.583 | 0.333 | 0.125 | 0.347 |
| *3* | Wang Fang | 0.500 | 0.567 | 0.067 | 0.378 |
| | Zhang Jie | 0.533 | 1.000 | 0.167 | 0.567 |
| | Zhang Yan | 0.867 | 1.000 | 0.367 | 0.744 |
| | Li Wei | 0.633 | 1.000 | 0.133 | 0.589 |
| | Liu Li | 0.033 | 0.067 | 0.000 | 0.033 |
| | Zhang Tao | 0.367 | 0.700 | 0.100 | 0.389 |
| *4* | Wang Fang | 0.192 | 0.346 | 0.077 | 0.205 |
| | Zhang Jie | 0.654 | 1.000 | 0.154 | 0.603 |
| | Zhang Yan | 0.923 | 1.000 | 0.115 | 0.679 |
| | Li Wei | 0.192 | 1.000 | 0.115 | 0.436 |
| | Liu Li | 0.269 | 0.231 | 0.038 | 0.179 |
| | Zhang Tao | 0.385 | 0.308 | 0.077 | 0.256 |

To complement the main text on long-term evolution, this section applies the above equations to compute the positive transition rate $R^+$ across behavior (B), emotion (E), and cognition (C), as summarized in Tab. A13. Overall, we observe a pattern of *gradual improvement followed by a mild*

*pullback*: many students improve steadily from Session 1 to 3 (e.g., Zhang Yan: $0.391 \rightarrow 0.744$; Wang Fang: $0.057 \rightarrow 0.378$; Zhang Jie: $0.483 \rightarrow 0.567$), whereas Session 4 shows partial regressions under higher cognitive demands (e.g., Wang Fang: $0.378 \rightarrow 0.205$; Zhang Tao: $0.389 \rightarrow 0.256$). We also observe "high start—dip—partial recovery" patterns (e.g., Li Wei: $0.632 \rightarrow 0.528 \rightarrow 0.589$), followed by another decline in Session 4 (0.436).

Dimension-wise, **behavior** is most sensitive, with several students peaking in Session 2 or 3 (e.g., Zhang Yan's B_Pos: $0.103 \rightarrow 0.867 \rightarrow 0.923$). **Emotion** remains high and stable for many students, reflecting effective teacher regulation (e.g., Zhang Jie and Li Wei with E_Pos = 1.000 in Sessions 2–4), though some show fluctuations (e.g., Wang Fang: 0.125/0.567/0.346; Liu Li drops after a high of 0.792). **Cognition** progresses more slowly with larger variability (e.g., Zhang Yan's C_Pos spikes to 0.367 in Session 3 and softens to 0.115 in Session 4), consistent with the view that cognitive growth requires extended accumulation and reflection.

At the individual level, trajectories reveal personality-consistent stability with interpretable divergence: Zhang Jie remains high and stable (Overall: 0.483/0.583/0.567/0.603); Zhang Yan makes a pronounced leap in Session 3 and sustains a high level in Session 4 (0.744/0.679); Li Wei starts high, dips, then partially recovers ($0.632 \rightarrow 0.528 \rightarrow 0.589$) before declining again (0.436); Wang Fang improves then recedes ($0.057 \rightarrow 0.378 \rightarrow 0.205$); Liu Li remains low overall with episodic recovery (0.011/0.306/0.033/0.179); and Zhang Tao shows greater volatility and context sensitivity (0.080/0.347/0.389/0.256). Altogether, $R^+$ provides a compact and interpretable quantification of learning progression and self-regulation, reinforcing the main text's conclusions on long-term evolution and individual differentiation (see Tab. A13).

## D.6 EXPERIMENT IV

To examine whether EduVerse generalizes beyond language-arts instruction, we extend our evaluation to a junior-secondary world-history lesson on the Renaissance. This experiment keeps the teacher agent, student personas, and interaction settings unchanged, altering only the subject domain, which differs substantially from Chinese literature in discourse style, instructional goals, and knowledge structure. We analyze model behavior from two perspectives: IRF discourse structures and student group interaction across three classroom layouts (lecture, collab, round).

### D.6.1 IRF STRUCTURE GENERALIZATION

Real-world evidence shows that IRF patterns vary substantially across subjects. Chinese literature typically emphasizes open-ended questioning, personal interpretation, and affective expression, resulting in frequent teacher prompts and dense IRF cycles. History lessons, by contrast, prioritize chronological reasoning, factual recall, and causal explanation. Questions are more convergent, and teachers rely more on continuous exposition, yielding sparser IRF structures with lower overall questioning frequency.

EduVerse successfully reproduces these subject-specific tendencies. Under identical teacher instructions, the virtual history class exhibits a noticeably lower IRF density than the Chinese literature classes (Spring, The Emperor's New Clothes, Dedication and Joy). Students respond to factual prompts in shorter turns, and the teacher initiates fewer open-ended probes—mirroring authentic disciplinary norms.

Despite these shifts in frequency, the canonical IRF sequence ("Initiation → Response → Feedback") remains stable across subjects. Students continue to provide aligned responses, and the teacher's feedback remains structurally appropriate. This consistency indicates that EduVerse maintains structural discourse robustness, while still adapting interaction frequency to the demands of a new discipline. These results provide initial evidence that the system captures cross-disciplinary transfer of pedagogical interaction patterns.

### D.6.2 GROUP INTERACTION ANALYSIS

We further analyze student-to-student social interactions using networks extracted from three seating layouts. Visualizations (Fig. A5) reveal clear cross-subject differences: Chinese literature lessons produce more peer elaboration chains, especially in collaborative and round-table layouts, aligning with the subject's emphasis on discussion and interpretive sharing. The history lesson shows shorter, more localized interaction edges, with fewer multi-hop exchanges—consistent with subject norms that require individual comprehension before discussion.

Network metrics confirm these observations. Across layouts, the history lesson yields network densities between 0.33 and 0.50, indicating that peer interaction still emerges, even in a domain with less built-in discussion. Degree averages (1.50–1.67) further demonstrate that students maintain a baseline level of collaborative engagement rather than collapsing into purely teacher-driven interaction.

These findings suggest that EduVerse adapts to different instructional cultures while preserving coherent, personality-driven peer interactions. The system demonstrates transferable group-behavior dynamics: interaction structures reorganize to match disciplinary demands, yet remain socially meaningful and pedagogically aligned.

### D.6.3 BEHAVIOR DISTRIBUTION OF INDIVIDUAL LEVEL ACROSS DIFFERENT SUBJECTS

Across subjects, lessons, and layout conditions, student agents exhibit distinct yet personality-consistent behavioral tendencies. As shown in Fig. A6- A11, the same student displays different behavior patterns when switching from Chinese literature to history, and from narrative texts to argumentative essays, reflecting subject-dependent cognitive demand and discourse style. However, the relative behavioral style of each agent remains stable — highly engaged learners (e.g., **Zhang Jie** ) consistently show interactive and positive states across subjects, while more reserved or fluctuating learners (e.g., **Zhang Tao** ) maintain their characteristic variability. Overall, the results demonstrate that EduVerse captures both context-driven behavioral adaptation and trait-driven intra-individual stability, aligning with established findings in educational psychology.

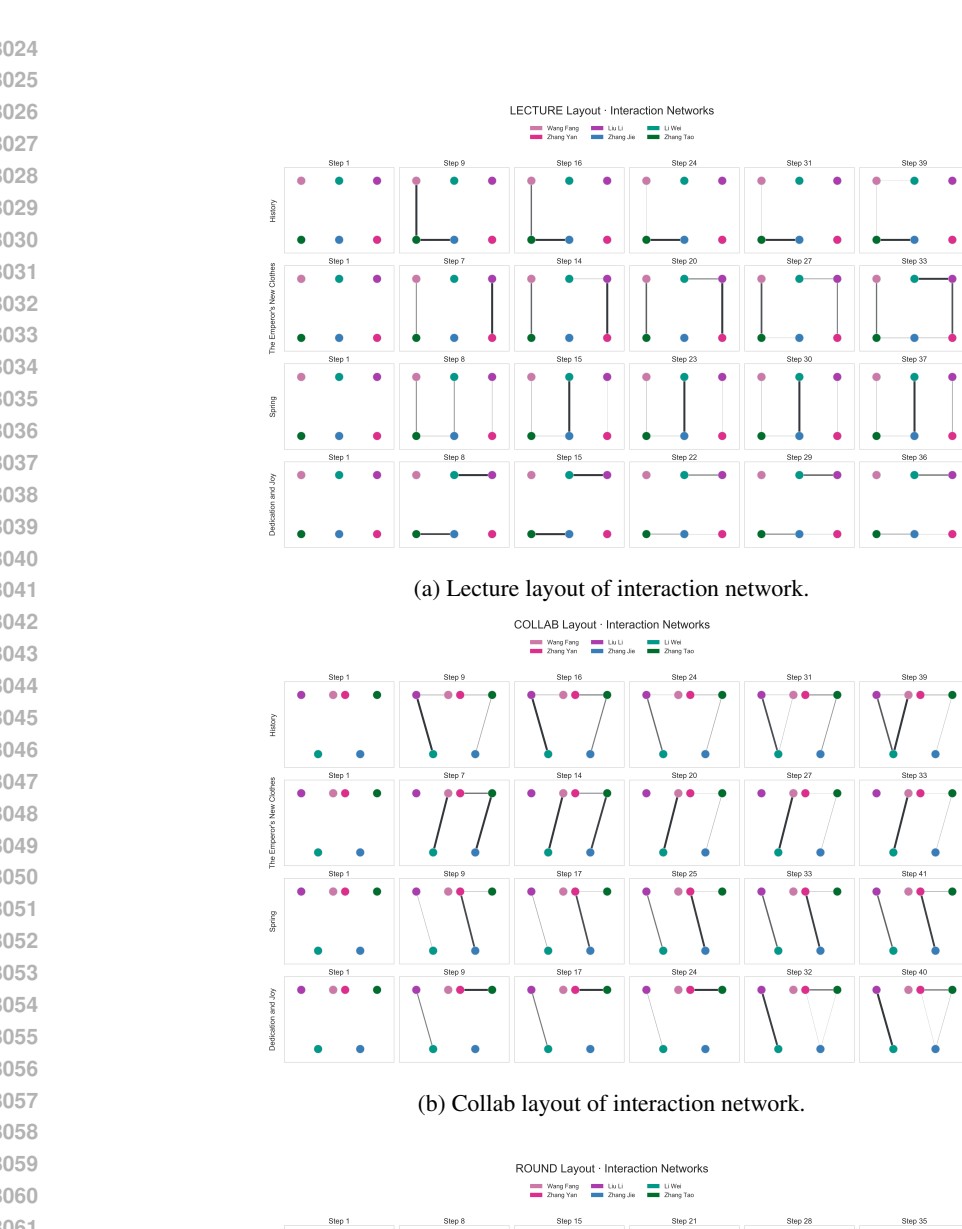

(a) Lecture layout of interaction network.

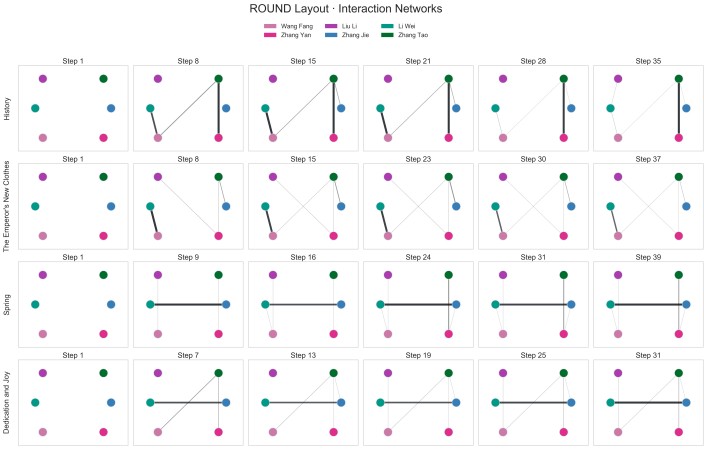

(b) Collab layout of interaction network.

(c) Round layout of interaction network.

Figure A5: Students Interaction Network in different Layouts

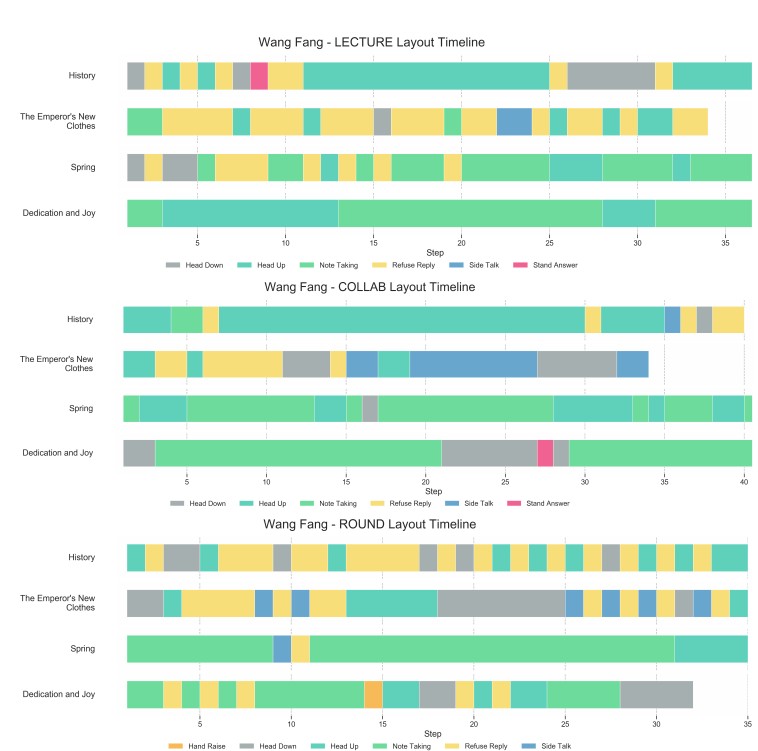

Figure A6: The behavior distribution of **Wang Fang** in different layouts.

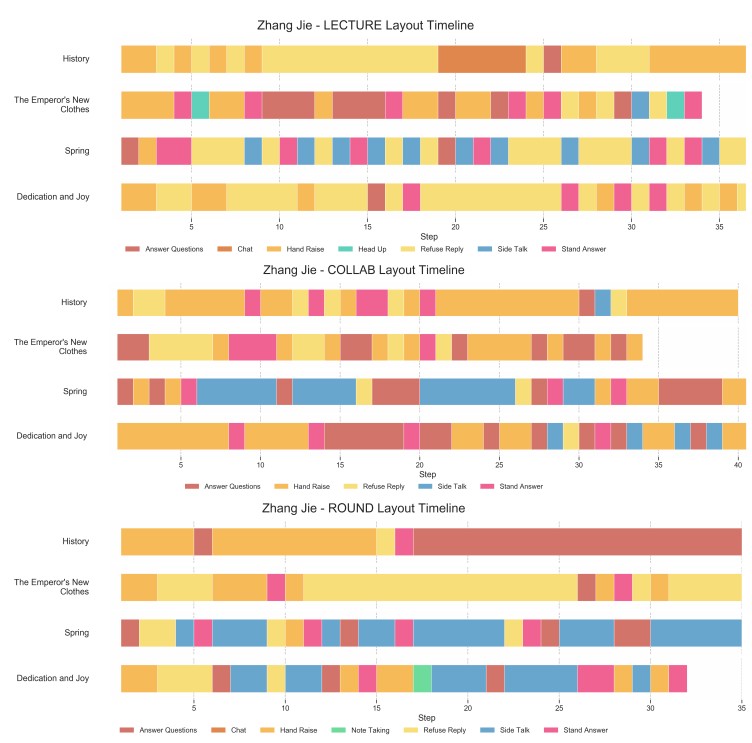

Figure A7: The behavior distribution of **Zhang Jie** in different layouts.

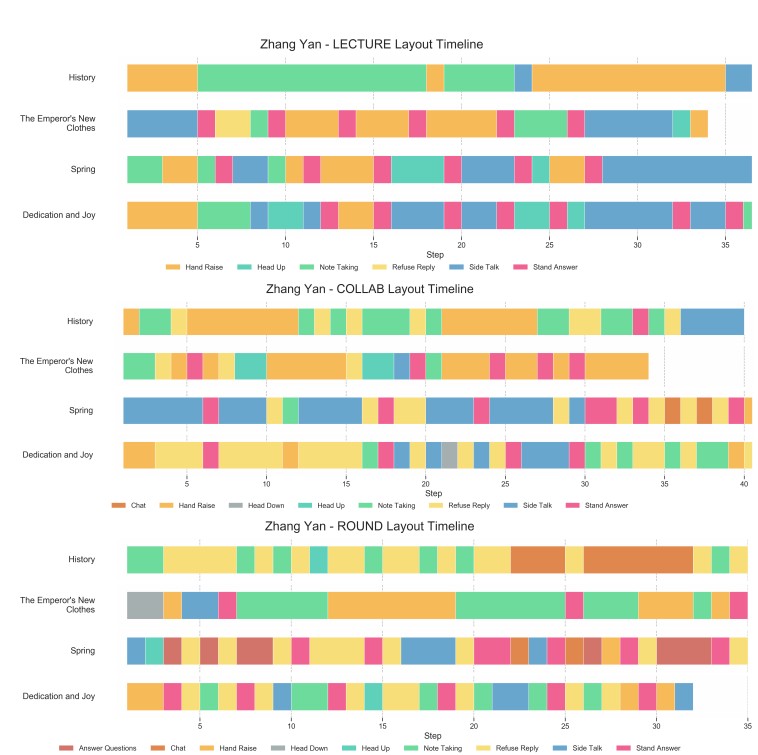

Figure A8: The behavior distribution of **Zhang Yan** in different layouts.

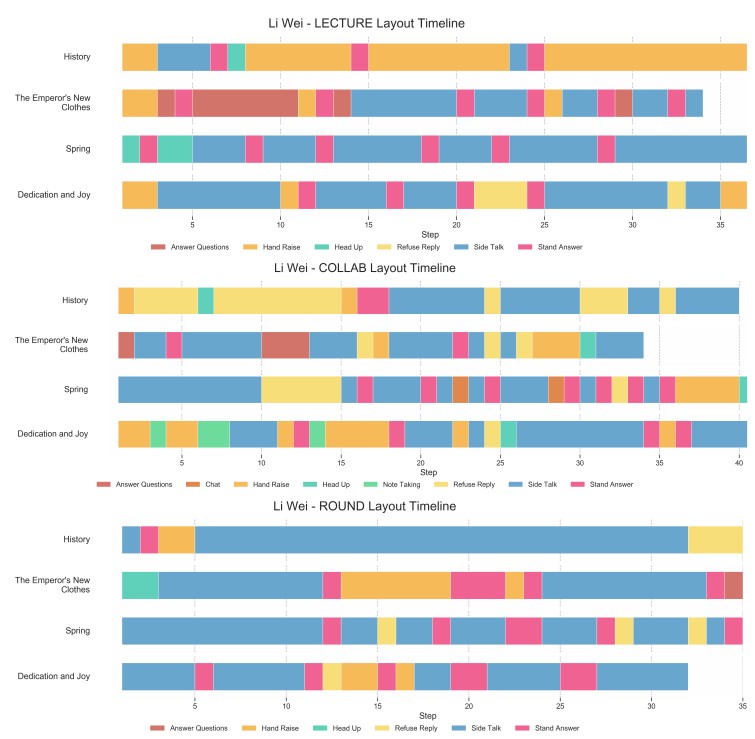

Figure A9: The behavior distribution of **Li Wei** in different layouts.

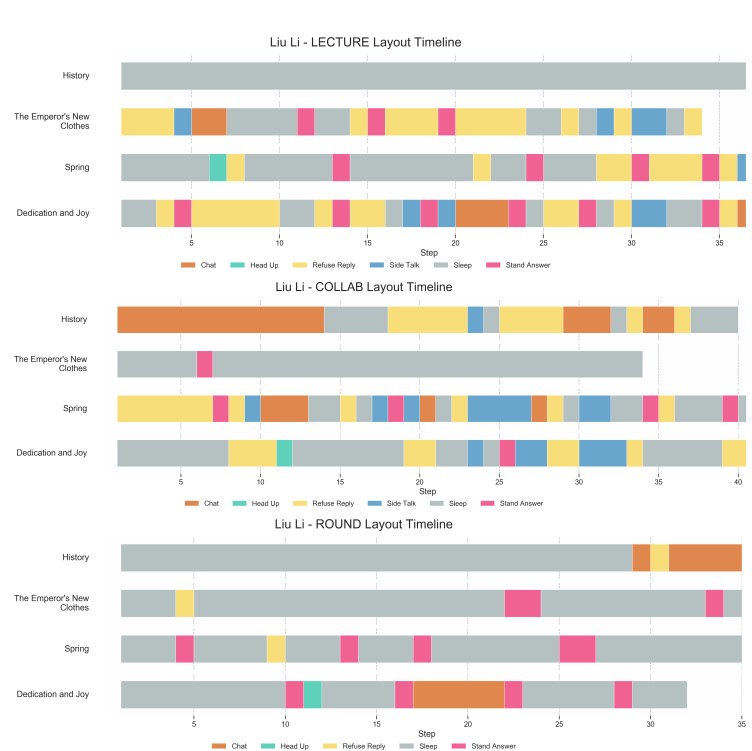

Figure A10: The behavior distribution of Liu Li in different layouts.

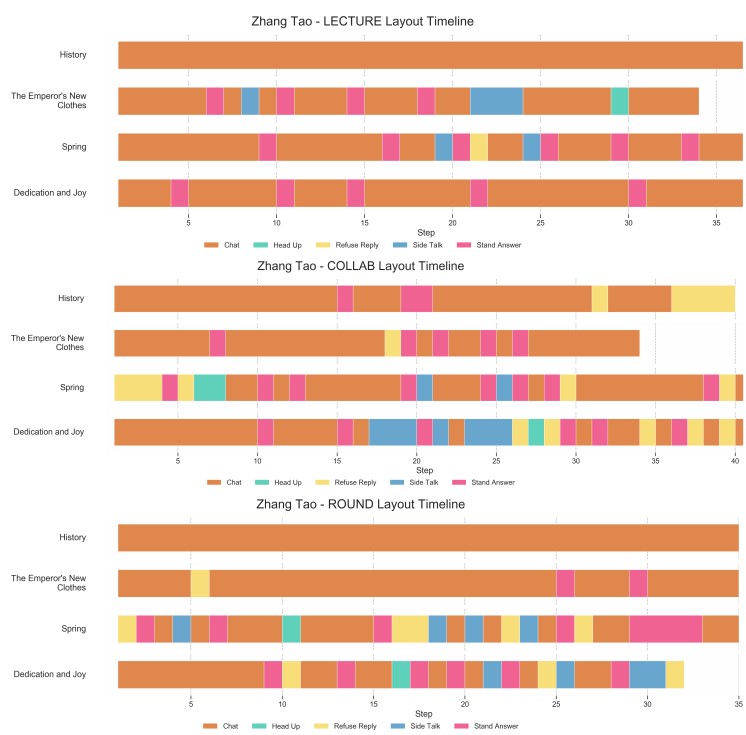

Figure A11: The behavior distribution of Zhang Tao in different layouts.

# E  MULTI-AGENT INSTRUCTIONAL INTERACTION CASES ACROSS GENRE-SPECIFIC TASKS

To illustrate how virtual students respond under varying instructional task types, we present representative multi-agent interaction cases from three distinct genres: the lyrical prose *Spring*, the foreign fable *The Emperor's New Clothes*, and the argumentative essay *Dedication and Joy*. These texts differ in linguistic style, cognitive structure, and affective demand, offering a natural basis for evaluating behavioral variation.

For each task, we sampled three consecutive instructional time steps from the full simulation. We report the complete behavioral logs of selected student agents during these windows, including learning plans, cognitive levels, emotional states, and social behaviors. These interaction snapshots offer micro-level evidence of how agents coordinate cognition, emotion, and participation in real-time.

These examples illustrate how agent behavior adapts to differences in instructional phase, task genre, and teacher–student dynamics. They also provide direct evidence of cross-genre generalizability and personality-driven response stability within the simulation system.

---

**Case: Multi-Agent Interaction in "Spring" (Step 8)**

**Teacher Content:** "Excellent, everyone. Your understanding and analysis are impressive. Now, I would like to ask: While reading this text, did you notice how Mr. Zhu Ziqing vividly portrayed the beauty of spring through delicate descriptions? For example: 'Peach trees, apricot trees, and pear trees are all blooming in competition—bright red like fire, pink like clouds, white like snow. The flowers are sweet-scented; close your eyes, and the trees seem already full of peaches, apricots, and pears.' With these metaphors and personifications, Zhu Ziqing presents spring as a sensory experience. Zhang Yan , could you share how you interpret this passage?"
**Teacher Emotion:** Encouragement **Teacher Behavior:** Questioning
**Student Responses:**

- Wang Fang → Zhang Yan : Side talk – "I agree with your point. Zhu Ziqing's use of personification really helps us feel the vibrancy of spring—it's like a person waking up, full of hope and energy." (Emotion: Positive; Cognition: Application)

- Zhang Jie : Question – "Teacher, I really love this passage. It feels like he's painting spring with a brush. The sentence 'Peach trees, apricot trees, pear trees. . . ' vividly brings to life the competitive blooming. And 'the flowers are sweet-scented; close your eyes, and the trees seem already full of fruit'—this conveys not just the beauty, but also the sweetness and hope of spring. I think this is what Zhu Ziqing intended: spring is not just visual, but deeply emotional." (Emotion: Positive; Cognition: Application)

- Zhang Yan : Stand-up response – "I believe Mr. Zhu's descriptions truly immerse us in the atmosphere of spring. The use of personification and metaphor makes the imagery come alive. That line about blooming trees really shows the vitality of spring." (Emotion: Positive; Cognition: Application)

- Li Wei : Question – "I agree with the teacher and classmates. Zhu's descriptions are not only vivid but also metaphorical. They reflect the beauty of life and hope. I think this is what he wants us to feel—resilience and positivity even in the face of difficulties. Do others feel the same?" (Emotion: Positive; Cognition: Application)

- Liu Li → Zhang Tao : Side talk – "Maybe. . . I'm not really sure." (Emotion: Positive; Cognition: Application)

- Zhang Tao → Liu Li : Side talk – "Ugh, don't overthink it. It's not a big deal anyway." (Emotion: Negative; Cognition: Understanding)

**Teacher Feedback:** "Liu Li , great improvement—keep up your active participation. Zhang Jie , your engagement is excellent—looking forward to your insights. Zhang Yan , your communication and understanding are outstanding—keep it up, and remember to listen to others too."

---

---

**Case: Multi-Agent Interaction in "Spring" (Step 9)**

**Teacher Content:** "Wonderful work, everyone. You've gained deep insights into Zhu Ziqing's text. Now let's think: How does he express his emotions when portraying spring? Through what language or imagery can we feel his admiration for spring? Wang Fang , would you like to share your thoughts first?"

**Teacher Emotion:** Encouragement **Teacher Behavior:** Questioning

**Student Responses:**

- Wang Fang : Stand-up response – "Uh. . . I think Zhu Ziqing. . . in 'Spring'. . . he uses delicate descriptions to. . . vividly portray spring. He uses emotional language to describe flowers, trees, rivers. . . and the word 'lively' to reflect his feelings of joy. That's how I understand his admiration." (Emotion: Positive; Cognition: Application)

- Zhang Jie : Hand raised – "Sure. I think his vivid language makes it feel like we're witnessing spring with our own eyes. It's not just about beauty—it's about his deep love for this season. His writing really impresses me." (Emotion: Positive; Cognition: Application)

- Zhang Yan → Wang Fang : Side talk – "Exactly, Wang Fang . The personification he uses really makes spring feel alive, full of hope." (Emotion: Positive; Cognition: Application)

- Li Wei : Head up, listening, no verbal response (Emotion: Positive; Cognition: Understanding)

- Liu Li : Hand raised – "Uh. . . I think Zhu Ziqing. . . really liked spring. . . his words, um. . . show he really loved it." (Emotion: Positive; Cognition: Understanding)

- Zhang Tao → Liu Li : Side talk – "Liu Li , can you explain how Zhu Ziqing shows his love for spring through language and imagery?" (Emotion: Confused; Cognition: Understanding)

**Teacher Feedback:** "Liu Li , great participation and deep thinking—just be mindful of classroom order. Zhang Yan , well done with your insightful understanding of 'Spring'—keep up the collaborative spirit."

---

**Case: Multi-Agent Interaction in "Spring" (Step 10)**

**Teacher Content:** "Now let's examine the structure of 'Spring'. Zhu Ziqing divides the essay into three parts. First, he begins with anticipation: 'Looking forward, looking forward, the east wind is coming, and spring's footsteps are near.' Then uses vivid metaphors and delicate imagery to depict spring scenes like blooming trees. Finally, he uses personification—spring is a newborn baby, a little girl, a strong youth—bringing vitality to the season. This layered structure gives the essay rhythm and emotional resonance. Zhang Jie , would you like to share your understanding of this structure?"

**Teacher Emotion:** Encouragement **Teacher Behavior:** Explanation

**Student Responses:**

- Wang Fang → Zhang Yan : Side talk – "Zhang Yan . . . you're right. . . Zhu's personification really shows the vitality and hope of spring!" (Emotion: Positive; Cognition: Application)

- Zhang Jie : Stand-up response – "Of course, teacher. I think the structure is brilliant. The beginning sets the tone of anticipation, the middle provides vivid descriptions, and the end gives spring symbolic meaning—energy, rebirth, youth. It's rhythmic and emotionally deep." (Emotion: Positive; Cognition: Application)

- Zhang Yan → Wang Fang : Side talk – "I'm glad you felt the vitality too, Wang Fang . His words awaken everything—dispelling winter and bringing hope." (Emotion: Positive; Cognition: Application)

- **Li Wei** → **Zhang Yan** : Side talk – "**Zhang Yan** , what's your unique take on Zhu's metaphors? I love his personification—it inspires our own writing, don't you think?" (Emotion: Positive; Cognition: Understanding)

- **Liu Li** → **Zhang Tao** : Side talk – "**Zhang Tao** , I think Zhu's vivid words reflect not just spring, but also his love for life and respect for nature." (Emotion: Positive; Cognition: Understanding)

- **Zhang Tao** → **Liu Li** : Side talk – "Hmm, I like 'Back Shadow' more than 'Spring'. Have you read that one?" (Emotion: Positive; Cognition: Understanding)

**Teacher Feedback:** "**Zhang Jie** , wonderful—your structural analysis is spot on. **Liu Li** , your input shows a deep grasp of the text—keep it up! **Zhang Yan** , excellent discussion—maintain your positive attitude and spirit of exploration."

---

**Case: Multi-Agent Interaction in "Dedication and Joy" (Step 12)**

**Teacher Content:** "Now, let us explore the meaning of 'finding joy in one's work' (*leye*). It refers to discovering happiness in our work or studies, enjoying what we do. In this way, our tasks become less of a burden and more of a pleasure. How does Liang Qichao interpret *leye*? In his essay, he writes, 'Be at peace with your dwelling, and find joy in your work'—suggesting that we should experience peace and pleasure in what we do. **Zhang Jie** , could you share your understanding of *leye*?"

**Teacher Emotion:** Encouragement **Teacher Behavior:** Questioning

**Student Responses:**

- **Wang Fang** : Note-taking – No verbal response (Emotion: Positive; Cognition: Understanding)

- **Zhang Jie** : Stand-up response – "I'm happy to share my view. I think *leye* means finding joy and value in what we do. As Liang Qichao said, 'Be at peace with your dwelling, and find joy in your work'—to me, this means experiencing happiness and satisfaction through learning or working. When we love what we do, it becomes our source of joy rather than pressure." (Emotion: Positive; Cognition: Application)

- **Zhang Yan** : Hand raised – "I think *leye* is about finding enjoyment in our learning or work. As Liang Qichao says, we should find peace and joy in our duties. That way, work becomes something we enjoy. This reflects my experience—I enjoy the learning process and hope others can find that joy too." (Emotion: Positive; Cognition: Application)

- **Li Wei** : Hand raised – "I believe *leye* is about finding joy and satisfaction in the process of doing things. Liang Qichao's quote applies not only to work but also to learning. If we find joy in learning, we'll better understand knowledge and be more motivated. But I wonder, when facing stress or difficulties, how can we still embrace *leye*?" (Emotion: Positive; Cognition: Application)

- **Liu Li** : Head down sleeping – No response (Emotion: Negative; Cognition: Memory)

- **Zhang Tao** → **Liu Li** : Chatting – "**Liu Li** , did you watch that basketball game yesterday? That dunk was insane!" (Emotion: Negative; Cognition: Memory)

**Teacher Feedback:** "**Liu Li** , great to see your improvement—keep participating actively. **Zhang Jie** , your engagement is excellent. **Zhang Yan** , you've shown strong understanding and communication—keep it up, and remember to listen to your peers."

---

**Case: Multi-Agent Interaction in "Dedication and Joy" (Step 13)**

**Teacher Content:** "Let's now see how Liang Qichao argues for the importance of *leye*. In his essay 'Dedication and Joy', he uses the phrase 'Be at peace with your dwelling, and find joy in your work' to explain his perspective. He believes that only when we feel peace and happiness in our duties can we truly understand what *leye* means. The 'joy' here refers not

only to enjoyment but also to love and appreciation. Zhang Jie raised a great question: is *leye* limited to our work, or can it also apply to learning, life, and hobbies? Li Wei , what do you think?"

**Teacher Emotion:** Encouragement **Teacher Behavior:** Questioning
**Student Responses:**

- Wang Fang : Note-taking – No response (Emotion: Positive; Cognition: Application)

- Zhang Jie : Question – "I fully agree with Liang Qichao. He helps us realize that only by loving our work can we find true joy and meaning in it. Then it's not a burden anymore. But I have a question: can *leye* apply to things beyond work, like learning, life, or hobbies? I think anything we love deeply could be part of *leye*. What do you think, teacher?" (Emotion: Positive; Cognition: Application)

- Zhang Yan : Question – "Teacher, I've also been thinking about Liang's view. He's suggesting that when we genuinely love and enjoy what we do, it becomes a joy. This attitude should apply to learning as well—finding enjoyment rather than just completing tasks. But what if we're not feeling that joy? How should we adjust our mindset?" (Emotion: Positive; Cognition: Application)

- Li Wei : Stand-up response – "Liang's reasoning is inspiring. He shows that when we love what we do, it stops being pressure—it becomes joy. But I'm a bit confused about 'being at peace with your dwelling.' Does that mean we must accept and embrace our current life situation to experience *leye*? I'd love to hear others' thoughts." (Emotion: Positive; Cognition: Application)

- Liu Li : Head up listening – No response (Emotion: Positive; Cognition: Understanding)

- Zhang Tao : Head up listening – No response (Emotion: Positive; Cognition: Understanding)

**Teacher Feedback:** "Zhang Yan , your thoughtful and active participation is wonderful—keep it up! Zhang Jie , your engagement and depth of understanding help lead the class. Li Wei , your ideas are insightful—continue contributing in discussions!"

---

**Case: Multi-Agent Interaction in "Dedication and Joy" (Step 14)**

**Teacher Content:** "Your reflections have been fantastic. Zhang Jie asked a great question—can *leye* apply beyond work, such as to learning or hobbies? I believe *leye* is broad—it applies to any domain where we invest passion. As for Zhang Yan 's question about learning struggles: First, we must accept that difficulties are part of the process. Then, try to see them as challenges and growth opportunities. That's how we rediscover joy in learning. Does anyone want to share how they've done this?"

**Teacher Emotion:** Encouragement **Teacher Behavior:** Questioning
**Student Responses:**

- Wang Fang : Hand raised – "Uh, I. . . I think when we face learning difficulties. . . we shouldn't give up. We. . . should accept them and. . . find solutions. If we stay optimistic, we'll rediscover *leye*." (Emotion: Positive; Cognition: Application)

- Zhang Jie : Stand-up response – "Teacher, I think *leye* is a broad concept—it definitely applies to learning, life, and hobbies. If we commit and love what we do, that's *leye*. When facing challenges, we should adjust our mindset and find opportunities—that's how we enjoy learning." (Emotion: Positive; Cognition: Application)

- Zhang Yan : Stand-up response – "I agree. *Leye* means loving not just work but all aspects of life. When we face challenges in learning, changing our perspective helps us turn obstacles into joy. Has anyone had a similar experience? How did you regain your joy in learning?" (Emotion: Positive; Cognition: Application)

- Li Wei : Question – "Everyone's ideas are inspiring. I also believe *leye* is a holistic attitude that includes work, learning, and hobbies. As long as we're passionate,

we'll find joy and self-worth. Though there may be difficulties, they are growth opportunities. I'd like to hear how others adjusted their mindset during such times." (Emotion: Positive; Cognition: Understanding)

- **Liu Li** : Head up listening – No response (Emotion: Positive; Cognition: Understanding)

- **Zhang Tao** : Head up listening – No response (Emotion: Positive; Cognition: Understanding)

**Teacher Feedback:** "**Wang Fang** , great focus and thinking—keep it up, and don't let anxiety affect your learning. **Zhang Yan** , your helpfulness and ability to apply learning are admirable—keep going!"

---

**Case: Multi-Agent Interaction in "The Emperor's New Clothes"(Step 12)**

**Teacher Content:** "Today, we will be reading *The Emperor's New Clothes*, a satirical tale by the famous Danish author Hans Christian Andersen. Andersen is one of the most renowned figures in children's literature, with a diverse range of works spanning fairy tales, novels, poems, and plays. Some of his most beloved stories include *The Little Match Girl*, *The Ugly Duckling*, and *The Little Mermaid*. In his fairy tales, Andersen often reveals human flaws and social injustice from a unique perspective, offering us valuable life lessons."

**Teacher Emotion:** Neutral **Teacher Behavior:** Lecture

**Student Responses:**

- **Wang Fang** : Head up listening – No response (Emotion: Positive; Cognition: Understanding)

- **Zhang Jie** : Question – "Teacher, I find this story very interesting. The emperor is tricked by two swindlers who claim that only intelligent people can see their fabric. The emperor never checks the clothes himself but sends others. I wonder—if he had gone himself, would he have seen through the deception? Or if he had the courage to admit he saw nothing, could he have avoided the embarrassment? This story not only mocks vanity and fear of losing face, but also reminds us to have the courage to face the truth. What do others think?" (Emotion: Positive; Cognition: Analysis)

- **Zhang Yan** : Question – "Teacher, this story made me reflect a lot. Andersen's works always carry deep meaning. Through this tale, he highlights human vanity and blind conformity, showing us the importance of honesty and truth. That's probably why his works remain so popular—they are both entertaining and meaningful. I also want to ask: does this story still hold relevance today? Don't we sometimes act like the emperor, pursuing superficial appearances while ignoring real value?" (Emotion: Positive; Cognition: Analysis)

- **Li Wei** : Question – "Teacher, I've also thought about what **Zhang Jie** and **Zhang Yan** said. To me, this story not only satirizes people who care too much about appearances, but also encourages independent thinking. That's a key takeaway Andersen wants us to have. I'd love to hear how others interpret it." (Emotion: Positive; Cognition: Analysis)

- **Liu Li** → **Zhang Tao** : Chatting – "**Zhang Tao** , have you watched the latest Marvel movie *Eternals*? I heard the plot is great and the effects are amazing. What did you think?" (Emotion: Negative; Cognition: Memory)

- **Zhang Tao** → **Li Wei** : Chatting – "**Li Wei** , do you think our school's football team will win this weekend's match?" (Emotion: Negative; Cognition: Memory)

**Teacher Feedback:** "**Zhang Tao** , please stay focused and participate actively so you can better understand the story. **Liu Li** , you show strong independent learning ability, but classroom interaction is important too—try joining the discussion!"

**Case: Multi-Agent Interaction in "The Emperor's New Clothes"(Step 13)**

**Teacher Content:** "Students, the questions raised by Zhang Jie , Zhang Yan , and Li Wei were excellent and show deep thinking about the story. Now, can anyone tell me what genre *The Emperor's New Clothes* belongs to? Is it a novel, a poem, an essay, or a fairy tale?"
**Teacher Emotion:** Encouragement **Teacher Behavior:** Questioning
**Student Responses:**

- Wang Fang : Note-taking – No response (Emotion: Positive; Cognition: Application)
- Zhang Jie : Stand-up response – "I think it's a fairy tale. The plot—like the emperor being deceived—is common in fairy tales. And it teaches us a lesson: to think independently and not blindly follow others." (Emotion: Positive; Cognition: Analysis)
- Zhang Yan : Stand-up response – "I agree—it's a fairy tale. Although the characters are familiar—like emperors and tailors—the story includes magical elements such as invisible clothes and collective pretending. It also teaches a moral, which is typical of fairy tales." (Emotion: Positive; Cognition: Application)
- Li Wei : Stand-up response – "I also think it's a fairy tale. It includes fantasy elements and makes us think deeply. But I'd also like to hear what others think." (Emotion: Positive; Cognition: Application)
- Liu Li : Head down reading – No response (Emotion: Negative; Cognition: Understanding)
- Zhang Tao → Liu Li : Chatting – "Liu Li , what book are you reading? It looks interesting." (Emotion: Negative; Cognition: Understanding)

**Teacher Feedback:** "Liu Li , let's try to stay focused—your intelligence should be used to understand the story. Wang Fang , your thoughtful interpretation is excellent—please keep sharing. Zhang Tao , let's all focus on the lesson—we're eager to hear your ideas too."

**Case: Multi-Agent Interaction in "The Emperor's New Clothes"(Step 14)**

**Teacher Content:** "Great work, everyone. Zhang Jie mentioned that if the emperor had the courage to admit he couldn't see the fabric, he could have avoided embarrassment—a sharp observation. Zhang Yan asked about the story's modern relevance, and that's a very valuable question: do we, too, sometimes pursue surface appearances and ignore real value? Li Wei talked about the importance of independent thinking—something we should all cultivate. We'll continue exploring these ideas in future lessons."
**Teacher Emotion:** Encouragement **Teacher Behavior:** Questioning
**Student Responses:**

- Wang Fang : Head up listening – No response (Emotion: Positive; Cognition: Analysis)
- Zhang Jie : Stand-up response – "I think this story is really meaningful. Like Zhang Jie said, the emperor's fear of embarrassment kept him from facing the truth. This happens in real life too. The story teaches us to be brave and value independent thinking—just like Li Wei said. I'll try to apply this understanding in the future and learn from my classmates' views as well." (Emotion: Positive; Cognition: Analysis)
- Zhang Yan : Stand-up response – "I agree with both Zhang Jie and Li Wei . Courage and critical thinking are crucial. I'd add that in chasing a beautiful life, we shouldn't lose sight of what's real. Like the emperor, we risk embarrassment when we chase appearances. This story warns us not to be modern-day emperors." (Emotion: Positive; Cognition: Analysis)
- Li Wei : Stand-up response – "Thank you, teacher. I agree with Zhang Jie 's point about courage and Zhang Yan 's insights on modern relevance. We often do focus too much on appearance and forget real values. This story encourages us to think

independently, face truth, and care about substance over form." (Emotion: Positive; Cognition: Analysis)

- Liu Li → Zhang Yan : Chatting – "Zhang Yan , have you seen that popular TV show *Green Fields Under Moonlight*? The characters are so well-written. You should check it out." (Emotion: Negative; Cognition: Understanding)

- Zhang Tao → Li Wei : Chatting – "Li Wei , did you notice Liu Li was chatting during class? Do you know what they were talking about?" (Emotion: Negative; Cognition: Understanding)

**Teacher Feedback:** "Wang Fang , great job observing and thinking—don't be anxious, we're all learning together. Try sharing your ideas next time. Zhang Yan , your active participation and deep thinking are a real asset to the class. Zhang Jie , your analytical skills and eloquence are excellent—keep up the enthusiasm!"

