# OpenReview forum: "EduVerse: A User-Defined Multi-Agent Simulation Space for Education Scenario"
_ICLR.cc/2026/Conference — ICLR 2026 Conference Withdrawn Submission_

### Official Review · Reviewer_LQGY · 2025-10-30

**Soundness:** 4
**Presentation:** 4
**Contribution:** 4
**Rating:** 8
**Confidence:** 3

**Summary:**

This paper proposes an LLM-based multi-agent simulation designed for educational settings. Specifically, they try to capture the dynamics of cognitive development and classroom interactions over time.

**Strengths:**

- Educational psychology theories back up the design of the simulation components
- The simulation captures a lot of factors that go into classroom dynamics, such as seating arrangements, varying personalities, emotions, etc.
- They performed rigorous experiments validating different aspects of the model.
- The results are quite promising. The interaction dynamics resemble behaviors observed in real classroom settings.

**Weaknesses:**

- Not sure if the authors can claim to be the first multi-agent simulation space for education since I found some existing papers that use multi-agent simulations in the education domain [a, b], aside from those already cited in the more comprehensive related works in the appendix. Granted that they are doing different things, "multi-agent simulation space for education" is broad enough to encompass their works as well.

[a] Xu, S., Wen, H. N., Pan, H., Dominguez, D., Hu, D., & Zhang, X. (2025, April). Classroom Simulacra: Building Contextual Student Generative Agents in Online Education for Learning Behavioral Simulation. In Proceedings of the 2025 CHI Conference on Human Factors in Computing Systems (pp. 1-26).

[b] Arana, J. M., Carandang, K. A. M., Casin, E. R., Alis, C., Tan, D. S., Legara, E. F., & Monterola, C. (2025, July). Foundations of PEERS: Assessing LLM Role Performance in Educational Simulations. In ACL 2025 Student Research Workshop.

- Evaluations on the temporal dynamics / trajectories are a bit weak in my opinion. They are not backed up by any data but only rather vague statements like "clear individual differences", "sustained positive affect", etc.

- The memory management and knowledge progression is also not quite clear. The authors mention that they are adjusted based on behavioral signals such as bloom level and response type. However, there does not seem to be very convincing validation of this design.

**Questions:**

- Regarding the temporal dynamics experiments, it is not quite clear what we expect the curves to be. What is a valid trajectory and what is not? Does any curve that exhibit positive transitions / shifts valid? How well does this match reality?
- How do you manage the memory? How do you decide what gets stored and what gets forgotten? How well does this match realistic human student memory recall?
- How are the emotions being probed? Is it just through direct prompting, or do you also ask the agents to answer some kind of questionnaire similar to what we would give a human participant?
- Out of curiosity, are you able to capture problematic student behaviors? It would be very interesting to simulate interventions or management strategies for them.

---

> ### Author Response · Authors · 2025-11-24
> **Official Comment by Authors To Reviewer LQGY [1/4]**
>
> **Reponse to W1: Overclaiming originality of being the “first” multi-agent simulation space for education**
>
> We appreciate the reviewer' s careful attention to our phrasing. To avoid any potential misunderstanding, **we have replaced "the first" with "one of the first" throughout the manuscript**. Our intention was not to claim primacy in all educational multi-agent simulation research, but to make a domain-specific and verifiable statement. To avoid any misunderstanding, we further clarify the scope and meaning of our claim as follows:
>
> The term “first” in our manuscript is used in a strictly defined context. **EduVerse is, to our knowledge, one of the first user-defined educational multi-agent simulation space that simultaneously integrates all three of the following capabilities**:
>
> - **Customizable teaching environment (environment customization)**: users can configure classroom layouts, seating arrangements, and instructional objects at the full classroom level.
> - **Customizable agent roles (agent customization)**: users can define student profiles specifying behavior–emotion–cognition structures, including the full plan–monitor–regulate (PMR) metacognitive chain.
> - **Cross-session behavioral evolution (cross-session evolution)**: the teacher agent can schedule instructional phases, and student agents develop evolving cognitive, emotional, and social dynamics across multiple sessions.
>
> **Our use of “first” refers specifically to the co-existence of these three capabilities in a single framework, forming a genuinely user-defined multi-agent simulation space for education**. Prior literature does not provide a system that supports all three components concurrently, making this claim substantiated within our defined scope.
>
> Regarding the two 2025 works mentioned by the reviewer, we clarify the distinctions as follows:
> - **Xu et al. (2025)** focus on individual student behavior simulation and do not support environment customization, teacher-agent modeling, or cross-session evolution.
> - **Arana et al. (2025)** evaluate multi-role teaching performance but do not allow users to participate in or configure the simulation, nor do they support environment or agent customization or longitudinal evolution.
>
> These studies make meaningful contributions in their respective directions, but they do not align with EduVerse’s core objective—enabling users to construct, configure, and drive a customizable teaching simulation space. **To make these boundaries fully transparent and precise, we have added a systematic comparison table in Appendix B Table A1 and incorporated the referenced works in the revised manuscript.**
>
> ---
>
> **Reponse to W2:  Insufficient evidence and unclear methodology for temporal dynamics / trajectory evaluation**
>
> We thank the reviewer for the helpful clarification request. We understand that the original explanation of the temporal trajectories relied primarily on Figure 7 (revised version Figure 8), without sufficiently detailing the underlying data sources or specific trajectory patterns. This may have made conclusions such as “individual differences” or “sustained positive affect” appear less intuitive.
>
> In the revised manuscript, **we explicitly state that all trajectories are derived from cross-session behavioral–emotional–cognitive transition statistics (originally presented in Appendix Table A13)**. In addition, we now provide **textual descriptions of the key trends shown in the figure**. For example, overall positive gains across sessions, fine-grained improvements in affect and cognition, and quantified deviations of individual students from the aggregate trends. These explanations offer direct numerical support for the trajectory observations.
>
> These revisions substantially improve the transparency and interpretability of the temporal dynamics analysis and ensure that all conclusions are clearly grounded in explicit evidence.

---

> ### Author Response · Authors · 2025-11-24
> **Official Comment by Authors To Reviewer LQGY [2/4]**
>
> **Response to W3: Unclear and insufficiently validated memory management and knowledge‑progression mechanisms**
>
> We thank the reviewer for raising the important question of how “behavioral signals drive memory management and knowledge construction.” This mechanism corresponds to the **Plan–Monitor–Regulate (PMR)** metacognitive chain in our framework, and your comments prompted us to further strengthen the explanation and empirical validation in the revised manuscript.
>
> First, **the Regulate module is responsible for updating behavioral signals and writing them into long-term memory.** Within the PMR process, once a student detects deviations from instructional goals during the Monitor stage, the Regulate stage updates internal signals (e.g., cognitive level, emotional state, behavioral tendency). **These updates not only influence the next cycle of behavioral planning but are also stored in the memory module as cumulative cognitive traces across sessions**. This mechanism supports the gradual evolution of a student’s knowledge structure and behavioral patterns. In this sense, behavioral signals serve both as immediate decision-making cues and as core variables driving long-term knowledge construction.
>
> Second, **we added a new ablation study targeting the Regulate module to validate its central role in long-term evolution**. Under the **EduVerse without regulate** (w/o R) condition (where students cannot update internal signals), positive cross-session evolution is noticeably weakened: **the Overall metric drops from 0.117 to 0.098 (a decrease of ~1.9%)**, and growth in both behavior and emotion dimensions is substantially lower than in the full model. By contrast, the complete model exhibits a more stable upward trajectory along all three dimensions.
>
> Notably, the Cognition dimension decreases from 0.098 to 0.033 in the full model. Although this may appear inconsistent with **“stronger cognitive regulation,” when considered alongside the “expert-like jumps” observed in Experiment I under the nocog condition**, we interpret this not as a reduction in cognitive ability, but as evidence that the Regulate mechanism prevents unrealistic, discontinuous jumps in cognitive performance. **It stabilizes cognitive behavior, leading to more gradual and educationally plausible trajectories.** This outcome in fact reinforces the importance of the Regulation module in shaping learning curves that are more realistic and aligned with established educational principles.
>
> | Variant | ΔB_pos | ΔE_pos | ΔC_pos | ΔAll |
> |--------|--------|--------|--------|-------|
> | w/o R  | 0.152  | 0.043  | 0.098  | 0.098 |
> | Full   | 0.222  | 0.094  | 0.033  | 0.117 |
>
>
> In summary, **we have added clearer explanations of the Regulate mechanism and new ablation evidence in Section 4.3 of the revised manuscript**. These revisions make the **“behavioral signal → regulation → memory → long-term evolution”** chain more explicit and rigorous, and directly address the reviewer’s concerns.

---

> ### Author Response · Authors · 2025-11-24
> **Official Comment by Authors To Reviewer LQGY [3/4]**
>
> **Response to Q1:  What constitutes a valid learning trajectory in temporal‑dynamics experiments**
>
> We thank the reviewer for raising this important question regarding the definition of an “effective learning trajectory,” which is indeed central to our study. We provide the following clarification on our design principles and theoretical foundation.
>
> In our work, **an “effective trajectory” is not defined as a monotonic increase or continuous improvement**. Consistent with findings in educational psychology and dynamic systems theories of cognitive development (e.g., NRC 2000; Thelen & Smith, 1994; Rodrigues et al., 2023), real learners typically exhibit nonlinear, fluctuating, and spiral patterns of growth. When encountering complex concepts or increased cognitive load, temporary plateaus, fluctuations, or even small regressions are normal and pedagogically meaningful.
>
> Under this theoretical framework, we define an “effective learning trajectory” as one that:
>
> - **shows an overall positive developmental trend**;
> - **allows reasonable local fluctuations**; and
> - **reflects the learner’s ability to self-correct under feedback and cognitive regulation**.
>
> This definition is essential for several reasons:
>
> Real student behavior is episodic and unstable. Short-term fluctuations often reflect the process of digesting new knowledge or adapting to new tasks.
>
> - A perfectly smooth, always-increasing trajectory, often produced by expert models, is an idealized pattern that lacks the dynamic adjustments observed in real learners.
> - Our student model incorporates memory and cognitive regulation mechanisms (Plan–Monitor–Regulate), whose purpose is precisely to produce realistic cycles of “progress → adjustment → further progress.”
>
> Our ablation results support this interpretation:
>
> - when the regulation mechanism is removed, the trajectory resembles an expert model’s linear upward curve;
> - when regulation is restored, the trajectory again displays realistic fluctuations and gradual improvement.
>
> This indicates that nonlinearity is not a defect but evidence that the model captures authentic learning dynamics.
>
> Moreover, higher-order cognitive development requires sustained accumulation (Bloom’s Taxonomy; Vygotsky’s ZPD) and is unlikely to show dramatic short-term improvement within a brief instructional period. Therefore, temporary decreases or non-increasing cognitive indicators should be interpreted as normal and theoretically consistent phenomena.
>
> To strengthen the clarity and rigor of this concept, we have added a formal definition of “effective learning trajectory” and its educational grounding in **Appendix D.5.1** of the revised manuscript. We sincerely thank the reviewer for prompting us to articulate this more explicitly.
>
> Learn, H. P. (2000). Brain, mind, experience, and school. Committee on Developments in the Science of Learning, 14-15.
>
> Thelen, E., & Smith, L. B. (1994). A dynamic systems approach to the development of cognition and action. MIT press.；Rodrigues, H., Jesús, A., Lamb,
>
> R., Choi, I., & Owens, T. (2023). Unravelling Student Learning: Exploring Nonlinear Dynamics in Science Education. Int. J. Psychol. Neurosci, 9, 118-137.
>
> ---
>
> **Response to Q2: What is the mechanism for memory storage and forgetting, and its alignment with human memory theories**
>
> We thank the reviewer for the thoughtful question. Our memory module draws on the **Atkinson–Shiffrin (1968)** multi-store model and is adapted into a two-tier structure for instructional settings. **Short-term memory** stores high-fidelity information within a session, including teacher inputs, student behaviors, and reflections, supporting in-class processing. After each session, these contents are distilled into **long-term representations** (instruction summaries, updated learning profiles, and regulatory reflections), which guide planning, monitoring, and self-regulation in later sessions.
>
> To simulate the natural human pattern of “detail fading but structure retention,” we adopt **a time-window heuristic**: the system preserves full short-term memory only for the current and immediately preceding session, while older information persists solely in its distilled long-term structural form. This approach avoids the need for complex neural retrieval mechanisms while effectively capturing memory dynamics commonly observed in educational cognition.
>
> Overall, our memory management does not seek biological fidelity at the neural level. Instead, **it builds on established psychological models to implement a mechanism in which detailed information is preserved in the present, structural representations are retained across sessions, and learning evolves gradually over time.** This design enables virtual students to display retention and forgetting patterns that align with widely accepted educational theories.

---

> ### Author Response · Authors · 2025-11-24
> **Official Comment by Authors To Reviewer LQGY [4/4]**
>
> **Response to Q3: How are the emotions being probed (prompt-based vs. Questionnaire-based)**
>
> We thank the reviewer for the thoughtful question regarding our emotion-label generation mechanism. In this work, emotional states are obtained through **a structured, questionnaire-like prompt that simulates human self-reporting**. Specifically:
>
> The system uses structured self-assessment prompts at the end of each task, **guiding the virtual student to select one or more emotions that best match its internal state**. For example:
>
> >“Based on your recent performance and feelings, please select one or more emotions that best describe your current state: positive, confused, negative.”
>
> This mechanism is extensible and grounded in widely adopted practices in multi-agent modeling. **Our design is inspired by emotional self-expression methods used in Generative Agents (Park et al., 2023) and similar systems**, where emotional states reflect the agent’s internal, subjective perception rather than judgments imposed by an external annotator.
>
> ---
>
> **Response to Q4: Whether EduVerse can model problematic student behaviors for intervention studies**
>
> We thank the reviewer for raising this highly valuable and forward-looking question. EduVerse was designed from the outset with this capability in mind, and **we provide operable mechanisms across behavioral, affective, and profile-level modeling to support the generation, expression, and tracking of problem behaviors**.
>
> First, **at the behavioral level**, student agents are equipped with observable off-task behaviors (e.g., sleeping, chatting). These behaviors are automatically triggered under appropriate classroom conditions and fully recorded for downstream analysis. In parallel, the teacher agent can respond with interventions such as “remind” or “stop,” enabling problem behaviors to be generated, monitored, and intervened upon within the system.
>
> Second, **at the affective level**, our structured emotion-inference mechanism (e.g., confused, negative) captures states such as low motivation, frustration, or disengagement. These emotional signals influence the agent’s attention, response style, and interaction patterns, providing realistic dynamic features for studying behavioral or motivational challenges and associated interventions.
>
> Finally, **at the profile modeling level**, the cognitive, motivational, and personality dimensions naturally distinguish “high-engagement students” from “low-motivation or easily distracted students.” As observed in our experiments, the virtual student Zhang Tao repeatedly exhibited side conversations and off-task chatting, which were quantifiable in both the social network structure and behavioral sequences and appropriately triggered teacher interventions.
>
> In summary, EduVerse already supports the generation, visualization, and tracking of problem behaviors, as well as teacher strategies, intervention dynamics, and group-level effects. Empirical evidence can be found in **Section 4.1** (personality-driven stability analysis) and **Appendix D.6.3**. We believe this capability provides an important methodological foundation for future research on classroom management, behavioral intervention, and instructional regulation.

---

### Official Review · Reviewer_Zq4a · 2025-10-30

**Soundness:** 3
**Presentation:** 3
**Contribution:** 3
**Rating:** 4
**Confidence:** 4

**Summary:**

This paper presents EduVerse, a user-defined multi-agent simulation framework for educational scenarios. It introduces a Cognition–Interaction–Evolution (CIE) architecture to simulate realistic classroom dynamics among virtual students and teachers. EduVerse enables customization of agents, environments, and interactions, while supporting human-in-the-loop participation. The authors evaluate the system through simulated and real classroom experiments in Chinese language teaching, showing that EduVerse can reproduce authentic teaching dynamics and capture long-term learning evolution. The platform demonstrates promising potential for educational research, intelligent tutoring, and social learning analysis.

**Strengths:**

Solid theoretical foundation – The CIE framework is conceptually well-motivated and systematically designed, combining cognitive modeling, social interaction, and evolution mechanisms.

Rich and diverse experiments – The authors conduct multiple experiments across different educational aspects (cognitive alignment, group interaction, long-term evolution), providing strong empirical support.

**Weaknesses:**

Unclear system details – The description of the system’s user interface and real-user interaction mechanism (how students and teachers use EduVerse) is vague and underdeveloped.

Limited explanation of real-world experiments – Although the paper claims real classroom validation, the implementation details of these experiments (e.g., how data were collected, how participants interacted) are not clearly stated..

Scalability and generalization – The experiments are confined to a specific subject (Chinese language classes), and the system’s adaptability to other domains remains untested.

**Questions:**

Questions:
1. What does "IRF" mean? This abbreviation appears in abstract without providing any full name before.
2. For the LLM, you mentioned that you use "InternVL" and "GPT-4" (In line 216-219), do you fine-tuning the LLMs via education data to get better results?
3. You mention that "EduVerse provides a human-in-the-loop interface that admits real students or teachers alongside virtual agents" (Line 281-282), how can students and teachers in the real world interact with the system? Does the user interface (UI) is something like the UI of ChatGPT?
4. Do the names appear in the experiment part like "Zhang Jie", "Liu Li" are the names of your simulated student agent or real student name in the real world?
5. Do you have more information about experiments conducted in real world classrooms? It seems that all the experiments in the experiment part are conducted in simulators.

Suggestions:
1. As the author provide so much appendix, I recommend add a table of contents before appendix part.

Typos:
1. In Fig1, ②: Mr. Zhuvividly => Mr. Zhu vividly
2. In Fig1, circle a: Cognition Engin => Cognition Engine

**Details Of Ethics Concerns:**

The paper should explicitly report ethical approval and detailed procedures for user experiments to meet HCI and educational research standards (e.g., IRB review, informed consent, participant demographics, and data handling).

---

> ### Author Response · Authors · 2025-11-24
> **Official Comment by Authors To Reviewer Zq4a [1/3]**
>
> **Response to W1&Q3: Insufficient system details, especially regarding UI and human-agent interaction**
>
> We thank the reviewer for the attention to system-level details. **We would like to clarify that the core contribution of this work lies in proposing EduVerse, a user-defined multi-agent classroom simulation framework, and in validating its effectiveness in modeling teaching behaviors and interaction evolution, rather than delivering a fully engineered educational system**. Nevertheless, we have implemented **a lightweight but functional V1 prototype interface** to support human–agent interaction and visualization of the CIE cognitive chain.
>
> The interface provides the following features:
>
> - A ChatGPT-like input window enabling direct dialogue with multiple virtual students;
> - Real-time updates of each student's Plan–Monitor–Regulate (PMR) states across time steps;
> - Clickable student cards that expand to reveal their cognition–emotion–behavior chains for inspecting internal reasoning;
> - Synchronized visualization of overall classroom interaction trajectories and state changes.
>
> These functionalities demonstrate how the EduVerse framework operates in a multi-agent teaching scenario and allow human users to participate in debugging and validation.
>
> In real experiments, human interact with virtual students through the input window. The system records their inputs in real time and simultaneously updates all students’ PMR chains. The chain texts and behavioral trajectories are saved as experimental data for analyzing classroom interaction structures.
>
> We have added interface illustrations and brief descriptions in **Section 4.2 and Appendix D.4** of the revised manuscript to help reviewers better understand the operational flow and interaction logic of EduVerse. We appreciate the reviewer’s feedback, which helped us improve the clarity of this section.
>
> ---
>
> **Response to W3: Limited generalization due to subject confinement to Chinese language classes**
>
> We thank the reviewer for raising the question regarding “cross-disciplinary generalization.” This concern aligns closely with our consolidated response in Common Concern 2, which includes the full cross-disciplinary experimental design, comparative results between real and simulated Chinese language vs. history classrooms (e.g., replication of subject-specific differences in IRF_rate), and an explanation of the system’s transfer mechanisms. In brief, the CIE framework is a discipline-agnostic modeling structure, and the history-class experiments demonstrate that the system can automatically adapt to the interaction characteristics of different subjects, exhibiting consistent transfer dynamics. Complete experimental results and visual analyses are provided in Common Concern 2.

---

> ### Author Response · Authors · 2025-11-24
> **Official Comment by Authors To Reviewer Zq4a [2/3]**
>
> **Response to W2&Q5: Lack of implementation details for real-world classroom experiments**
>
> We thank the reviewer for the attention to the implementation details of the real classroom experiments. Below, we provide a clear explanation of the data sources, annotation procedures, and the role of real classroom data in our validation design, to facilitate a more intuitive understanding of this part of the study.
>
> **1. Source of Real Classroom Data and IRF Annotation Procedure**
>
> The real classroom data used in our study were obtained from the National Primary and Secondary School Smart Education Platform, **a national-level platform supervised by the Ministry of Education of China and recognized for hosting high-quality lessons taught by exemplary frontline teachers**. From this platform, we selected three Chinese-language lessons that correspond directly to the three text genres used in our experiments (lyrical prose, argumentative essay, and foreign fiction), ensuring that the baseline classrooms are pedagogically well-structured and credible.
>
> To guarantee annotation quality and reproducibility, we adopted **a two-stage “human calibration + AI-assisted” workflow**. A researcher with an educational background first constructed a reference IRF annotation standard based on classical theory, and 100% human agreement was achieved on short calibration segments. We then employed KIMI, a Chinese LLM developed by Moonshot AI, to generate initial automatic annotations, which were compared line by line with human labels.
>
> Through multiple rounds of correction, we established unified IRF decision rules, including boundary definitions for Initiation and Feedback, merging rules for multi-turn probe–F sequences, segmentation strategies for long student responses, and handling of group responses. After stabilizing the annotation paradigm, the full 120-minute lesson set was annotated with AI assistance and manually reviewed segment by segment to ensure consistency and accuracy. Recognizing that teacher discourse styles may introduce ambiguity in IRF boundaries, we adhered strictly to the principle of “human standards first, AI follows” throughout the process.
>
> The revised manuscript (**Section 4.1 and Appendix D3**) now includes the genre and grade information of the real lessons, the complete annotation workflow, key decision rules, and de-identified IRF sample excerpts, ensuring transparency and reproducibility.
>
> **2. Role of Real Classroom Data in the Experiment: Structural Validation**
>
> The purpose of the real classroom data is not to replicate specific content, but to serve as a structural comparator to assess whether the simulated classroom exhibits pedagogically realistic interaction patterns. Details of this comparison are provided in Common Concern 1.
>
> Using IRF structure as the primary comparison index, we find that:
>
> - **Structural consistency**:
> Both simulated and real classrooms exhibit a stable I–R–F interaction cycle, matching typical Chinese-language classroom patterns.
>
> - **Differences are interpretable**:
>   - **Initiation and Response** segments align closely across real and simulated classrooms.
>   - Differences arise mainly in the **Feedback phase**: simulated teachers tend to provide direct evaluative feedback, whereas real teachers more often adopt guiding follow-up questions or emotionally supportive feedback. This reflects a strategy difference, not a structural deviation.
>
> - **Qualitative evidence included**:
> **Appendix D.3** of the revised manuscript now provides side-by-side excerpts from real and simulated classrooms, illustrating both differences and consistencies in interaction chains and expression styles.
>
> These comparisons validate that the proposed framework produces pedagogically interpretable and human-like interaction structures, offering direct evidence of its alignment with real classroom dynamics.
>
> ---
>
> **Response to Q1: Undefined abbreviation “IRF” in the abstract**
>
> We thank the reviewer for pointing out the issue with the IRF abbreviation. We have added the complete definition of IRF in the revised manuscript. This omission was an oversight, and we appreciate your careful review!
>
> ---
>
> **Response to Suggestion & Typros**
>
> We thank the reviewer for the helpful suggestion. We have completed the appendix table of contents and updated Figure 1 in the revised manuscript. This omission was an oversight, and we appreciate your careful review!

---

> ### Author Response · Authors · 2025-11-24
> **Official Comment by Authors To Reviewer Zq4a [3/3]**
>
> **Response to Q2: Whether LLMs (e.g., InternVL, GPT-4) were fine-tuned using educational data**
>
> We thank the reviewer for the question regarding our model training setup. We provide the following clarification:
>
> **1.Base model selection and fine-tuning:**
> We conducted fundamental Chinese-language capability tests (text paraphrasing and reading comprehension, evaluated using accuracy) across several VLMs, including InternVL, LLaVA, Qwen-VL, and MiniCPM. InternVL showed the strongest performance and was therefore selected as the base model. We then applied instruction fine-tuning using our dataset of approximately 6k teacher–student dialogue samples annotated with personality traits, enabling persona-conditioned student modeling.
>
> **2.Use of GPT-4:**
> GPT-4 was not involved in training or any weight updates. It serves solely as a cognitive generator, producing the agents’ cognitive chains and final responses based on scenario settings and dialogue history.
> Thus, in this study, only InternVL was fine-tuned on educational data, while GPT-4 was used exclusively for agent cognition generation. We have added this clarification to **Section 3.2.1** of the main text and **Appendix C.2.3**. We appreciate the reviewer’s careful reading and helpful suggestion.
>
> ---
>
> **Response to Q4: Whether names in experiments refer to real or simulated students**
>
> We thank the reviewer for highlighting the concern regarding student names. **All names used in our experiments are randomly generated virtual identities formed by combining common surnames and given names**; they do not correspond to any contain identifiable information. The names are serving only to make the virtual students’ dialogues appear more natural and aligned with real classroom.
>
> This follows a common practice in multi-agent and virtual population research (e.g., Generative Agents, Park et al., 2023), where **“common but non-referential”** names help maintain natural discourse without compromising ethics. We have clarified this point in Section 3.3 of the revised manuscript, and we appreciate the reviewer’s attention to ethical considerations.
>
> ---
>
> **Response to Ethical Concern**
>
> We thank the reviewer for the thoughtful concerns regarding ethics. We provide the following clarifications on the data sources and usage procedures in this study to prevent any misunderstanding.
>
> **(1) All real classroom data come from nationally released public educational platforms**
>
> The classroom videos and transcripts used in this work are entirely drawn from officially published national-level open educational resources. These lessons are publicly released by educational authorities and do not involve any restricted materials, private information, or non-public sources. The content used in our analysis includes only publicly available instructional dialogue and does not involve student faces or identifiable personal information.
>
> **(2)No new data involving minors were collected**
>
> This study did not conduct any data collection, experiments, interviews, or contact with minors. We did not obtain any additional information from schools, teachers, or students beyond what is already publicly available.
>
> **(3)The majority of the data are model-generated and contain no real personal information**
>
> Aside from the publicly released instructional transcripts, all classroom dialogues, behavior labels, and emotion annotations used in this study were generated by large language models. These generated data do not contain any real student identities or personal information.
>
> **(4)Human–agent interactions were limited to internal technical testing and did not involve external participants**
> - The human–agent interaction component of this study was used solely to verify system functionality and interaction robustness.
> - All interactions were conducted by internal research team members.
> - No personal information about the testers was recorded or analyzed.
> - The purpose of these interactions was system debugging, not the study of human behavior or cognition.
> - The nature of these tests is consistent with standard internal usability testing in software development.
>
> Therefore, these activities do not constitute human-subject research and pose no ethical or safety concerns.
>
> **(5) All data have been anonymized and no raw transcripts will be released**
>
> The publicly available instructional transcripts used for structural analysis are anonymized, will not be published in full, and will not be released externally. All data are used strictly for research purposes, are not used commercially, and contain no identifiable personal information.
>
> In summary, this study adheres strictly to principles of data transparency, anonymization, and minimal risk. It does not involve private data, minors, external participant recruitment, or any form of sensitive human experimentation. We have revised the **Ethics Statement** in the manuscript to ensure transparency and adherence to standard research practices.

---

> > ### Comment · Reviewer_Zq4a · 2025-11-25
> >
> > Thank you for the authors' detailed response, which has helped clarify several aspects of the research. I appreciate the effort put into addressing the reviewers' concerns.
> >
> > However, after carefully considering the rebuttal, I maintain my assessment that the paper is not yet ready for publication. My primary concerns are:
> >
> > 1. Lack of User/Expert Evaluation: The current methodology lacks a rigorous evaluation process involving real users or domain experts. This is a critical gap that undermines the validity of the findings and limits our ability to assess the practical utility of the approach.
> >
> > 2. Writing Quality: The manuscript still requires substantial revision to meet publication standards. The writing needs further refinement to ensure clarity, coherence, and adherence to the expected quality bar for this venue.
> >
> > I encourage the authors to:
> >
> > Incorporate a comprehensive user study or expert evaluation to validate the results
> > Carefully revise the manuscript with attention to structure, clarity, and presentation
> > Consider resubmitting after these substantial improvements have been made
> > I believe this work has potential, but it would benefit significantly from another iteration before being ready for publication.

---

> > > ### Author Response · Authors · 2025-11-26
> > > **Response to Reviewer Zq4a [1/2]**
> > >
> > > We sincerely thank the reviewer for the prompt follow-up and your positive acknowledgment of our previous rebuttal. We are glad that our earlier clarifications have resolved most of your key concerns. Regarding the two remaining points, we further clarify the positioning of our research paradigm, our structured validation pipeline, and the strengthened presentation in the revised manuscript.
> > >
> > > ---
> > >
> > > **Response to concern 1: Lack of User/Expert Evaluation**
> > >
> > > We fully understand the reviewer’s suggestion to strengthen the evaluation perspective. To ensure our contributions are clearly interpreted, we briefly clarify the nature of this work, its validation objectives, and its core contributions.
> > >
> > > **(1) Nature of the work**
> > >
> > > This paper presents a foundational multi-agent classroom simulation framework, different from user-centered system interaction research.
> > >
> > > The core contribution of this work is the construction of:
> > > - a generalizable multi-agent instructional simulation framework,
> > > - capable of generating classroom structures aligned with real pedagogical patterns,
> > > - supporting human-in-the-loop multi-agent interaction and cross-session evolution.
> > >
> > > Evaluations for such a framework-level contribution focus on examining:
> > >
> > > - the rigor of classroom interaction structures (IRF, BEC, interaction graphs),
> > > - the interpretability of agent behavior mechanisms,
> > > - the stability of multi-agent interaction dynamics, and
> > > - the framework’s generalization across subjects, layouts, and instructional sessions.
> > >
> > > These aspects constitute the core criteria for validating the credibility of a simulation framework.
> > >
> > > In contrast, user studies for real classroom deployment belong to applied-system research in later stages, not the central contribution of a foundational framework.
> > >
> > > In other words, our goal is to build a framework worth conducting user studies on, not to directly evaluate a teaching-support system.
> > >
> > > This positioning is fully aligned with the proper evaluation logic for foundational simulation research.
> > >
> > > **(2) Validation objectives**
> > >
> > > **Our structured expert-in-the-loop and empirical validation pipeline already covers the most essential evaluation needs for framework-level research.**
> > >
> > > To ensure scientific rigor and credibility, we designed a systematic and reproducible validation chain, including:
> > > - expert-based IRF annotation and alignment between real and simulated classrooms (see Common Concern 1);
> > > - quantitative structure verification using IRF/BEC metrics and group interaction graphs;
> > > - multi-context generalization tests across subjects, classroom layouts, and instructional sessions;
> > > - a complete human-agent interaction loop through the simulation interface.
> > >
> > > These validation strategies combine theoretical grounding, quantitative reproducibility, and cross-scenario consistency—representing the gold standard for establishing the reliability of a simulation framework.
> > >
> > > Accordingly, from the perspective of evaluating a foundational framework, the depth and completeness of our validation already match or exceed comparable works (see Appendix Table A1).
> > >
> > > **(3) Core contribution**
> > >
> > > The overarching goal of our framework is to provide a transparent, stable, and controllable infrastructure for future instructional interaction studies.
> > >
> > > As demonstrated in Section 4.1 (Human–Agent Interaction), the current system already supports:
> > > - a full human–agent interaction loop,
> > > - controllable teaching-phase scheduling,
> > > - stable multi-agent behavioral dynamics,
> > > - explicit cross-session evolution structures.
> > >
> > > These properties ensure that future user studies will not only be feasible, but also more interpretable and analytically grounded.

---

> > > ### Author Response · Authors · 2025-11-26
> > > **Response to Reviewer Zq4a [2/2]**
> > >
> > > **Response to concern 2: Writing Quality**
> > >
> > > Thank you for your recommendation to further improve the clarity of the presentation. In the revised manuscript, we have conducted a systematic, comprehensive refinement. All updated parts are marked in blue for easy reference. We kindly invite you to review the revised version to more accurately evaluate our improvements, which include:
> > >
> > > - Reorganized the structure of the Introduction to make the research motivation more focused and coherent (Sec. 1).
> > > - Reworked the modular section titles of the framework to ensure clearer boundaries and stronger alignment with the main figure (Sec. 3).
> > > - Standardizing terminology and adding illustrative figures to enhance readability
> > > - Strengthening the logical flow and transitions within each experimental subsection (Sec. 4)
> > > - Refining overall writing for conciseness and stylistic consistency throughout the manuscript
> > >
> > > These adjustments substantially improve the transparency of the structure and the smoothness of information flow.
> > >
> > > We sincerely appreciate the reviewer’s suggestions, which have motivated us to further enhance the presentation quality. We hope that these revisions significantly improve the overall clarity and readability of the paper, and more accurately convey the research contributions of our work.
> > >
> > > ---
> > >
> > > We hope the above clarifications sufficiently address your concerns, and we sincerely appreciate the reviewer’s thoughtful analysis and constructive feedback from multiple perspectives. Across both review rounds, we deeply value your expertise in HCI and AI-in-education, which has been instrumental in helping us further refine the positioning, methodology, and interpretations of our work. We also truly appreciate the interdisciplinary suggestions and have incorporated clarifications wherever possible within the scope of our current experiments. We look forward to future opportunities for continued discussion and collaboration, and sincerely thank you again for your insightful contributions.

---

### Official Review · Reviewer_2hzb · 2025-10-31

**Soundness:** 3
**Presentation:** 3
**Contribution:** 4
**Rating:** 4
**Confidence:** 5

**Summary:**

This paper presents a framework, EduVerse, for user-defined multi-agent simulation in the context of AI in education. The authors deployed  EduVerse in middle school Chinese language classes with diverse educational  tasks, rich emotional expression, and complex interaction structures. The authors also conducted empirical experiment with existing frameworks.

**Strengths:**

- Timely topic focusing on the AI in education and LLM
- In-depth analysis of related work
- The proposed framework  combines the cognitive, interactive, and  evolutionary dynamics of developmental agents in the context of AI in education
- Deployed in classrooms showcases the practical impact
- Human-in-the-loop interface allows real teachers and students  to enabling simulation, causal testing, and validation

**Weaknesses:**

For designing an intelligent tutoring system, it is crucial to take into account the subject domain. For example, students cognitive, help seeking behavior, and peer discussion vary widely across math vs writing an essay in literature vs introductory programming.

The prior work by other researchers cited by the authors are also domain specific. Would the authors say how to incorporate the framework for a specific subject domain with different question difficulties and knowledge base?

**Questions:**

Please see weakness

---

> ### Author Response · Authors · 2025-11-24
> **Official Comment by Authors To Reviewer 2hzb**
>
> **Response to W1 & W2: Subject Adaptability and Modeling of Cross-Disciplinary Differences**
>
> We thank the reviewers for raising these important questions. We have provided a systematic discussion of EduVerse’s cross-disciplinary generalization capability in Common Concern 2, and we offer a focused response here.
>
> **1.Framework-Level Cross-Disciplinary Generality and Adaptation Mechanisms**
>
> The CIE framework of EduVerse is deliberately designed to avoid binding any subject-specific knowledge. Instead, it is built upon three pedagogical elements that can be reused across subjects:
>
> - **Teacher behaviors** (questioning / explanation / feedback),
> - **Instructional phases** (introduction – explanation – consolidation – exercise – summary),
> - **Question types** (open-ended / closed-ended).
>
> These elements, combined with the student profile dimensions of **personality–motivation–cognition style**, form a stable cognitive decision pathway. Therefore, the same modeling structure and decision mechanisms remain applicable across different subject domains, no additional adjustments are required.
>
> In implementation, we adopt a **“lesson-plan–driven + instructional phase × question type”** design, enabling the teacher agent to automatically adjust pacing based on subject content. Because student profiles are decoupled from subject matter, students with the same personality traits exhibit consistent behavioral tendencies across disciplines. **This mechanism maintains structural stability while allowing natural differentiation in interaction density, knowledge progression speed, and social style across subjects**.
>
> **2.Experimental Cross-Disciplinary Validation: Chinese Language vs. History**
>
> To verify that the above mechanism holds in practice, we added simulation experiments for the **World History subject**. Under the condition that **only the lesson plan is replaced, without modifying any framework logic**, the model exhibits subject-specific behavioral differences consistent with real classrooms. Key findings include:
>
> - **Real classroom patterns**:
> Chinese language classes display denser IRF (Initiation–Response–Feedback) structures (IRF_rate ≈ 0.423), while history classes are more lecture-oriented with a lower IRF_rate (≈ 0.333).
>
> |      Subjects     |   I   |  R  |   F   | IRF_rate |
> |:-----------------:|:-----:|:---:|:-----:|:--------:|
> | Chinese_realclass | 0.432 | --- | 0.600 |   0.423  |
> | History_realclass | 0.359 | --- | 0.513 |   0.333  |
>
> - **Simulated classroom patterns (without changing any model logic)**:
> After replacing only the instructional input and lesson plan (chinese → history), the IRF interaction frequency of simulated students decreases significantly in the history class, mirroring real-world trends.
>
> |      Subjects      |   I   |   R   |   F   | IRF_rate |
> |:------------------:|:-----:|:-----:|:-----:|:--------:|
> | Chinese_simulation | 0.416 | 0.201 | 0.345 |   0.423  |
> | History_simulation | 0.201 | 0.189 | 0.376 |   0.226  |
>
> - **Cross-disciplinary stability**:
> Despite frequency differences, both subjects consistently maintain the core interaction pattern **teacher initiation → student response → teacher feedback**, demonstrating robust structural transfer across disciplines.
>
> | Environment |   I   |   R   |   F   | IRF_rate |
> |:-----------:|:-----:|:-----:|:-----:|:--------:|
> |   Lecture   | 0.205 | 0.269 | 0.385 |   0.282  |
> |    Collab   | 0.256 | 0.226 | 0.415 |   0.282  |
> |    Round    | 0.143 | 0.071 | 0.329 |   0.114  |
>
> - **Social network differentiation**:
> Visualization of student social graphs reveals substantial differences in social activity and centrality between literature and history classes (See **Appendix D.6.2&D6.3**), indicating subject-specific behavioral divergence after cross-disciplinary adaptation.
>
> In summary, **EduVerse demonstrates stable cross-disciplinary adaptability at both the framework level and the experimental performance level**. Full data trends, IRF excerpts, and visualization results are presented in Common Concern 2, with additional details included in **Section 4.4** and **Appendix D.6** of the revised manuscript.

---

### Official Review · Reviewer_pnJm · 2025-11-01

**Soundness:** 1
**Presentation:** 2
**Contribution:** 2
**Rating:** 2
**Confidence:** 3

**Summary:**

The authors introduce EduVerse, a “user‑defined multi‑agent simulation space” for virtual classrooms built around a Cognition–Interaction–Evolution (CIE) architecture layered over a Perception–Cognition–Action loop. Users can customize the seat graph/layouts, teacher/student agents,, and sessions (multi‑lesson trajectories). A human‑in‑the‑loop interface lets real users join a simulated class. Figure 1 lays out the three components, user‑defined environment, CIE agent modeling, and interaction/evolution experiments.
The authors’ core claim is the simulated instructional realism of a typical classroom (measured by IRF rates).

**Strengths:**

I think this is a very interesting idea with a good approach but I have alot of reservations about the claims made by the authors.

Focusing on the positives, I think the work done itself is good. There are plenty of good uses for a simulator of this type, especially ones that involve a human in the loop.

I do like the modular CIE breakdown, the explicit teacher pacing controller, and that the tasks are already implemented. The range of evaluation criteria is good, even if I have some concerns about them. IRF alignment, B/E/C distributions, small‑graph network summaries, ablations, human–agent tasks, and a cross‑session measure.

My favorite part is probably the CIE-based agent modeling. I think there’s alot of potential in the ideas that the authors outlined here with how the process of teacher-led group discussion can play out.

**Weaknesses:**

While there’s alot to like about this paper, I think there are some pretty severe issues with the main claim:

- The authors position EduVerse as the “first” user‑defined multi‑agent classroom simulator. But they even acknowledge other pre-existing multi-agent class room simulators in their own related work, and other general agent set ups (ie, AgentVerse) that already support role‑based, IRF‑style interactions.

- The Abstract and Table 1 frame IRF rates as “close” to real classes, but Table A4 shows sizeable divergences (e.g., Argumentative Essay, Lecture: 0.639 vs. 0.417 real). ESPECIALLY with a signal as noisy as teacher-led discussion in classes, I feel like it's hard to take any purely quantitative analyses at face value without some kind of qualitative evidence to back it up.
- There’s a lack of details about how many classes/schools were used as the comparison baseline and, again, who annotated the logs who could provide qualitative evidence as backup.
- The system labels its own cognition (Bloom) and emotion during the Monitor step, then reuses these labels for evaluation (BEC distributions). If im not misunderstanding, this is basically just the model asking itself if it's correct, which doesnt seem super reliable.
- The authors  fine‑tune VLM backbones (InternVL/LLaVA/Qwen‑VL/MiniCPM) for text‑only style, trained on ~6k utterances. Why VLMs for text style control? The authors also report InternVL “achieved the highest scenario‑grounded performance,” but the metric and protocol aren’t shown.
- Not a huge negative but a heads up, for Figure 1, the middle section has “Cognition Engin” instead of “Cognition Engine”

**Questions:**

- Did the authors inspect the generated EduVerse logs vs the conversation logs of a real classroom?
- Were all experiments/baselines drawn from the same classroom or different classrooms?
- What was the motivation for using VLM backbones for what seems, to me, a largely text-based scenario?

**Details Of Ethics Concerns:**

I might have missed it but my concern is primarily with the participants in the 'Chinese language middle school classrooms'. I don't believe the authors mentioned any IRB or consent process for collecting data from these students? I'm not sure if this is a significant issue but I figured I should flag it to be safe.

---

> ### Author Response · Authors · 2025-11-24
> **Official Comment by Authors To Reviewer pnJm [1/3]**
>
> **Response to W1: Overstatement of the “first” user‑defined multi-agent classroom simulator claim**
>
> We thank the reviewer for the thoughtful reminder regarding the use of the term “first.” We fully agree that such wording must be used with great caution. **Accordingly, during manuscript preparation we conducted a systematic review of representative work** in educational simulation and multi-agent systems over the past three years, including AgentVerse, Generative Agents, Simulating Classroom, MathVC, GPTeach, among others, and we discussed and cited them explicitly in the paper.
>
> To avoid ambiguity, **we have replaced "the first" with "one of the first" throughout the manuscript**. Our claim does not concern multi-agent educational platforms broadly, but specifically **“user-defined classroom simulators”** integrating environment customization, agent customization, and longitudinal evolution. **EduVerse is, to the best of our knowledge, the first system to simultaneously integrate three key capabilities within a unified framework**:
>
> - **Customizable classroom environment**:
> EduVerse enables users to freely edit a full classroom-level space, including blackboard, podium, seating layout, and various classroom organizations (lecture, collaborative, round-table, etc.). While some frameworks support multi-agent environment construction, they typically target general-purpose tasks rather than structured instructional settings; existing educational simulators rarely allow users to directly define the physical classroom layout.
>
> - **Customizable agent modeling**:
> EduVerse allows users to define both student and teacher profiles, decomposed into personality traits, motivation, cognitive styles, and PMR-based metacognitive processes. This profile → cognitive chain → behavioral generation pipeline is user-configurable—an ability largely absent in prior systems, which often provide only preset roles or superficial parameter controls rather than full educational profile configurability.
>
> - **Multi-session instructional evolution**:
> EduVerse supports multi-session continuous simulation, where the teacher agent can dynamically advance or adjust teaching phases based on student states, and students exhibit observable cognitive and behavioral evolution across sessions. Most existing multi-agent systems operate at the level of a single class or single interaction round and lack a user-controllable longitudinal evolution mechanism.
>
> Because these three capabilities have not been offered together within a single framework in prior work, we use the term “first” to characterize EduVerse’s originality in this specific direction. **To make this distinction clearer, we have added a concise comparison summary in the revised manuscript (Table A1 in Appendix B) and refined the phrasing to make it more precise and cautious**:
>
> >"one of the first user-defined multi-agent classroom simulator supporting customizable environment, customizable agents, and multi-session evolution."
>
> We hope this clarification accurately conveys EduVerse' s unique positioning and resolves any possible misunderstanding.
>
> ---
>
> **Response to W2&Q1: IRF discrepancies between simulation and real classrooms**
>
> We thank the reviewer for the insightful comments. The numerical differences between Table 1 and Table A4 arise from their distinct analytic purposes: Table 1 summarizes macro-level averages across environments and genres, whereas Table A4 reports fine-grained scenario-specific statistics that naturally vary with teacher style and classroom conditions. This variability is theoretically expected and does not indicate inconsistency. We have provided a systematic explanation in Common Concern 1 regarding the theoretical basis of IRF variability, the influence of teacher style, and the notion of “acceptable variation” in educational research. We kindly invite the reviewer to refer to that section for the complete discussion.
>
> In addition, we have added **qualitative dialogue examples** in the revised manuscript and explicitly defined the criteria for identifying the IRF stages (i.e., boundary definitions and usage rules for Initiation / Response / Feedback). We observe that **real and simulated classrooms share highly consistent textual structures in the I and R phases**. The primary source of differences lies in the F phase: real teachers tend to use guiding follow-up questions, whereas simulated teachers more often provide direct feedback. This stylistic difference is theoretically explainable and does not undermine the assessment of structural consistency.
>
> We apologize for the confusion. Following the reviewer' s suggestion, we have now added relevant tables (see **Appendix Table A8**). We hope these additions make the evaluation process clearer and help reviewers quickly understand our argument regarding classroom structure alignment.

---

> ### Author Response · Authors · 2025-11-24
> **Official Comment by Authors To Reviewer pnJm [2/3]**
>
> **Response to W3&Q2: Insufficient details about baseline classrooms and annotation process**
>
> We thank the reviewer for the insightful comments on the source of real classroom data and the IRF annotation procedure. To address these concerns, we have added clearer descriptions and full documentation in the revised manuscript.
>
> The real classroom data used in our study were obtained from the National Primary and Secondary School Smart Education Platform, **a national-level platform supervised by the Ministry of Education of China and recognized for hosting high-quality lessons taught by exemplary frontline teachers**. From this platform, we selected three Chinese-language lessons that **correspond directly to the three text genres used in our experiments (lyrical prose, argumentative essay, and foreign fiction)**, ensuring that the baseline classrooms are pedagogically well-structured and credible.
>
> To guarantee annotation quality and reproducibility, we adopted **a two-stage “human calibration + AI-assisted” workflow**. A researcher with an educational background first constructed a reference IRF annotation standard based on classical theory, and 100% human agreement was achieved on short calibration segments. We then employed KIMI, a Chinese LLM developed by Moonshot AI, to generate initial automatic annotations, which were compared line by line with human labels.
>
> Through multiple rounds of correction, we established unified IRF decision rules, including boundary definitions for Initiation and Feedback, merging rules for multi-turn probe–F sequences, segmentation strategies for long student responses, and handling of group responses. After stabilizing the annotation paradigm, the full 120-minute lesson set was annotated with AI assistance and manually reviewed segment by segment to ensure consistency and accuracy. Recognizing that teacher discourse styles may introduce ambiguity in IRF boundaries, we adhered strictly to the principle of “human standards first, AI follows” throughout the process.
>
> The revised manuscript (**Section 4.1 and Appendix D3.2**) now includes the genre and grade information of the real lessons, the complete annotation workflow, key decision rules, and de-identified IRF sample excerpts, ensuring transparency and reproducibility.
>
> ---
>
> **Response to W4: Concerns about using self-generated cognition/emotion labels for evaluation**
>
> We sincerely thank the reviewer for the thoughtful comments regarding the use of BEC (Behavior–Emotion–Cognition) labels in the Monitor stage. **We would like to clarify that the design does not constitute a “self-evaluation loop,” for the following reasons**.
>
> First, **this task is not a traditional “prediction accuracy evaluation” task, but rather a subjective behavioral trajectory modeling task**. The goal of our work is for virtual students to express their self-perceived learning experience shaped by their personality and situational context, rather than to generate labels that are subsequently used to evaluate model performance. Therefore, the BEC labels produced in the Monitor stage function as an explicit representation of the **agent’s internal state**, not as an **external metric for assessing model quality**.
>
> Second, **this design choice has a clear methodological foundation**. The BEC structure aligns with well-established constructs in educational psychology, including self-report measures and metacognitive theory (Plan–Monitor–Regulate), **both of which emphasize how learners perceive their own current learning state rather than how they objectively perform on external tasks**. Similar self-state–writing mechanisms are also widely used in multi-agent systems. For instance, in Generative Agents (Park et al., 2023), agents likewise record their internal emotions, motivations, and behavioral reflections to support world modeling and long-term evolution. **Our use of BEC as a form of agent self-expression follows this established paradigm**.
>
> In summary, BEC labels in our study should be understood as a quantified representation of the virtual student’s subjective awareness, functionally analogous to self-report data rather than an external performance metric. We will further emphasize this distinction in **Section 4.1 and Appendix D2.2** of the revised manuscript to prevent potential misunderstanding.

---

> ### Author Response · Authors · 2025-11-24
> **Official Comment by Authors To Reviewer pnJm [3/3]**
>
> **Reponse to W5&Q3: Unclear rationale for using VLMs and missing evaluation protocol for InternVL.**
>
> We thank the reviewer for raising this question. The core task of our study is classroom dialogue generation. Although the selected model possesses visual capabilities, we use only its language-generation module in all experiments; thus, no visual dependency is involved. **We chose a vision-language model (VLM) primarily to ensure future compatibility when EduVerse is extended to multimodal classroom scenarios involving images, instructional materials, and visual cues**.
>
> Regarding the evaluation protocol for InternVL, we provide two categories of preliminary assessments conducted to ensure the model’s suitability for the tasks in this study:
>
> - **Evaluation of Chinese language competence (before fine-tuning)**: We evaluated candidate models on two categories of Chinese-language secondary-school tasks—text paraphrasing and text comprehension—using **accuracy** as the metric. InternVL demonstrated the strongest performance across both tasks, making it the most appropriate base model for Chinese classroom simulation.
>
> - **Evaluation of persona fine-tuning effectiveness (after fine-tuning)**: After performing stylistic fine-tuning on the unified virtual-student dataset, we compared models using human subjective ratings and GPT-4 large-scale consistency evaluation to assess persona coherence and stylistic expressiveness. InternVL outperformed all other candidates in both stability and controllability.
>
> Based on these evaluations, we selected InternVL as the primary model for EduVerse. We have added the corresponding assessment tasks to **Appendix C.2.3** to enhance transparency. We appreciate the reviewer’s comment and have strengthened the explanation in the revised manuscript.
>
> ---
>
> **Reponse to W6: Typographical error in Figure 1 (“Cognition Engin”)**
>
> We thank the reviewer for pointing out the spelling error. **We have corrected it to “Cognition Engine” in the revised manuscript**. This was an oversight on our part, and we appreciate your careful reading!
>
> ---
>
> **Reponse to Ethical concerns**
>
> We thank the reviewer for the thoughtful concerns regarding ethics. We provide the following clarifications on the data sources and usage procedures in this study to prevent any misunderstanding.
>
> **(1) All real classroom data come from nationally released public educational platforms**
>
> The classroom videos and transcripts used in this work are entirely drawn from officially published national-level open educational resources. These lessons are publicly released by educational authorities and do not involve any restricted materials, private information, or non-public sources. The content used in our analysis includes only publicly available instructional dialogue and does not involve identifiable personal information.
>
> **(2)No new data involving minors were collected**
>
> This study did not conduct any data collection, experiments, interviews, or contact with minors. We did not obtain any additional information from schools, teachers, or students beyond what is already publicly available.
>
> **(3)The majority of the data are model-generated and contain no real personal information**
>
> Aside from the publicly released instructional transcripts, all classroom dialogues, behavior labels, and emotion annotations used in this study were generated by large language models. These generated data do not contain any real student identities or personal information.
>
> **(4)Human–agent interactions were limited to internal technical testing and did not involve external participants**
>
> - The human–agent interaction component of this study was used solely to verify system functionality and interaction robustness.
> - All interactions were conducted by internal research team members.
> - No personal information about the testers was recorded or analyzed.
> - The purpose of these interactions was system debugging, not the study of human behavior or cognition.
> - The nature of these tests is consistent with standard internal usability testing in software development.
>
> Therefore, these activities do not constitute human-subject research and pose no ethical or safety concerns.
>
> **(5) All data have been anonymized and no raw transcripts will be released**
>
> The publicly available instructional transcripts used for structural analysis are anonymized, will not be published in full, and will not be released externally. All data are used strictly for research purposes, are not used commercially, and contain no identifiable personal information.
>
> In summary, this study adheres strictly to principles of data transparency, anonymization, and minimal risk. It does not involve private data, minors, external participant recruitment, or any form of sensitive human experimentation. We have revised the **Ethics Statement** in the manuscript to ensure transparency, rigor, and adherence to standard research practices.

---

> ### Comment · Reviewer_pnJm · 2025-11-24
> **Quick response: Acknowledgment of Rebuttal. Will go through revised manuscript and reconsider score.**
>
> I thank the reviewers for their comprehensive and thorough responses to my review.
>
> I will make sure to review the updated manuscript to make sure it is fresh in my context, reach out if there are any further questions, and adjust my score as needed.
>
> edit: That first 'reviewers' should be 'authors'. Will need to make sure I get some sleep before I revisit the revised paper.

---

> > ### Author Response · Authors · 2025-11-24
> > **Thank you for the quick response**
> >
> > Dear Reviewer pnJm,
> >
> > Thank you for your prompt response and for taking the time to revisit our revised manuscript. The updated version has been uploaded. Please feel free to let us know if any further clarification is needed. We truly appreciate your effort and consideration.
> >
> > Sincerely,
> >
> > The Authors

---

### Official Review · Reviewer_jPWs · 2025-11-03

**Soundness:** 2
**Presentation:** 2
**Contribution:** 2
**Rating:** 2
**Confidence:** 4

**Summary:**

The paper focuses on reproducing realistic classroom dynamics. To achieve this, the authors present EduVerse, a novel user-defined multi-agent simulation platform that introduces a Cognition–Interaction–Evolution (CIE) architecture. This architecture models the long-term cognitive, emotional, and behavioral development of virtual agents within customizable classroom environments.

**Strengths:**

Human–Agent Interaction provides valuable insights through experimental studies.

**Weaknesses:**

- The work attempts to address multiple aspects, including individual modeling, role-differentiated social interaction, and longitudinal instructional adaptation. But does not clearly explain them.

- The evaluation is vague.
  - Please include the key metrics in the main paper instead of the appendix. This would improve both readers’ understanding and reviewers’ efficiency.
  - Table 1 does not show how the simulation aligns with real classroom data. For example, IRF_rate on Lyrical Prose (0.336 vs. 0.486) contradicts the claim of only minor genre-specific variations.
  - Figure 5 lacks a clear caption about the ablation study, making it difficult to follow the analysis and interpret the bar chart.
  - Much of the analysis focuses on individual cases, while the main focus should be at the class level.

- The work is limited to Chinese language classes. Cross-domain or cross-linguistic experiments would strengthen the generalization of this work.

- Figure 4 is too small and hard to review.

**Questions:**

Please see weaknesses.

---

> ### Author Response · Authors · 2025-11-24
> **Official Comment by Authors To Reviewer jPWs [1/2]**
>
> **Response to W1: Insufficient clarity in explaining multiple modeling aspects**
>
> We thank the reviewer for the thoughtful comments regarding the multi-level modeling design. We fully understand that when several cross-module mechanisms appear together, readers may benefit from a more integrated and centralized explanation. Below, we provide a concise clarification of how the three modeling components are positioned within the overall framework, and we have strengthened the structural presentation accordingly in the revised manuscript.
>
> - **Individual Modeling (Individual Modeling → Cognition) maps to the Cognition layer of the CIE framework.**
> It captures learner differences through personality traits, motivation, and cognitive styles, and organizes learning behaviors via the PMR (Plan–Monitor–Regulate) structure. Its consistency and effectiveness are validated in Experiment I.
>
> - **Role-Differentiated Social Interaction (Role-Differentiated Interaction → Interaction) corresponds to the Interaction layer.**
> It models students’ interaction-role preferences through a willingness function and an extended IRF (Initiate–Respond–Feedback–Regulate) mechanism. The resulting social-role structures are reflected in the social network and influence analyses in Experiment II.
>
> - **Longitudinal Teaching Adaptation (Longitudinal Adaptation → Evolution) maps to the Evolution layer.**
> It models cross-session behavioral and cognitive changes through instructional-phase progression mechanisms and memory updates. Experiment III validates the stability of its longitudinal behavioral–cognitive trajectories.
>
> To further improve readability, **we added the following clarifications in the revised manuscript**:
>
> - A framework-positioning sentence at the beginning of **Section 3.2**;
> - Explicit one-to-one alignment markers between modeling modules and experiments (**Section 4.1, Section 4.2, Section 4.3**).
>
> These additions ensure that readers can clearly understand the modeling logic of EduVerse without needing to integrate information across multiple sections.

---

> ### Author Response · Authors · 2025-11-24
> **Official Comment by Authors To Reviewer jPWs [2/2]**
>
> **Response to W2&W4: Evaluation procedure is vague, Figure 4 is too small and difficult to read**
>
> We sincerely thank the reviewer for the valuable suggestions regarding our evaluation procedures and visualizations. Based on your feedback, **we have made focused revisions and enhancements to the key sections of the manuscript**. The major updates are as follows:
>
> - **Core evaluation metrics have been moved from the appendix to the main text.**
> To improve readability, we relocated key metrics, including IRF ratios, B/E/C distributions, group network measures, and positive-transition statistics, into **Section 4**. Readers can now access essential results without navigating across sections.
>
> - **Clarification on differences in IRF rates across text genres**.
> The goal of IRF analysis is to **compare structural trends rather than pursue identical numerical values**. Prior educational studies have repeatedly shown that IRF proportions vary due to teacher style, instructional goals, and textual characteristics; such differences are expected and theoretically grounded. **We removed misleading phrasing such as “minor difference” and added explanations of the sources and interpretability of these variations in the main text Section 4.1**. A more systematic discussion can be found in Common Concern 1.
>
> - **Revised captions for the ablation results in Figure 5.**
> We rewrote the figure caption to clearly explain the logic behind each ablation setting (removing PMR, removing stylistic modules, full framework) and explicitly marked the experimental condition represented by each bar. This makes the ablation path and resulting differences more intuitive.
>
> - **Improved layout and clarity for Figure 4.**
> We enlarged the figure, strengthened label visibility, and expanded the caption with additional explanations of personality categories and phase-level interaction structures, making the visual patterns easier to interpret.
>
> - **Regarding the concern that “analysis should focus on the class level.”**
> Our evaluation has consistently addressed both class-level structures and individual-level differences, presented explicitly across all three experiments:
>
>   - Experiment I analyzes class-level IRF/BEC structure and stability, complemented by individual-level consistency checks.
>
>   - Experiment II models class social structure via network density and centrality, while individual influence scores capture heterogeneous roles.
>
>   - Experiment III examines class-level trends through session-level evolution and supplements this with individual longitudinal trajectories.
>
> In sum, our experimental design follows **a principled micro-mechanism → macro-structure logic** and remains fully aligned with the class-level focus.
>
> ---
> **Response to W3: Limited subject scope (only Chinese language classes)**
>
> We thank the reviewer for raising the question regarding the scope of subject domains. We fully agree that **cross-disciplinary and cross-lingual experiments are important for validating the generalization capability of the system**, and we have provided a detailed response in Common Concern 2. Here, we offer additional key clarifications:
>
> - **On cross-lingual capability**
> Our current experiments are conducted in Chinese classrooms primarily because the stylized fine-tuned model used in this work was trained in a Chinese linguistic context. To ensure consistency across experiments, we therefore adopted Chinese classroom scenarios. However, **the framework itself is language-agnostic**. By replacing the fine-tuned model with an English counterpart, the system can directly support cross-lingual classroom simulation without any modification to the underlying architecture.
>
> - **On cross-disciplinary capability**
> To demonstrate that the model does not rely on characteristics specific to Chinese language instruction, we added a new experiment on the history subject, **World History (The Renaissance)**, in the revised manuscript (see **Section 4.4**). History classrooms differ markedly from language classrooms in knowledge structures, questioning patterns, and interaction density. Yet, the model successfully reproduced structural trends observed in real history classrooms (e.g., lower IRF density, stronger logical sequencing and temporal structure), indicating that EduVerse maintains stable structural transfer across subjects.
>
> In summary, **the focus on Chinese language classrooms reflects an experimental choice, not a methodological limitation.** The newly added history-class experiment further supports EduVerse' s potential for cross-lingual, cross-domain, and cross-pedagogical transfer.

---

### Author Response · Authors · 2025-11-24
**Common Concern 2: On the Generalization and Cross-Disciplinary Adaptability of EduVerse**

We thank **Reviewer jPWs**, **Reviewer 2hzb**, and **Reviewer Zq4a** for their thoughtful comments regarding EduVerse' s cross-disciplinary generalization capability. **EduVerse was designed from the outset with generality in mind, its framework architecture, student modeling logic, and behavior generation mechanisms are all naturally transferable across subject domains.** In the revised manuscript, we have added cross-disciplinary experiments and corresponding analyses. Our responses are as follows:

**1. The CIE framework provides discipline-agnostic structural generality.**

EduVerse adopts the Cognition–Interaction–Evolution (CIE) modeling framework, in which all instructional processes are driven by general variables such as teaching phases, question types, and student profiles, **without relying on any subject-specific rules**. Personality traits, motivation, and cognitive styles in the student model are decoupled from instructional content, meaning that switching subjects requires no change to the system architecture, **only the lesson plan needs to be replaced**. This abstraction ensures natural transferability across disciplines.

**2. Chinese language classrooms represent a high-complexity setting that tests the system' s upper capability bound.**

We selected Chinese language classes as the primary evaluation environment because they involve high interaction density, strong emotional engagement, rich question types, and complex cognitive chains. **These characteristics make language classrooms one of the most challenging domains for structural interaction modeling.** The system' s ability to stably generate structured interaction patterns in this setting already provides a strong foundation for cross-disciplinary transfer.

**3. Cross-disciplinary experiments reveal clear and consistent transfer dynamics.**

- **Real classrooms exhibit inherent subject differences:**
Language classes involve frequent questioning and high IRF density, whereas history classes emphasize logical explanation and exhibit sparser IRF structures. In real classroom observations, **IRF_rate is 0.423 for Chinese and 0.333 for history, reflecting intrinsic domain differences**.

|      **Subjects**     |   **I**   | **R** |   **F**   | **IRF_rate** |
| :-----------------:  | :-----: | :-: | :-----: | :--------: |
| Chinese_realclass | 0.432 | --- | 0.600 |   **0.423**  |
| History_realclass | 0.359 | --- | 0.513 |   **0.333**  |

- **Simulated classrooms successfully reproduce these trends:**
When only the subject content is changed, **the simulated history classroom shows a significantly lower IRF_rate than the simulated language classroom (0.423 vs. 0.226)**, mirroring the trend observed in real data. This indicates that the model captures structural subject-level transfer rather than merely memorizing patterns.

|      **Subjects**      |   **I**   |   **R**   |   **F**   | **IRF_rate** |
|:------------------:|:-----:|:-----:|:-----:|:--------:|
| Chinese_simulation | 0.416 | 0.201 | 0.345 |   **0.423**  |
| History_simulation | 0.201 | 0.189 | 0.376 |   **0.226**  |

- **IRF structural patterns remain stable across subjects and classroom layouts:**
Despite differing frequencies, the fundamental interaction pattern“teacher initiation → student response → teacher feedback”is consistently generated in both history and language classes across lecture, collaborative, and round-table settings. The structure remains stable even when the surface conditions change.

| **Environment** |   **I**   |   **R**   |   **F**   | **IRF_rate** |
|:-----------:|:-----:|:-----:|:-----:|:--------:|
|   Lecture   | 0.205 | 0.269 | 0.385 |   0.282  |
|    Collab   | 0.256 | 0.226 | 0.415 |   0.282  |
|    Round    | 0.143 | 0.071 | 0.329 |   0.114  |

- **Subject-specific divergence is also observable in student social networks**:
In history classrooms, student centrality and interaction density are lower than in language classrooms, indicating that after transfer, the system generates social behaviors aligned with domain characteristics rather than replicating the same interaction style across subjects (see in **Appendix D.6.3**).

In summary, EduVerse demonstrates clear cross-disciplinary adaptability and stable transferability across the theoretical framework, student modeling design, and empirical validation. We have added key results from the history-class experiments in the revised manuscript (**see Experiment IV, Appendix D.6**). We again thank the reviewers for their insightful comments and recognition of the system’s extensibility.

---

### Author Response · Authors · 2025-11-24
**Common Concern 1: Clarifications and Additions Regarding the Comparison Between Virtual and Real Classrooms**

We thank **Reviewer jPWs**, **Reviewer pnJm**, and **Reviewer Zq4a** for their thoughtful questions on the comparison between real and simulated classrooms. As this component directly supports our core contribution of reconstructing authentic classroom interaction structures, we have substantially expanded the revised manuscript to clarify analysis objectives, data sources, annotation workflow, and example evidence, thereby making the validation chain more transparent and traceable.

**1. IRF ratios are used to examine structural consistency, not numerical similarity.**

In educational research, IRF (Initiation–Response–Feedback) is a structural pattern, and its distribution varies significantly across subjects, teacher styles, and instructional goals; therefore, no “standard percentage” exists to be reproduced.

For example: Zaswita (2022) reported only 39.69% complete IRF sequences; Estaji & Mirzaei (2022) explicitly state that IRF represents a structural mode rather than a fixed ratio; Hidayatullah (2024) also highlights that the proportions of I/F-dominant segments may vary substantially due to teacher style.

Based on these insights, **we removed potentially misleading terms such as “minor difference” or “close” in the revision**. We now explicitly define our analysis goal as **assessing whether simulated classrooms consistently reproduce the structural trends of real classrooms**, rather than reproducing identical values.

**2. Quantitative–qualitative integration: clarified table purposes and added IRF examples.**

In response to reviewers’ questions regarding the statistical tables, we clarified the distinct roles of each table:

- Table 1 presents **overall trends** across environment × text genre;
- Table A4 (revised version Table A7) presents **fine-grained variations within individual sessions**.

These correspond to macro- and micro-level analyses, respectively.

Additionally, we added **multiple IRF example segments from both real and simulated classrooms in Appendix Table A8**, together with explicit IRF coding rules (e.g., Initiation boundaries, multi-round Feedback structures).

These examples show that differences stem mainly from teacher expression styles: simulated teachers tend to offer direct feedback, while real teachers frequently employ guiding questions. These stylistic differences are explainable and do not undermine structural consistency (details in the revised main text **Section 4.1**).

**3. Real classroom sources and annotation workflow fully disclosed to improve reproducibility.**

All real classroom data come from the National Smart Education Platform, with text genres matched one-to-one with the simulated lessons. In **Section D.3.2** of the revised appendix, we added details of **our two-stage “human verification + model-assisted” annotation pipeline**, along with the full IRF coding guidelines (time-step segmentation, boundary rules, multi-turn feedback categorization, etc.). We also included raw IRF excerpts from the real classrooms in **Appendix Table A8**, allowing direct side-by-side comparison.

In summary, **simulated classrooms consistently reproduce the IRF structural characteristics of real classrooms across environments and text genres**. Observed numerical differences are attributable to teacher style rather than systematic deviation. These additions directly address reviewers’ concerns and strengthen the clarity and evidential grounding of our contribution on authentic classroom structure reconstruction.

[1] Zaswita, H. (2022). Classroom Turn-Taking Process: A Study of IRF (Initiation-Reply-Feedback). Journal of Teachinf and Learning, 7(2), 102-109.

[2] Estaji, M., & Mirzaei Shojakhanlou, M. (2022). Realization of initiation, response, and feedback in teacher-student interactions in EFL classrooms: Learning realities and opportunities. Journal of English Language Teaching and Learning, 14(30), 91-114.

[3] Hidayatullah, E. (2024). Analyzing classroom interactions focusing on IRF patterns and turn-taking. English Learning Innovation (englie), 5(2), 186-196.

---

### Author Response · Authors · 2025-11-24
**Thank you to Reviewers & Rebuttal Summary**

**We sincerely thank all five reviewers for their careful evaluation and constructive comments**. This paper introduces EduVerse, a user-defined multi-agent classroom simulation environment that models virtual students with personalities, cognitive chains, and evolving behaviors through the Cognition–Interaction–Evolution (CIE) framework, aiming to simulate authentic classroom interaction structures and learning processes. **We appreciate the reviewers’recognition of the value of this work from various perspectives**, including:

- **A forward-looking research direction with strong educational significance** (Reviewer pnJm, Reviewer 2hzb, Reviewer LQGY)
- **A clear modeling framework with solid theoretical grounding** (Reviewer pnJm, Reviewer 2hzb, Reviewer Zq4a, Reviewer LQGY)
- **Innovative and practically promising Human-in-the-loop design** (Reviewer jPWs, Reviewer pnJm, Reviewer 2hzb)
- **Comprehensive experimental design with detailed classroom modeling** (Reviewer Zq4a, Reviewer LQGY)
- **Systematic review of related work and in-depth theoretical analysis** (Reviewer 2hzb)

We are deeply grateful for these positive evaluations and acknowledgments.

Building on this strong overall support, we note that reviewers’ main concerns converge around two key areas, corresponding to our Global Response for **Common Concern 1** and **Common Concern 2**:

- **Details regarding the comparison between real and simulated classrooms** (Reviewer jPWs, Reviewer pnJm, Reviewer Zq4a):

Reviewers requested clarification on how IRF ratios are interpreted, how quantitative and qualitative analyses are combined, the transparency of real classroom data sources and annotation procedures, and the relationship among different statistical tables. These points are addressed in Common Concern 1.

- **Cross-disciplinary adaptability and generalization of the system** (Reviewer jPWs, Reviewer 2hzb, Reviewer Zq4a):

Reviewers were interested in whether EduVerse can maintain stable structural and behavioral consistency beyond Chinese language classes, and whether the CIE framework can generalize across different subject domains. These issues are covered in Common Concern 2.

To address these core concerns, **we have added several clarifications and supplementary analyses in the revised manuscript**. These additions not only respond directly to reviewers’ questions but also strengthen the logical coherence and rigor of the paper:

- **Stronger evidence for classroom authenticity**:

We clarified the goal of the IRF analysis, added details about the real classroom sources, annotation workflow, and coding criteria, and included example IRF segments from both real and simulated classrooms. These additions provide more concrete evidence for evaluating whether EduVerse can faithfully reproduce classroom interaction structures.

- **Clearer demonstration of cross-disciplinary adaptability**:

A newly added history-class experiment includes IRF distributions, student social structures, and BEC trajectory analyses, showing that the model maintains structured interaction patterns and behavioral consistency in a different subject domain. This further validates the generality of the CIE framework and the transferability of the system.

Overall, these enhancements reinforce the paper' s core contributions in **classroom structure reconstruction**, **multi-agent interaction modeling**, and **cross-scenario generalization**.

Beyond addressing these two common concerns, we have also provided point-by-point responses to each reviewer' s specific comments to ensure that every question receives a clear and direct reply. We sincerely hope that these clarifications resolve the reviewers' concerns, and we welcome further feedback to ensure a shared understanding of this work' s contributions and to move toward a mutually agreed conclusion. **All newly added content is marked in blue in the revised manuscript for easy reference**.

Once again, we sincerely thank all reviewers for their invaluable suggestions throughout this process.

---

### Note · Authors · 2025-12-03

**Comment:**

We sincerely thank the five reviewers for the time and effort they devoted to evaluating our submission, as well as for the positive recognition of the work’s *forward-looking motivation*, *systematic modeling framework*, *rigorous experimental design*, and *human-in-the-loop evaluation*.

EduVerse, as **one of the first user-defined multi-agent classroom simulation frameworks**, makes three core contributions:
* **It constructs a reusable, extensible, and cross-disciplinary classroom simulation environment** that supports flexible role configuration, instructional workflows, and spatial layouts;
* **It proposes the CIE (Cognition–Interaction–Evolution) architecture**, which systematically models virtual students’ cognitive chains, interaction patterns, and multi-session evolutionary processes, enabling interpretable internal agent states;
* **It aligns simulated classrooms with real classroom data** through IRF structural analysis, BEC trajectory modeling, and social network characterization, and further supplements this with historical-classroom experiments to validate structural consistency, interpretability, and cross-disciplinary generalization.

**The work aims to evaluate the structural authenticity, behavioral mechanisms, and evolutionary logic of a foundational simulation framework**, whereas some comments adopt the expectations of an applied teaching system or user-study-oriented paradigm, leading to a mismatch in evaluation criteria. To address these differences, we clarified and supplemented the manuscript and rebuttal with:
* the sources of real classroom data and the IRF human annotation procedure;
* coding guidelines and IRF structural comparison examples;
* additional cross-disciplinary experimental results;
* strengthened explanations linking the CIE architecture with key structural indicators.

At the same time, due to the large-scale anonymity breach that occurred at ICLR this year, the conference implemented special policies to preserve fairness in the review process. These policies prevented further author–reviewer clarification in later stages. **Within this objective constraint, we recognize that the limited communication space in the current mechanism makes it difficult for a foundational framework like this one to have its structural contributions and evidence chain fully presented and fairly evaluated.** Out of respect for both scientific rigor and the review process, we have decided to voluntarily withdraw the submission. We will continue to refine the manuscript based on the reviewers’ feedback and rebuttal clarifications, and subsequently resubmit it to a venue better suited for framework-oriented contributions.

**Withdrawal Confirmation:**

I have read and agree with the venue's withdrawal policy on behalf of myself and my co-authors.